# MODUS: Decoder-Only Any-to-Any Modeling of Diverse Modalities

**Mingqiao Ye** [* 1]   **Zhaochong An** [* 1 3]   **Zhitong Gao** [1]   **Xian Liu** [4]   **Oğuzhan Fatih Kar** [† 1 2]   **Jesse Allardice** [† 2]
**Roman Bachmann** [† 1 2]   **David Mizrahi** [† 2]   **François Fleuret** [5]   **Chuan Li** [6]   **Amir Zadeh** [6]   **Serge Belongie** [3]
**Afshin Dehghan** [2]   **Amir Zamir** [1]

https://modus-multimodal.epfl.ch/

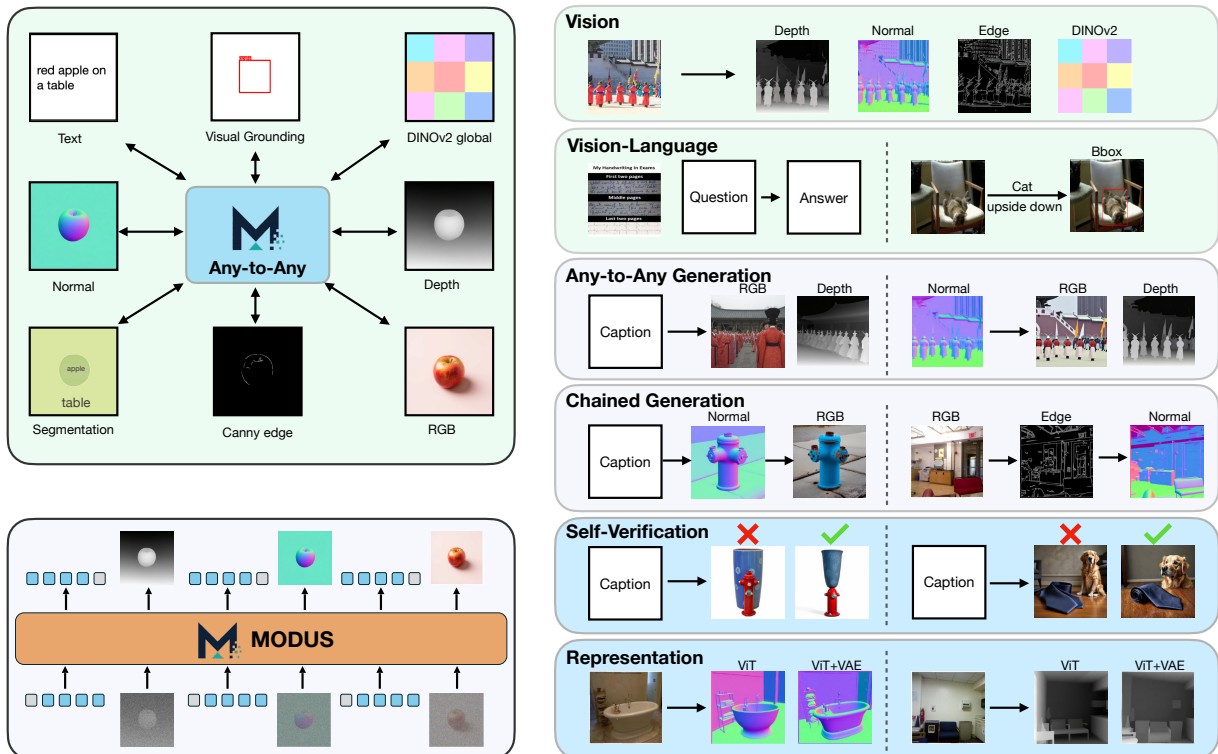

*Figure 1.* (1) Top left: **Any-to-Any**. MODUS performs any-to-any modeling, *i.e.*, predicting any modality given any other, and supports diverse 1D and 2D modalities. (2) Bottom left: **Decoder-Only.** All modalities are processed and generated within a single decoder using a unified autoregressive token sequence without modality-specific heads. (3) Right: **Multimodal Capabilities.** The resulting model can support a wide range of capabilities in a single model: core vision tasks such as converting RGB to depth, normals, edges, or DINO features; vision–language tasks including VQA, and visual grounding; flexible any-to-any generation across modalities; chained generation through intermediate modalities; self-verification, where one modality can evaluate the output of another; and visual representation analysis, where semantic features and reconstruction latents jointly contribute to faithful generation.

## Abstract

Any-to-any modeling aims to flexibly relate arbi-

[*]Equal contribution [†]Equal technical advising [1]EPFL, Lausanne, Switzerland [2]Apple [3]University of Copenhagen, Copenhagen, Denmark [4]The Chinese University of Hong Kong, Hong Kong, China [5]University of Geneva, Geneva, Switzerland [6]Lambda AI. Correspondence to: Mingqiao Ye <mingqiao.ye@epfl.ch>.

*Proceedings of the $43^{rd}$ International Conference on Machine Learning*, Seoul, South Korea. PMLR 306, 2026. Copyright 2026 by the author(s).

trary modalities within a single system, a requirement that arises across multimodal learning and scientific domains such as ecology and astronomy. However, existing any-to-any approaches are typically trained from scratch using encoder–decoder or diffusion architectures, limiting empirical performance and the use of pretrained models. We investigate **decoder-only any-to-any multimodal modeling**, which treats all modalities symmetrically and supports arbitrary modalities as in-

puts and outputs without modality-specific heads, losses, or task pipelines. As a consequence of this unified design, the resulting model MODUS naturally enables chained generation through intermediate modalities, cross-modal consistency verification, and analysis of visual representations by combining semantic and reconstruction features. Across a range of benchmarks, MODUS demonstrates strong out-of-the-box performance and flexible multimodal composition within a single model.

## 1. Introduction

Any-to-any modeling addresses the need to flexibly map between arbitrary modalities. In multimodal learning, this paradigm has been explored by models such as Unified-IO (Lu et al., 2022) and 4M (Mizrahi et al., 2023), which aim to unify diverse tasks and modality combinations within a single framework. Beyond that, similar any-to-any requirements naturally arise in scientific domains such as genomics (Nair et al., 2025), ecology (Sastry et al., 2025), and astronomy (Parker et al., 2025), where diverse data sources can be related in a compositional and reusable manner. These applications highlight that any-to-any modeling is not a niche capability, but a general and widely applicable requirement for multimodal systems.

While existing any-to-any systems demonstrate that task-agnostic modeling across modalities is feasible, they are typically trained from scratch using encoder–decoder or diffusion-based architectures. This training paradigm requires jointly learning modality structure and cross-modal alignment without leveraging the strong priors available in large pretrained models, leading to higher training costs and limited scalability. As a result, despite their conceptual generality, these approaches struggle to match the semantic fidelity, generalization, and efficiency of large-scale pretrained foundation models.

Decoder-only architectures provide a compelling paradigm due to their unified next-token prediction objective, strong zero-shot generalization, and efficient inference enabled by mechanisms such as KV-caching. Their success in large-scale language modeling (Hurst et al., 2024; Touvron et al., 2023; Team et al., 2023) and subsequent extension to image–text generation (Lin et al., 2023; Chen et al., 2024; Deng et al., 2025) demonstrates that a single autoregressive decoder can effectively leverage large-scale pretrained priors when extending from text to image modalities. These properties make decoder-only models an attractive candidate for realizing any-to-any multimodal generation at scale.

Despite their strengths, existing decoder-only multimodal models remain largely confined to limited modalities, most commonly treating text and RGB images as privileged inputs or outputs. While recent systems extend decoder-only architectures to additional modalities (Zhan et al., 2024; Peng et al., 2023), they often rely on modality-specific heads, task-dependent losses, or text-centric pivots, limiting their ability to support arbitrary modality-to-modality generation within a single symmetric framework. As a result, the potential of decoder-only models for any-to-any multimodal generation remains underexplored.

We introduce MODUS, a decoder-only architecture for any-to-any multimodal generation. The model treats all modalities symmetrically and avoids modality-specific components, losses, or task pipelines, allowing arbitrary modality pairs to serve as conditioning inputs or generation targets within a single model. Beyond text and RGB images, MODUS supports geometric modalities such as depth (Yang et al., 2024) and surface normals (Ke et al., 2025), structural modalities such as edge maps, semantic modalities such as segmentation masks and object grounding (Ren et al., 2024), and representational modalities such as DINOv2 features (Oquab et al., 2023), thereby extending decoder-only modeling to a broader multimodal setting.

MODUS is a unified framework that builds upon BAGEL (Deng et al., 2025) and can jointly handle 1D sequential modalities and 2D spatial modalities within a single decoder. Discrete modalities such as text, object grounding (Liu et al., 2024), and self-supervised feature representations (Oquab et al., 2023) are modeled autoregressively using next-token prediction, while continuous spatial modalities including RGB images, depth (Yang et al., 2024), surface normals (Ke et al., 2025), segmentation masks (Ren et al., 2024), and edge maps are generated using flow matching in latent space (Lipman et al., 2022). During training, we randomly sample subsets of modalities as conditioning inputs and use instruction-following prompts to specify the target modality, enabling flexible any-to-any mappings.

To train MODUS at scale, we construct SPECTRUM-25M, a 25M-sample multimodal corpus that aligns geometric (depth, surface normals), structural (canny edges), semantic (segmentation, grounding boxes), and representational (DINOv2 features) annotations on a shared image base derived from BLIP-3o (Chen et al., 2025a). This alignment enables training across arbitrary (input, target) modality pairs, including transformations rarely covered by existing corpora such as depth $\rightarrow$ canny edge or canny $\rightarrow$ surface normal. To stabilize joint training across multiple modalities, we adopt uniform timestep sampling for flow matching and a staged training procedure that efficiently extends the model to diverse modalities. Uniform timestep sampling improves instruction adherence by providing balanced supervision across diffusion steps and reducing modality confusion during generation. The staged training procedure progressively

expands the set of supported modalities and the number of conditioning modalities per sample. Together, these design choices improve cross-modal alignment, long-context handling, and chained any-to-any generation.

The unified decoder-only design of MODUS gives rise to four composable capabilities within a single model:

**Any-to-any Generation.** MODUS maps arbitrary modality pairs within one architecture. Unlike task-specific approaches that require $\mathcal{O}(n^2)$ models to cover $n$ modalities, MODUS scales linearly with the number of modalities and supports multi-target generation without severe degradation.

**Chained Generation.** Generated outputs can be reused as conditioning inputs, allowing tasks to be solved through intermediate modalities (*e.g.*, RGB $\rightarrow$ canny edge $\rightarrow$ normal). Our analysis (Section 4.3) shows that this provides a structured way to inject *complementary, spatially-aligned* signal into a generation task.

**Cross-modal Verification.** The model can generate an auxiliary modality (*e.g.*, grounding, VQA) conditioned on each candidate output and select the most cross-modally consistent one, without external verifiers.

**Visual Representation Composition.** For each 2D modality, MODUS composes complementary semantic (ViT) and reconstruction (VAE) features within the unified sequence. Single-feature variants reveal a clear trade-off: ViT-only conditioning hallucinates geometry, VAE-only conditioning loses semantics, and jointly using both yields more faithful 2D outputs than either alone.

***Conflict of Interest Disclosure.*** One of the models this paper evaluates is 4M, which was developed by part of the authorship team. 4M is fully open-sourced.

## 2. Related Work

### 2.1. Decoder-Only Multimodal Models

Decoder-only transformer architectures (Hurst et al., 2024; Touvron et al., 2023; Team et al., 2023) have become a dominant paradigm due to their scalability, unified training, and strong zero-shot generalization. Early multimodal extensions such as LLaVA (Liu et al., 2023), MiniGPT-4 (Zhu et al., 2023), InternVL (Chen et al., 2024), and Qwen-VL (Bai et al., 2025) couple visual encoders with large language decoders to enable multimodal understanding (Li et al., 2026; 2025), but their outputs remain text-only. More recent decoder-only models, including Chameleon (Team, 2024), Show-O (Xie et al., 2024), EMU (Wang et al., 2024), Janus (Wu et al., 2024a), Janus-Pro (Chen et al., 2025b), Janus-Flow (Ma et al., 2025b), BLIP-3o (Chen et al., 2025a), and BAGEL (Deng et al., 2025), unify text and image generation within a single backbone. However, these models

primarily focus on text and RGB images and do not support general any-to-any generation across various modalities.

### 2.2. Any-to-Any Models

Any-to-any models are designed to support flexible mappings between multiple modalities within a shared architecture. Encoder–decoder systems such as Unified-IO (Lu et al., 2022) and 4M (Mizrahi et al., 2023), as well as unified diffusion models like OneDiffusion (Le et al., 2025), demonstrate that task-agnostic modeling across modalities is achievable using modality-agnostic objectives. Similar any-to-any trends also arise in other subjects, with recent models such as AION-1 (Parker et al., 2025) in astronomy, ProM3E (Sastry et al., 2025) in ecology, and Nona (Nair et al., 2025) in functional genomics. However, these approaches are typically trained from scratch, requiring the joint learning of modality structure and cross-modal alignment without leveraging large-scale pretrained foundation models, which can limit scalability and semantic fidelity. In parallel, LLM-centric approaches such as NExT-GPT (Wu et al., 2024b) and AnyGPT (Zhan et al., 2024) integrate diverse modalities by bridging them through a language model, training primarily on Text-to-Any and Any-to-Text tasks. While effective for semantic alignment, this text-centric paradigm constrains native modality-to-modality interactions. In contrast, MODUS emphasizes architectural and procedural unification within a single decoder-only model, enabling flexible any-to-any generation without predefined tasks or reliance on a textual bridge. Concurrent work, Vision-Banana (Gabeur et al., 2026), similarly argues that image generation can serve as a universal interface for diverse vision tasks, reinforcing our view that generative modeling is a strong foundation for unified visual capability. MODUS extends this perspective to any-to-any modeling across both 1D and 2D modalities, including text, RGB, depth, surface normal, edges, segmentation, grounding, and DINOv2 features, with explicit support for chained generation and cross-modal self-verification.

## 3. Method

In this section, we present the design of MODUS, a unified decoder-only architecture for any-to-any multimodal generation. We first introduce a unified tokenization scheme for diverse modalities (Section 3.1), then describe the architectural extensions that enable processing different modalities within a single decoder (Section 3.2). We next present a stabilized training strategy that mitigates modality mixing during joint optimization (Section 3.3). We then describe the SPECTRUM-25M dataset that enables this training paradigm at scale (Section 3.4). Finally, we describe the inference-time procedures (Section 3.5).

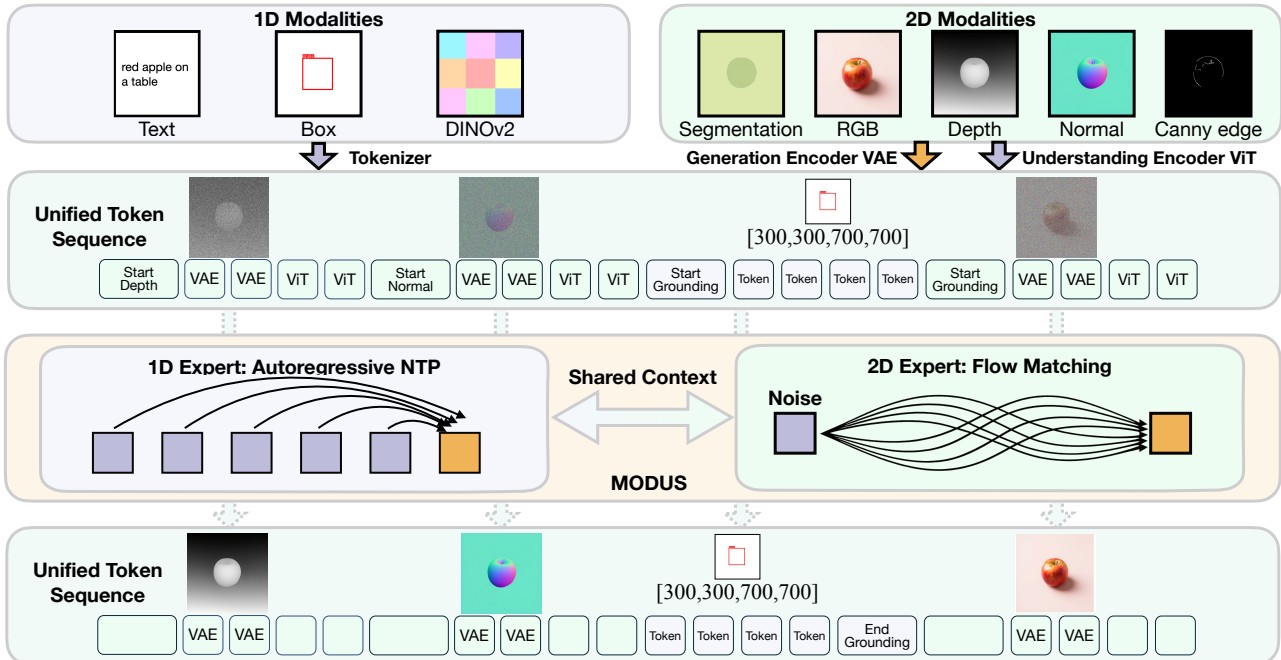

Figure 2. **Overview of the MODUS framework. (a) Modalities.** MODUS supports 1D sequential modalities including text, visual grounding boxes, and DINOv2 global tokens, and 2D spatial modalities including RGB images, depth, surface normals, segmentation, and canny edges. **(b) Unified tokenization.** All modalities are mapped into a shared sequence with modality instructions. 1D modalities use modality-specific tokenizers. 2D modalities are represented by VAE reconstruction latents and ViT semantic features, with noise added to VAE latents for denoising. **(c) Decoder-only architecture.** MODUS employs two experts within a single decoder, a 1D Expert for discrete modalities trained with autoregressive next-token prediction and a 2D Expert for continuous spatial modalities trained with flow matching in latent space. Inside each modality, 1D modalities use causal attention and 2D modalities use bidirectional attention. Among modalities, MODUS maintains a shared causal attention context. **(d) Outputs and objectives.** The 1D Expert predicts the next token with a cross-entropy loss. The 2D Expert predicts the velocity field with a mean squared error loss.

## 3.1. Unified Tokenization Scheme

We design a unified tokenization scheme that represents various modalities in a single sequential format. Common modalities can be divided into two groups: *sequential 1D modalities*, which can be modeled through next-token prediction, and *spatial 2D modalities*, which can be generated using flow-matching denoising.

**1D Modality Representation.** 1D modalities include text, image captions, detection bounding boxes, and DINOv2 (Oquab et al., 2023) global features. To represent them, we use different discrete tokenizers to obtain token sequences. For text inputs such as VQA questions and captions, we use a standard text tokenizer (Bai et al., 2023). For bounding boxes, we normalize the coordinates to the range $[0, 1000)$ and represent them in the x1 y1 x2 y2 format, using 1000 discrete tokens for each coordinate. For DINOv2 global features, we adopt an MLP-based tokenizer following (Mizrahi et al., 2023) with a codebook size of 8192, producing 16 tokens per image.

**2D Modality Representation.** 2D modalities include RGB images, monocular depth, surface normals, segmentation maps, and canny edge maps. These modalities share spatial structure and can be represented as 2D, RGB-like images.

To capture both semantic and reconstruction-level information, MODUS represents each 2D modality using a combination of semantic features and continuous reconstruction latents. Specifically, a pretrained SigLIP-2 (Tschannen et al., 2025) ViT encoder extracts high-level semantic features, while a pretrained VAE from FLUX (Batifol et al., 2025) encodes the same input into continuous latent representations. The VAE latents serve as the target space for generation and are modeled using a flow-matching objective, enabling high-fidelity spatial synthesis across modalities. For segmentation, we additionally prepend the mask category as text tokens to the sequence.

**Unified Sequence Format.** The unified sequence can be formed from any subset of 1D and 2D modalities. Although all modalities share this unified sequence, the tokenization mechanism differs by type: 1D modalities use modality-specific discrete tokenizers, while 2D modalities use the shared ViT–VAE dual representation described above. During training, one modality is designated as the generation target, while between 1 and 3 of the remaining modalities (sampled randomly) serve as conditioning inputs; this exposes the model to both single-condition and multi-condition generation within the same training loop. We define the training objective as:

$$[\text{Cond}_1], [\text{Cond}_2], \ldots \rightarrow [\text{Target}] \tag{1}$$

At inference time, the model can generate arbitrary target modalities given any combination of conditioning inputs, and supports chained generation across modalities:

$$[\text{Cond}_1] \rightarrow [\text{Target}_1]$$
$$[\text{Cond}_1], [\text{Target}_1] \rightarrow [\text{Target}_2] \tag{2}$$

### 3.2. Any-to-Any Decoder-Only Architecture

MODUS is a decoder-only architecture for *any-to-any multimodal generation*, where arbitrary combinations of modalities can act as conditioning inputs and prediction targets within a single model. Instead of introducing modality-specific heads, task pipelines, or separate generators, MODUS adopts a decoder-only Mixture-of-Transformers structure and incorporates new modalities by unifying their training objectives and token representations within the same sequence modeling framework. This design preserves the simplicity and strong pretrained priors of decoder-only foundation models, while demonstrating that expressive any-to-any multimodal behavior can be achieved without architectural specialization.

**Unified Decoder Model.** Concretely, MODUS adapts the pretrained BAGEL-7B (Deng et al., 2025) Mixture-of-Transformers architecture and incorporates two experts within a single autoregressive decoder: a *1D Expert* for sequential modalities and a *2D Expert* for spatial modalities. The two experts maintain independent parameters but operate over a shared autoregressive token sequence and attend to the same causal self-attention context, such that tokens generated by either expert can condition subsequent tokens. This unified decoding behavior enables flexible modality composition, forming the architectural basis for any-to-any multimodal generation.

**Sequential and Spatial Modality Modeling.** The 1D Expert handles discrete sequential modalities, including text, object grounding, and self-supervised features (Oquab et al., 2023), using next-token prediction with causal attention. The 1D Expert captures long-range dependencies and supports semantic reasoning. Given a token sequence $x_{1:N}$, the training objective is

$$\mathcal{L}_{\text{NTP}} = -\sum_{i=1}^{N} \log p_\theta(x_i \mid x_{<i}), \tag{3}$$

The 2D Expert models continuous spatial modalities, including RGB images, depth maps, surface normals, segmentation masks, and edge maps, using flow matching (Lipman et al., 2022). Let $x_0$ denote a clean latent representation of a 2D modality encoded by the VAE. A noisy latent $x_t$ is constructed at timestep $t$ by perturbing $x_0$ with Gaussian noise. The model learns a velocity field $v_\theta(x_t, t)$ that

matches the time derivative $\dot{x}_t$ of the latent trajectory. The flow-matching objective is given by

$$\mathcal{L}_{\text{FM}} = \mathbb{E}_{t, x_0, \epsilon} \left[ \| v_\theta(x_t, t) - \dot{x}_t \|_2^2 \right]. \tag{4}$$

**Token Routing.** Each token's path depends on its modality type and role. 1D tokens (text, grounding, DINOv2) flow through the 1D Expert with causal attention and the NTP loss. For 2D modalities, ViT semantic tokens flow through the 1D Expert with intra-modality bidirectional attention, while VAE reconstruction tokens flow through the 2D Expert: clean as a condition, noised at a sampled timestep and supervised by flow matching as the target. Cross-modality attention remains causal so any token conditions all subsequent ones. At inference, condition tokens are encoded first to populate the KV cache before target tokens are decoded.

### 3.3. Training Strategies

**Timestep Sampling.** For flow matching training, we sample a timestep $t$ at each iteration. In standard diffusion generation models (Liu et al., 2025; Han et al., 2025b; An et al., 2026a;b), logit-normal sampling is commonly used since it improves generation quality (Esser et al., 2024). However, as shown in Figure 3, in our multimodal decoder-only model, we observe that logit-normal sampling frequently causes a *modality confusion issue*, where the model fails to follow the target instruction, *e.g.*, generating a normal map when the intended output is depth. We find that early timesteps usually determine the target modality distribution, while later timesteps mainly refine the visual quality. Since logit-normal sampling tends to oversample intermediate timesteps and underrepresent early ones, it leads to unstable modality output. To address it, we adopt uniform timestep sampling, which provides balanced training across all timesteps and effectively reduces modality confusion.

**Training Stages.** We initialize the model from the pretrained BAGEL-7B checkpoint, which supports image and text modalities, and train it through three progressive stages. In the first stage, training focuses on 1D modalities, including image captions, grounding bounding boxes, and DINOv2 global features (Oquab et al., 2023). For each sample, one modality is randomly selected as the conditioning input and another as the prediction target, encouraging the model to learn cross-modal alignment and shared priors. In the second stage, we incorporate 2D modalities such as depth, surface normals, segmentation, and canny edge maps. These spatial modalities exhibit faster convergence, benefiting from strong visual priors. In the third stage, we increase the number of conditioning modalities per sample to strengthen the model's ability to handle long-context and multi-condition generation. This stage further improves performance on complex chained generation across diverse modality combinations.

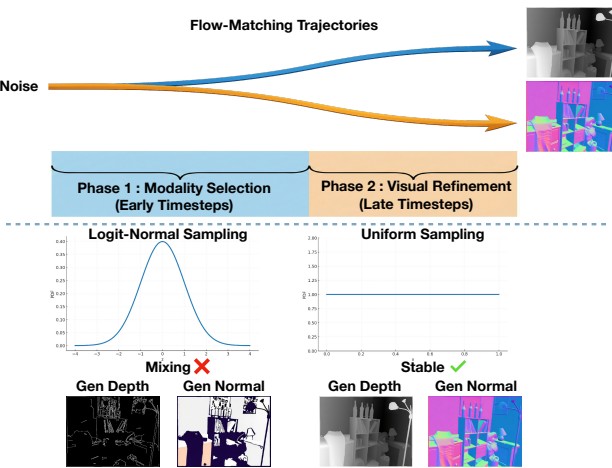

*Figure 3.* Top: **Multimodal Flow Matching.** Early timesteps play a crucial role in determining the target modality, whereas later timesteps primarily refine visual quality. Bottom: **Timestep Sampling.** Logit-normal sampling undersamples these early steps and oversamples intermediate ones, which causes the model to drift between modalities and produce mixed outputs. Uniform timestep sampling ensures balanced exposure across the full trajectory, enabling the model to reliably follow the target modality instruction and avoid mixing effects.

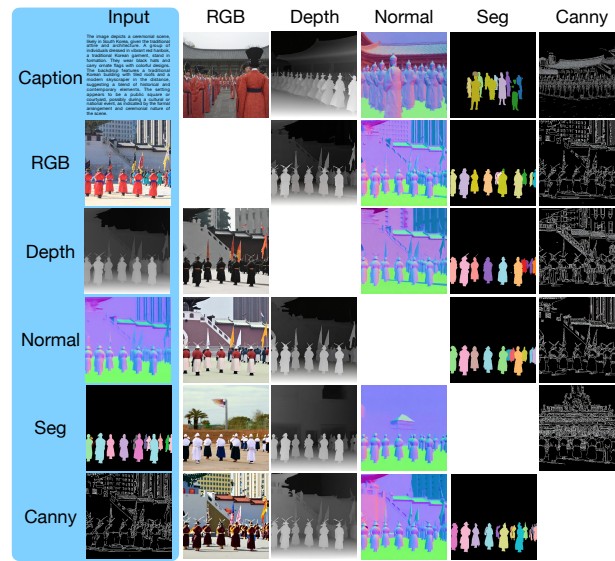

*Figure 4.* **Visualization of Any-to-Any Generation.** Given any input modality, MODUS generates all other target modalities within a single unified architecture. Each row uses the same scene rendered in a different input modality (leftmost column); the diagonal is intentionally blank (input modality = target modality).

### 3.4. SPECTRUM-25M Dataset

Any-to-any training requires all modalities to be aligned on the same underlying samples; each SPECTRUM-25M training sample is therefore an aligned tuple of one image together with annotations for all its modalities (caption, grounding boxes, depth, surface normals, segmentation, canny edges, and DINOv2 global tokens). We construct SPECTRUM-25M by extending BLIP-3o's image–caption corpus (Chen et al., 2025a) with aligned annotations from off-the-shelf experts: DepthAnything (Yang et al., 2024) for depth, Marigold (Ke et al., 2025) for surface normals, Grounded-SAM (Ren et al., 2024) for segmentation, GLaMM (Rasheed et al., 2024) for grounding boxes, and DINOv2 (Oquab et al., 2023) for representational features. The resulting corpus jointly covers 1D modalities (captions, grounding boxes, DINOv2 global tokens) and 2D modalities (RGB, depth, normals, segmentation, canny edges), enabling joint training over arbitrary (input, target) modality pairs as well as multi-condition and chained compositions. Detailed dataset statistics are in Appendix Section E.

### 3.5. Inference

During inference, the model first encodes all conditioning modalities to construct the KV-cache. The 1D Expert autoregressively predicts discrete tokens, whereas the 2D Expert follows the flow-matching trajectory by predicting velocities over a fixed set of denoising steps. Classifier-Free Guidance is applied to improve visual quality for 2D outputs.

Chained generation is enabled by reusing generated out-

puts as conditioning inputs for subsequent decoding steps. Since all modalities share a unified tokenized representation and are processed by the same decoder, outputs from one modality can be directly reused to generate another modality, enabling multi-step generation without retraining or architectural changes. The unified modeling also supports cross-modal self-verification. Given multiple candidate outputs, MODUS generates auxiliary modalities conditioned on each candidate, such as grounding or VQA, and selects more coherent results based on cross-modal agreement, without relying on external verifiers or separate scoring models.

## 4. Experiments

### 4.1. Implementation Details

We train MODUS on SPECTRUM-25M (Section 3.4), whose modality alignment supports cross-modal training and inference patterns difficult to study with conventional datasets, such as depth → canny or canny → surface normal. Training takes approximately 5,664 GH200 GPU-hours across three stages (35h, 31h, and 22.5h on 64 GPUs). We present more training details in Appendix Section E.

### 4.2. Zero-Shot Generation

We qualitatively evaluate the capability of MODUS to perform flexible any-to-any multimodal generation across a diverse set of modalities. Figures 4–6 present representative zero-shot visualizations demonstrating the model's unified generative behavior.

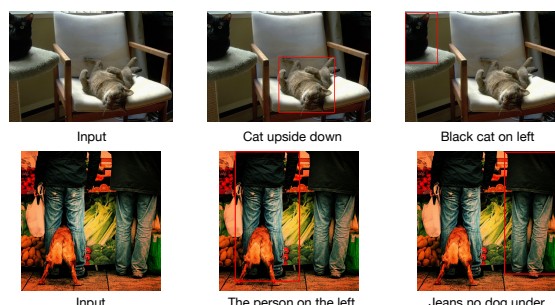

*Figure 5.* **Visualization of Zero-shot Grounding Results.** Given an image and a text query, MODUS predicts the corresponding region in a zero-shot manner within the unified decoder-only model.

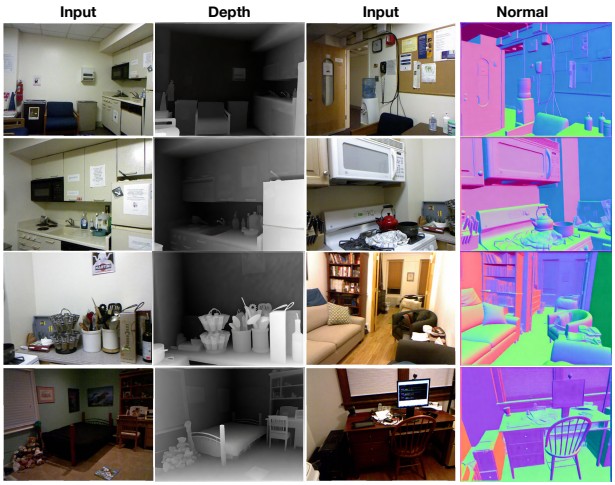

*Figure 6.* **Zero-shot Depth and Normal Estimation**. Visualization on the NYUv2 Dataset.

**Any-to-any generation.** As shown in Figure 4, MODUS can take an arbitrary input modality and generate all other target modalities within the same architecture. The results demonstrate consistent any-to-any translation across diverse representations.

**Visual grounding and geometric understanding.** Figure 5 shows zero-shot grounding results, where the model localizes text-specified regions in complex natural images. Figure 6 presents zero-shot depth and surface normal estimation on the NYUv2 dataset. In both cases, MODUS generalizes to unseen data and produces geometrically coherent predictions.

**System-level comparison.** As shown in Table 1, we present a system-level comparison between MODUS and other any-to-any and decoder-only models. Encoder–decoder any-to-any architectures are typically trained with masked autoencoding objectives and perform well on reconstruction tasks, but they are less effective for open-ended generation. Diffusion models excel at image generation but often struggle with text understanding and compositional generation. Pre-

vious decoder-only models primarily focus on image and text modalities, whereas our MODUS unifies all modalities within a single model for any-to-any generation, achieving strong zero-shot performance. Furthermore, most existing benchmarks are RGB- or text-centric, while MODUS supports a broader set of modality transformations beyond these settings, such as transforming canny edge to depth; quantitative results for such non-standard any-to-any and multi-condition mappings (*e.g.*, canny→depth, normal→depth, RGB+canny→depth) are reported in Section C.4.

### 4.3. Chained Generation

Because all modalities are produced by the same decoder, generated outputs can be reused as conditioning inputs without retraining or architectural changes. This raises a question that any-to-any systems are well positioned to study: *which intermediate modalities actually carry complementary information for a given target?*

We probe this on surface normal estimation by routing through three candidate intermediates: canny edges, depth, and DINOv2 global features (Table 2). The results reveal a clear pattern. **Spatially-aligned complementary cues help:** routing through canny edges improves normal prediction, since edges provide pixel-aligned low-level geometry that complements surface orientation. **Redundant cues do not:** depth offers no gain over the direct path, since depth and normals encode closely related geometric information. **Non-aligned semantic cues do not:** DINOv2 global features (Oquab et al., 2023) capture high-level semantics but are neither pixel-aligned nor geometrically informative.

The takeaway is not that "more steps are better," but that chaining provides a structured mechanism for injecting *complementary, spatially-aligned* signal, a study that requires exactly the modality composability our framework provides. As shown in Figure 7, the intermediate and final outputs remain visually consistent across these chains. Additional visualizations are in Appendix Section D.1.

**Caption-to-Any consistency.** Direct caption-to-target generation can produce visually divergent outputs across modalities (Figure 4), since caption conditioning is inherently sparse and leaves many visual details unspecified. Chained generation directly addresses this: by routing through an intermediate modality (*e.g.*, Text → Edge → RGB), the intermediate fixes the spatial layout before generating the final output, producing stronger structural consistency across modalities (see Appendix Section D.1).

**Inference cost.** Chaining trades additional decoding for higher fidelity; we report direct-vs.-chained inference-efficiency benchmarks in Section C.7.

*Table 1.* **Zero-shot Benchmark across Tasks.** ↑ Higher is better. ↓ Lower is better. ✗ denotes models that cannot solve the task out-of-the-box. The first groups present existing any-to-any models, including encoder–decoder and diffusion approaches. The last group shows decoder-only models. MODUS extends decoder-only models from image–text settings to diverse modalities and operates on all benchmarks in a zero-shot manner. It remains competitive with multitask any-to-any baselines and decoder-only image–text models, while supporting broader multimodal capabilities (any-to-any, chained generation, cross-modal verification) that prior systems do not. † results reproduced by us.

| | Modality
Task
Benchmark | RGB→Text
VQA
MMMU↑ | Text→RGB
T2I
GenEval↑ | RGB→Depth
Depth Est.
DIODE↓ | RGB→Normal
Normal Est.
NYUv2↓ | RGB→Det
Grounding
RefCOCO$_{val}$↑ | RGB→DINO
Retrieval
ImageNet↑ |
|---|---|---|---|---|---|---|---|
| Encoder-Decoder | 4M-21 (Bachmann et al., 2024) | ✗ | 0.37 | 0.331 | 37.28 | ✗ | 78.3 / 92.4 |
| | Unified-IO 2 (Lu et al., 2024) | – | – | 0.369 | 28.55 | – | ✗ |
| Diffusion | OneDiffusion (Le et al., 2025) | ✗ | 0.65 | 0.399 | – | ✗ | ✗ |
| Decoder-only | Bagel† (Deng et al., 2025) | 53.2 | 0.86 | ✗ | ✗ | ✗ | ✗ |
| | Kosmos-2 (Peng et al., 2023) | – | ✗ | ✗ | ✗ | 52.3 | ✗ |
| | Janus-Pro (Chen et al., 2025b) | 41.0 | 0.80 | ✗ | ✗ | ✗ | ✗ |
| | GPT-4o (Hurst et al., 2024) | 69.1 | 0.84 | ✗ | ✗ | ✗ | ✗ |
| | **MODUS (Ours)** | 51.1 | 0.81 | 0.285 | 19.92 | 54.5 | 77.9 / 92.5 |

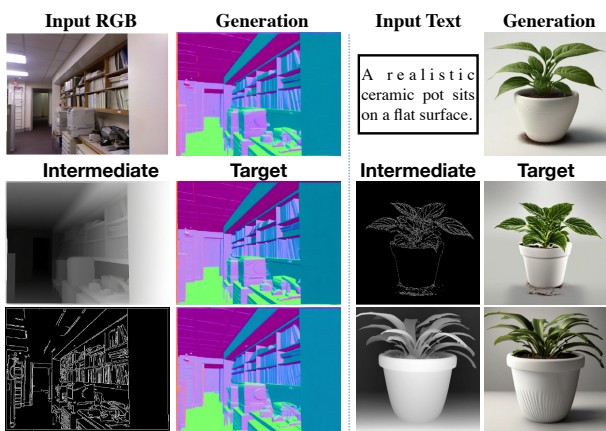

*Figure 7.* **Visualization of Chained Generation.** The first row shows direct generation, and the following rows show chained generation through intermediate modalities. The chaining enables consistent results across modalities.

*Table 2.* **Chained Generation** on NYUv2 surface normals.

| Source → Target | Intermediate | NYUv2 Normal ↓ |
|---|---|---|
| RGB → Normal | – | 20.02 |
| RGB → Depth → Normal | Geometry | 20.06 |
| RGB → DINO → Normal | Semantics | 20.71 |
| RGB → Canny → Normal | Layout | **19.87** |

## 4.4. Cross-Modal Verification

MODUS can leverage one modality to verify another, thereby further improving performance. We evaluate this cross-modal verification capability on image generation in Table 3. For each prompt we generate 4 images and use the model's own prediction scores to select the best candidate for evaluation. Both object grounding and VQA can improve image generation performance, demonstrating a compelling application of any-to-any modeling. Additional visualizations are presented in Appendix Section D.2.

*Table 3.* **Cross-Modal Verification**. We apply the verifier score to select the best-of-4 output on text-to-image generation.

| Verifier | GenEval ↑ |
|---|---|
| – | 0.81 |
| Object Grounding | 0.82 |
| VQA + Grounding | **0.84** |

*Table 4.* **Visual Representation Composition.** Depth estimation conditioned on different 2D feature representations. ViT provides high-level semantic features, while VAE provides low-level reconstruction features.

| ViT | VAE | NYUv2 Depth ↓ | NYUv2 Normal ↓ |
|---|---|---|---|
| ✓ | | 15.1 | 35.30 |
| | ✓ | 6.9 | 19.96 |
| ✓ | ✓ | **6.5** | **19.92** |

## 4.5. Visual Representation Composition

Our 2D modalities are represented using two complementary features: ViT features that capture high-level semantic structure and VAE features that encode low-level reconstruction details. During training, generation is conditioned on both features, with feature dropout applied to enable classifier-free guidance. To analyze each representation, we evaluate three conditioning schemes in Table 4: ViT-only, VAE-only, and ViT–VAE. The combined ViT–VAE setting achieves the best performance. As shown in Figure 8, conditioning solely on ViT features preserves coarse semantic identity but distorts fine-grained geometry; for example, in the highlighted regions, the predicted depth still corresponds to a chair, but its geometric shape is altered. In contrast, conditioning solely on VAE features maintains local geometric consistency but lacks semantic robustness; in the highlighted regions, the black display panel is incorrectly predicted as an empty cavity in the depth map.

Intriguingly, this characteristic error pattern has also been observed in recent work that evaluated GPT-4o (Ramachandran et al., 2025), suggesting that GPT-4o may have em-

| Input | ViT only | VAE only | ViT + VAE |

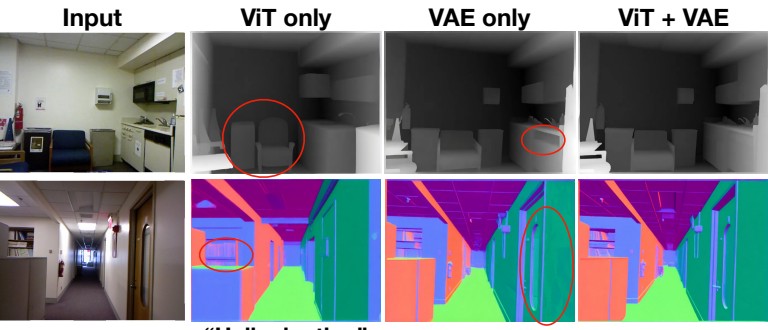

"Hallucination"

*Figure 8.* **Visual Representation Composition.** ViT-only conditioning produces plausible but less faithful outputs with structural changes. VAE features supply low-level details important for geometric consistency. Conditioning on both ViT and VAE yields the most accurate results.

*Table 5.* **Ablation on 2D Expert Timestep Sampling**. Logit-normal sampling undersamples early timesteps and leads to modality mixing, while uniform sampling provides balanced coverage and yields stable modality-correct outputs.

| TimeStep Sampling | GenEval ↑ | NYUv2 Depth ↓ | NYUv2 Normal ↓ |
|---|---|---|---|
| Logit-Normal | 0.77 | 21.9 | 50.54 |
| Mode | 0.80 | 8.6 | 22.47 |
| Uniform | 0.81 | 8.4 | 21.10 |

*Table 6.* **Ablation on Training Stages.** Our three-stage curriculum progressively expands MODUS's capabilities: Stage 1 trains 1D modalities, which converge slowly due to lacking strong priors; Stage 2 adds additional 2D modalities, improving performance across vision tasks; and Stage 3 increases conditioning inputs, enabling multi-condition and chained generation.

| Model | Training Tokens | MMMU ↑ | GenEval ↑ | NYUv2 ↓ | Retrieval Top-1 / Top-5 ↑ | Chained Gen. |
|---|---|---|---|---|---|---|
| Bagel | ∼5T | 53.2 | 0.86 | ✗ | ✗ | ✗ |
| MODUS-Stage1 | 30B | 51.4 | 0.81 | ✗ | 78.8 / 92.8 | ✗ |
| MODUS-Stage2 | 20B | 51.1 | 0.81 | 6.5 | 77.9 / 92.5 | ✗ |
| MODUS-Stage3 | 15B | 50.3 | 0.81 | 6.6 | 77.2 / 92.2 | ✓ |

ployed a similar higher-level feature conditioning. Additional visualizations are provided in Appendix Section D.3.

### 4.6. Ablations

**Timestep sampling.** As discussed in Section 3.3 and illustrated in Figure 3, timestep sampling is critical for training 2D experts. In Table 5, we compare logit-normal, mode, and uniform sampling under identical 6B token training budgets. Logit-normal sampling causes strong modality mixing: image-to-depth often collapses into other modalities (*e.g.*, normals, edges, RGB, segmentation). This arises because all modalities share the same noisy source distribution, and early timesteps contain weak modal boundaries; the model must learn these distinctions in the high-noise regime. Mode and uniform sampling allocate more probability to early timesteps, giving the model denser supervision where modality boundaries are most ambiguous. As a result, both strategies substantially reduce mixing and yield stronger performance. Additional visualizations of modality mixing are provided in Appendix Section B.1.

**Training stages.** As stated in Section 3.3, we train MODUS

using a three-stage curriculum. Stage 1 focuses on 1D modalities, including Grounding and DINOv2 (Oquab et al., 2023), which lack pre-existing priors in the model and require longer convergence times. In Stage 2, we incorporate additional 2D modalities, expanding the model's capabilities to various 2D tasks as demonstrated in Table 6. Finally, in Stage 3, we increase the number of conditioning inputs, enabling the model to perform multi-condition and chained generation. Training losses are in Appendix Section B.2.

**Additional analyses (appendix).** BAGEL + per-head baseline (Section B.4), image–text capability preservation (Section C.3), per-category GenEval breakdown (Section C.6), Janus-Flow generalization (Section B.5), and a contamination check (Section C.5).

## 5. Conclusion

We presented MODUS, a unified decoder-only model for any-to-any multimodal generation. MODUS extends the decoder-only paradigm beyond text and RGB images to support a wide range of modalities within a single model, while avoiding modality-specific heads or task pipelines through unified tokenization. Stable and scalable multimodal training is achieved using uniform timestep sampling to avoid modality mixing and a staged training procedure that efficiently extends the model to additional modalities and multi-condition settings. As a result, MODUS supports flexible any-to-any generation, chained generation, cross-modal self-verification, and visual representation analysis without additional architectural complexity. Across diverse benchmarks, MODUS demonstrates strong out-of-the-box performance and robust generalization across modality combinations. MODUS's current modality coverage is representative rather than exhaustive; extending to further modalities such as audio or 3D structure is mainly a matter of dataset construction and tokenization within the same framework, which we view as a promising direction for future work. This work highlights the potential of decoder-only architectures as a foundation for any-to-any multimodal modeling.

## Acknowledgments

We thank Jason Taskov, Kunal Pratap Singh and Muhammad Uzair Khattak for their valuable feedback on earlier versions of the manuscript. We acknowledge Lambda for supporting this paper through academic compute grant program, and a gift from Apple. This work was supported under project ID 43 as part of the Swiss AI Initiative, through a grant from the ETH Domain, with computational resources provided by the Swiss National Supercomputing Centre (CSCS) on the Alps infrastructure. This work has also received funding from the Swiss State Secretariat for Education, Research and Innovation (SERI). Zhaochong An and Serge Belongie are supported by funding from the Pioneer Centre for AI, DNRF grant number P1.

## Impact Statement

This paper aims to advance multimodal representation learning and generation through MODUS, a unified any-to-any model. Although the proposed model is not intended for negative use, its general-purpose design and support for diverse modality transformations allow it to be applied beyond the scenarios examined in this work. We position MODUS primarily as a research tool for exploring unified multimodal modeling and encourage thoughtful consideration when extending it to broader applications.

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

# Appendix

## Table of Contents

## A. Additional Related Work

**Decoder-only architectures** originated in language modeling and were later extended to vision–language tasks by connecting pretrained LLMs with visual encoders. Early systems such as LLaVA (Liu et al., 2023), Qwen-VL (Bai et al., 2025), and DeepSeek-VL (Wu et al., 2024c) follow this pattern: an image encoder (*e.g.*, CLIP (Radford et al., 2021) / SigLIP (Tschannen et al., 2025)) produces visual features that condition a pretrained GPT-style decoder, enabling image captioning, VQA, and multimodal instruction following. These works established that decoder-only LLMs can serve as a strong multimodal understanding engine, though their outputs remained text-only.

Subsequently, decoder-only transformers were explored for autoregressive image generation. Models such as LlamaGen (Sun et al., 2024) apply next-token prediction to VQ-based image tokens, while different tokenizers like VAR (Tian et al., 2024) Infinity (Han et al., 2025a) and FlexTok (Bachmann et al., 2025) increase perceptual fidelity and flexibility. Although these GPT-style generators showed competitive progress, AR image models still face challenges in capturing fine details compared to diffusion-based approaches, motivating further unification of understanding and generation.

A new class of models then aimed to unify image understanding and image generation within a single decoder-only backbone. Early examples include Chameleon (Team, 2024) and Show-O (Xie et al., 2024), which jointly model text and VQ-tokenized images. Later systems such as EMU-3 (Wang et al., 2024) and BLIP-3o (Chen et al., 2025a) improved visual representation quality by integrating pretrained vision encoders or hybrid continuous/discrete tokenizers, enabling stronger semantic reasoning and higher-quality generation.

A major refinement came from Janus (Wu et al., 2024a) and Janus-Pro (Chen et al., 2025b), which explicitly decouple visual understanding and generation: a semantic encoder (*e.g.*, SigLIP) produces embeddings for recognition tasks, while a VQ-based tokenizer serves the generation branch. JanusFlow (Ma et al., 2025b) further augments the generation pathway with a rectified-flow objective, improving realism while keeping the unified decoder-only architecture. Recent large-scale efforts such as BAGEL (Deng et al., 2025) and Hunyuan-Image (Cao et al., 2025) extend this paradigm using Mixture-of-Transformer-Experts, scaling decoder-only multimodal modeling to tens of billions of parameters while supporting both high-quality image synthesis and strong vision–language reasoning.

Decoder-only models are appealing because they scale effectively, inherit strong language understanding, support long context windows, and enable simple parameter sharing across tasks. However, existing models remain largely limited to image–text settings: they may accept other modalities, but they do not support general any-to-any generation across heterogeneous outputs. Our work addresses this gap by extending the decoder-only paradigm beyond photorealistic image and text modalities, supporting additional structured and pixel-dense outputs (*e.g.*, depth, normals, segmentation, edges, DINO features, grounding boxes), moving toward a truly unified any-to-any multimodal model.

**Modality-Specific Expert Models**   Specialized models continue to advance performance in individual visual domains. For geometry, Depth Anything (Yang et al., 2024), Marigold (Ke et al., 2024), DepthFM (Gui et al., 2025), and Lotus (He et al., 2025) achieve strong depth estimation, while Omnidata (Kar et al., 2022) and GeoWizard (Fu et al., 2024) address surface normal prediction. For semantic understanding and spatial localization, Grounding DINO (Liu et al., 2024), Grounded-SAM (Ren et al., 2024), and GLaMM (Rasheed et al., 2024) provide high-quality grounding and segmentation, and self-supervised learners such as DINOv2 (Oquab et al., 2023) supply robust visual representations. Although these expert models achieve state-of-the-art results within their respective domains, they remain modality-specific and isolated. MODUS instead aims to unify their functional strengths within a single decoder-only framework capable of any-to-any multimodal generation.

## B. Additional Ablations

### B.1. Modality Mixing and Timestep Sampling

We discuss modality mixing and its connection to timestep sampling in Sec. 3.3 of the main paper. With logit-normal sampling, we observe substantial modality confusion, such as cases where a depth-generation prompt results in a normal map. In this section, we provide additional visualizations for a more detailed explanation. As shown in Figure 9 and Figure 10, we present inference results generated using different numbers of denoising timesteps. We compare two models: one trained with **uniform** timestep sampling and another trained with **logit-normal** sampling.

For the model trained with uniform sampling, we observe that the predictions remain stable across timesteps for both depth

and normal estimation. Even with only a few early timesteps, the model already produces a reasonable target modality, while later timesteps primarily refine structural details. This behavior arises because uniform sampling provides balanced coverage over the entire trajectory during training, ensuring that the model frequently sees the early-timestep regime where modality selection is determined. As a result, the model reliably commits to the correct target modality early in the trajectory and avoids modality confusion during inference.

In contrast, logit-normal sampling places most probability mass on middle (cleaner) timesteps and significantly undersamples the early, noisier region during training. Since the model receives far fewer updates at these early timesteps, it struggles to correctly infer the target modality when the signal is still ambiguous. As shown in our visualizations, this often leads to unstable or mixed-modality outputs at early timesteps, and the model may fail to recover even as the trajectory progresses. This explains why logit-normal sampling, while suitable for single-modality diffusion models, can amplify modality confusion in multi-modality flow prediction.

Overall, these results highlight the importance of sufficient early-timestep coverage for reliable target-modality selection. Uniform sampling provides this property, whereas logit-normal sampling may require additional regularization or curriculum strategies to mitigate modality confusion.

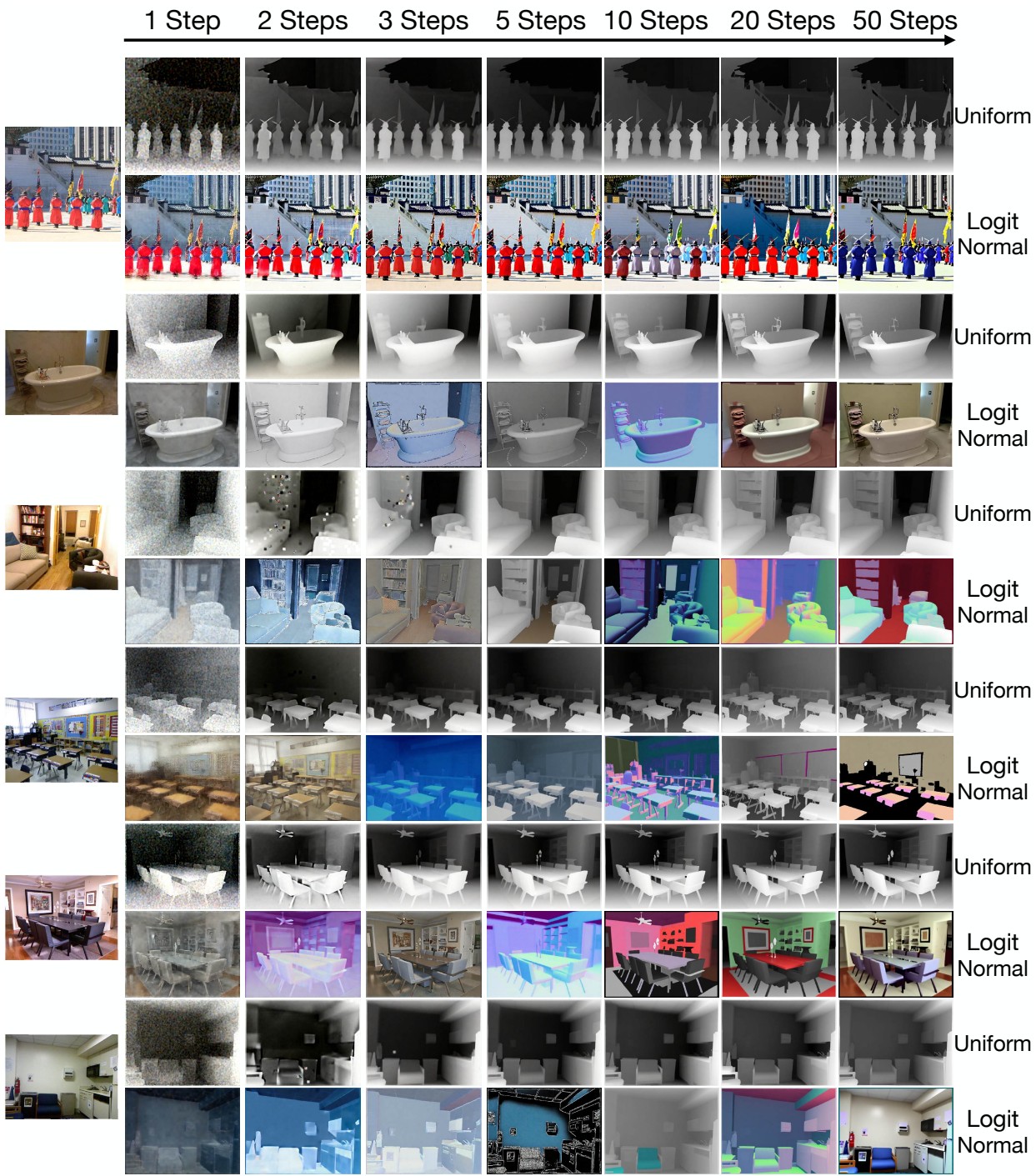

*Figure 9.* **Modality confusion and timestep sampling for depth prediction.** We compare inference trajectories across timesteps for models trained with uniform and logit-normal sampling. Uniform sampling covers early timesteps well, allowing the model to establish the correct depth modality early and produce stable predictions. Logit-normal sampling undersamples this regime, leading to early-stage modality mixing and failure to recover the correct output.

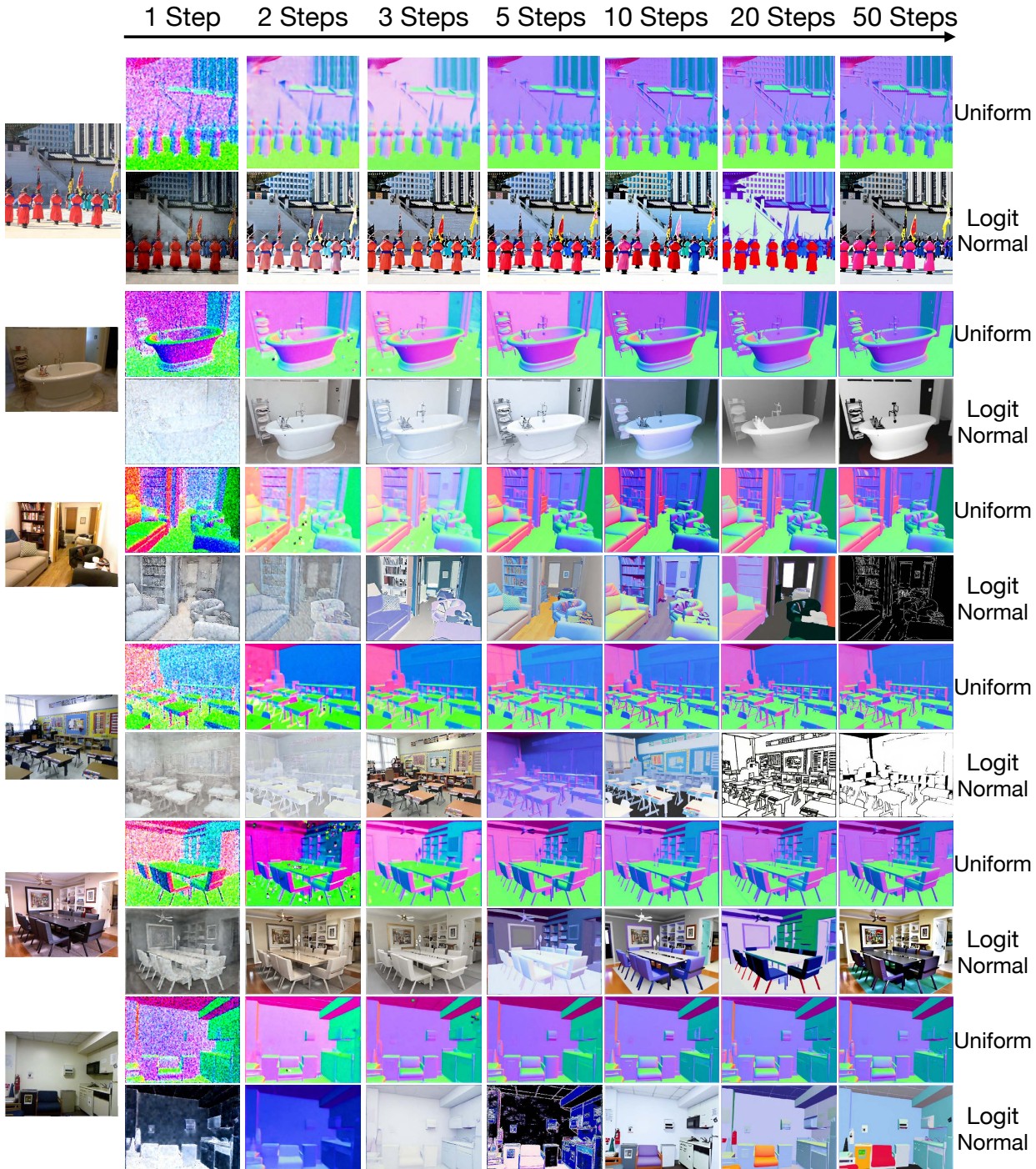

*Figure 10.* **Modality confusion and timestep sampling for surface normal prediction.** We compare inference trajectories from models trained with uniform and logit-normal sampling. With uniform sampling, the model consistently locks onto the surface normal modality in the early timesteps and refines details thereafter. In contrast, logit-normal sampling provides little supervision in this early region, causing the model to drift across modalities and often fail to produce a clean normal map.

## B.2. Training Stages

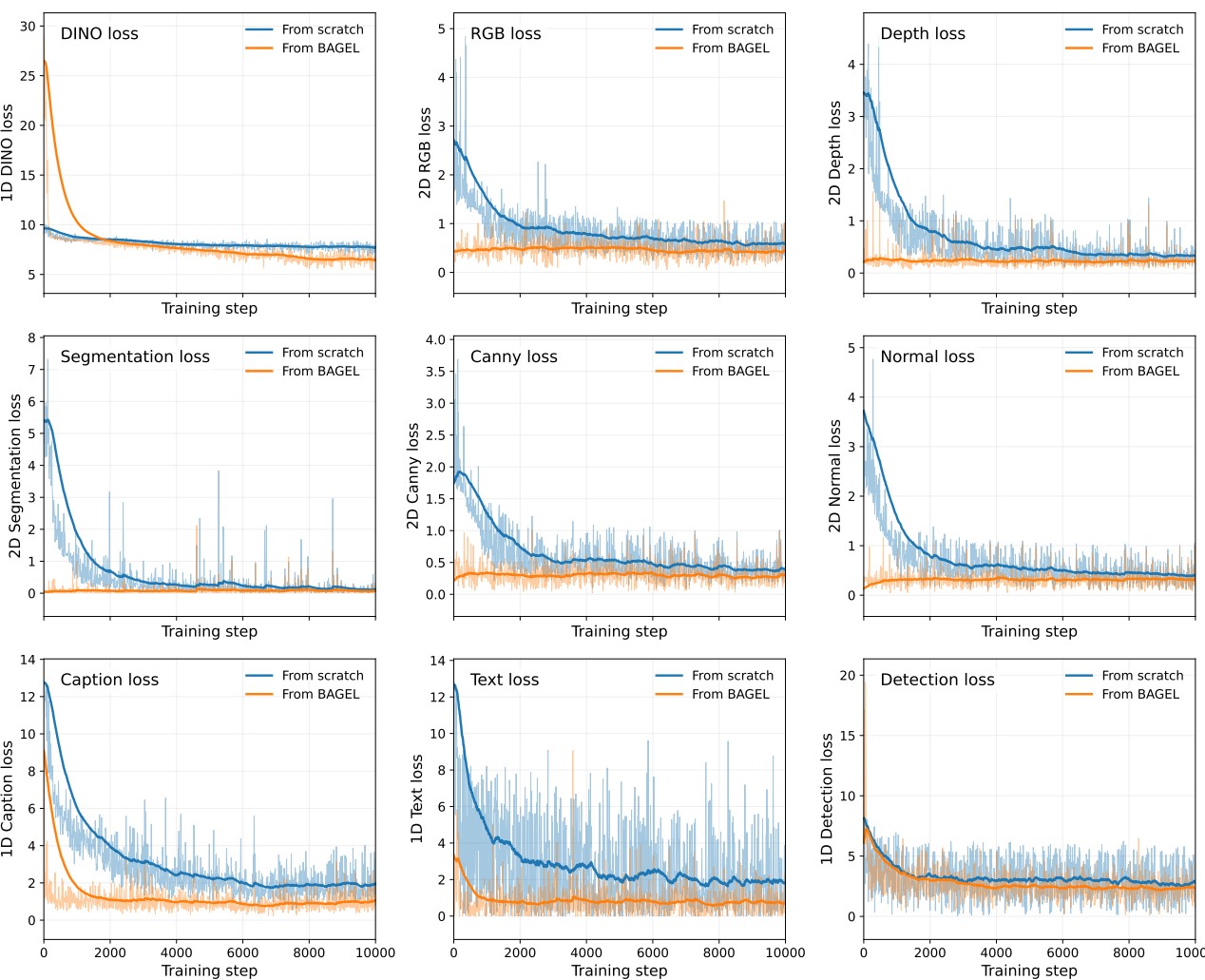

*Figure 11.* **Training losses across 1D and 2D modalities.** Loss curves for representative modalities during MODUS training. All modalities converge stably under the unified training regime, and initializing from BAGEL accelerates convergence across the board.

As shown in Table 7, we train MODUS using a three-stage curriculum. In the first stage, we introduce only the new 1D modalities so the model can learn modalities without strong priors, such as grounding and DINOv2 feature tokens, over a longer period of iterations. In the second stage, we incorporate the 2D modalities and allow the model to jointly learn the new 1D modalities together with the 2D modalities that come with stronger priors, enabling more efficient training. In the final stage, we enable multi-conditioned generation by mixing arbitrary combinations of modalities, allowing the model to perform any input–output transformation in a unified way. This progressive curriculum stabilizes optimization and helps the model adapt to the full set of modalities step by step. From Table 7, we observe that the model can be extended to new modalities efficiently while preserving performance on existing ones, and it can even support multi-conditioned chained generation.

## B.3. Training from Scratch

As shown in Figure 11 and Table 7, we also train MODUS from scratch for both 1D and 2D experts to assess the role of unified MLLM initialization. Training from scratch leads to noticeably slower convergence, especially for text–image related modalities, and results in lower performance on most tasks.

These results indicate that unified MLLM initialization provides useful multimodal priors that improve optimization stability

*Table 7.* Ablation on training stages. We adopt a three-stage training scheme that gradually incorporates 1D and 2D modalities and enables multi-conditioned generation in the final stage.

| | Training Tokens | RGB→Text VQA MMMU↑ | Text→RGB T2I GenEval↑ | RGB→Depth Depth Est. NYUv2↓ | RGB→Normal Normal Est. NYUv2↓ | RGB→Det Grounding RefCOCO$_{val}$↑ | RGB→Seg Inst. Segmentation COCO↑ | RGB→DINO Retrieval ImageNet↑ | Chained Generation |
|---|---|---|---|---|---|---|---|---|---|
| Bagel | ∼ 5T | 53.2 | 0.86 | ✗ | ✗ | ✗ | ✗ | ✗ | ✗ |
| **MODUS-Stage1** | 30B | 51.4 | 0.81 | ✗ | ✗ | 56.5 | ✗ | 78.8 / 92.8 | ✗ |
| **MODUS-Stage2** | 20B | 51.1 | 0.81 | 6.5 | 19.92 | 54.5 | 24.9 | 77.9 / 92.5 | ✗ |
| **MODUS-Stage3** | 15B | 50.3 | 0.81 | 6.6 | 20.02 | 53.8 | 25.0 | 77.2 / 92.2 | ✓ |
| MODUS-Scratch | 50B | 24.2 | N/A | 18.9 | 40.21 | 9.7 | 1.7 | 67.6 / 88.4 | ✗ |

*Table 8.* **Decoupled vs Unified I/O ablation.** Both variants share BAGEL-7B initialization, SPECTRUM-25M training data, and uniform timestep sampling. The decoupled variant replaces the unified sequence representation with small per-modality prediction heads. The unified-sequence formulation avoids fragmenting the representation across per-modality heads, with the largest gain on surface normal estimation. ↑ Higher is better. ↓ Lower is better.

| Variant | MMMU ↑ | DIODE ↓ | NYUv2 Normal ↓ |
|---|---|---|---|
| BAGEL + Specific Heads (decoupled) | 50.4 | 0.322 | 50.75 |
| **MODUS (unified sequence)** | **51.1** | **0.285** | **19.92** |

and overall performance in the any-to-any setting. Among all tasks, DINO feature prediction shows the smallest difference between the two training regimes, which is consistent with its lack of semantic priors in both settings.

### B.4. Decoupled vs Unified I/O

Many recent unified-modality models adopt modality-specific output heads to interface different modalities. To isolate the effect of MODUS's unified-sequence design from other factors, we train a controlled baseline that replaces the unified representation with small per-modality prediction heads (a 2D projector and head, a grounding regression head, and a DINO head), while keeping the BAGEL-7B initialization, SPECTRUM-25M training data, and uniform timestep sampling identical.

As shown in Table 8, the decoupled variant remains close to MODUS on MMMU (50.4 vs 51.1) but lags significantly on geometric tasks: 0.322 vs 0.285 on DIODE depth and 50.75 vs 19.92 on NYUv2 normal estimation. The large gap on normal estimation suggests that per-head decoupling fragments the representation space and weakens cross-modal transfer, whereas the unified sequence preserves cross-modal alignment without requiring per-head output design.

### B.5. Generalization to a Different Base Model

To assess whether the MODUS training recipe generalizes beyond BAGEL, we apply the same pipeline (unified-sequence tokenization, uniform timestep sampling, staged training, SPECTRUM-25M data) to Janus-Flow-1.3B (Ma et al., 2025b) as a base model. As shown in Table 9, this extension preserves most of Janus-Flow's MMMU performance (27.1 vs 29.3) while adding depth and surface normal estimation capabilities (0.304 DIODE, 36.53 NYUv2 normal). Janus-Flow originally requires 19.2k GH200 hours of pretraining; extending it with the MODUS pipeline takes an additional 1.5k GH200 hours, roughly 13× less than the original pretraining. This suggests that the unified-sequence and timestep-sampling choices transfer across base architectures rather than being specific to BAGEL.

## C. More Evaluations

### C.1. Depth Estimation

We present a detailed zero-shot depth estimation evaluation for MODUS. As shown in Table 10, we report results on three standard benchmarks, NYUv2, ScanNet, and DIODE, covering both indoor and outdoor scenes. Despite being a unified decoder-only model trained jointly with many other modalities, MODUS delivers competitive depth performance across all datasets, on par with multitask encoder–decoder and diffusion-based baselines. Its performance is also comparable to specialized single-task depth estimators, suggesting that unified multimodal training does not compromise depth estimation quality.

*Table 9.* **Generalization of the MODUS pipeline to a different base model.** Applying the MODUS training recipe to Janus-Flow-1.3B extends it with any-to-any capabilities while preserving most of its original MMMU performance, at approximately 13× less compute than the original Janus-Flow training. "✗" denotes capabilities not supported by the base model. ↓ Lower is better; ↑ Higher is better.

| Model | Scale | GH200 hrs | MMMU ↑ | DIODE ↓ | NYUv2 Normal ↓ |
|---|---|---|---|---|---|
| Janus-Flow (Base) | 1.3B | 19.2k | 29.3 | ✗ | ✗ |
| Janus-Flow + MODUS | 1.3B | 1.5k | 27.1 | 0.304 | 36.53 |
| 4M | 2.8B | 16.9k | ✗ | 0.331 | 37.28 |

*Table 10.* Zero-shot depth estimation comparison on NYUv2, ScanNet and DIODE benchmarks.

| Group | Method | NYUv2 | | ScanNet | | DIODE | |
|---|---|---|---|---|---|---|---|
| | | AbsRel↓ | $\delta_1$ ↑ | AbsRel↓ | $\delta_1$ ↑ | AbsRel↓ | $\delta_1$ ↑ |
| Single-task | MiDaS | 0.111 | 0.885 | 0.121 | 0.846 | 0.332 | 0.715 |
| | Omnidata | 0.074 | 0.945 | 0.075 | 0.936 | 0.339 | 0.742 |
| | DPT-large | 0.098 | 0.903 | 0.082 | 0.934 | 0.182 | 0.758 |
| | DepthAnything | 0.043 | 0.980 | 0.043 | 0.981 | 0.261 | 0.759 |
| | DepthAnything v2 | 0.043 | 0.979 | 0.042 | 0.979 | 0.249 | 0.752 |
| Encoder–Decoder | Unified-IO | 0.059 | 0.970 | 0.063 | 0.965 | 0.369 | 0.708 |
| | 4M-XL | 0.068 | 0.951 | 0.065 | 0.955 | 0.331 | 0.734 |
| Diffusion | OneDiffusion | 0.087 | 0.924 | 0.094 | 0.906 | 0.399 | 0.661 |
| | DiCeption | 0.061 | 0.960 | 0.072 | 0.944 | 0.289 | 0.722 |
| Decoder-only | (Ours) | 0.065 | 0.958 | 0.067 | 0.957 | 0.285 | 0.718 |

## C.2. Referring Object Grounding

We present zero-shot referring expression comprehension results in Table 11 on RefCOCO, RefCOCO+, and RefCOCOg. While prior specialist grounding models and existing decoder-only systems typically focus on a narrow set of tasks, MODUS achieves comparable or superior performance across all evaluation splits. Notably, Kosmos-2 is also a decoder-only model capable of both VQA and grounding, yet MODUS supports a far broader any-to-any generation setting while maintaining strong zero-shot grounding performance.

## C.3. Image–Text Capability Preservation

Extending a strong base model like BAGEL to many additional modalities risks degrading its original image–text capabilities. To mitigate this, MODUS mixes LLaVA-OneVision into SPECTRUM-25M training. Table 12 reports a broader image–text evaluation beyond MMMU.

Training on SPECTRUM-25M alone (without the LLaVA-OV mix) causes large drops across the board, *e.g.*, POPE 87.23 → 73.67, VizWiz 59.41 → 8.34, and MME-S 2377 → 1561, confirming the importance of the data mixture. The full MODUS setup recovers most of these losses: VizWiz stays within 1.2 points of BAGEL (58.21 vs 59.41), POPE stays within 0.04 of baseline, and MME scores remain within 3% of BAGEL. Document and high-resolution benchmarks (DocVQA, ChartQA) show some residual drop, consistent with the LLaVA-OV mix not including BAGEL's original full VQA training mixture.

## C.4. Multi-Condition and Direct Any-to-Any Generation

Most standardized benchmarks are RGB- or text-centric. To probe MODUS's behavior beyond these settings, we evaluate three families of mappings on DIODE depth and NYUv2 normal estimation: standard (RGB→X), multi-condition (multiple modalities → X), and direct any-to-any between non-standard pairs (Table 13).

Multi-conditioning improves over the standard RGB→Normal baseline (RGB+Depth→Normal: 19.58; Edge+Depth→Normal: 19.72, vs 20.02 standard). Notably, Edge+Depth→Normal approaches the RGB-based baseline without using RGB as input, suggesting that MODUS combines complementary signals across modalities rather than relying on RGB as a privileged input. Direct any-to-any transfers between sparser modality pairs are naturally weaker

*Table 11.* Zero-shot comparison on referring expression comprehension benchmarks (RefCOCO, RefCOCO+, RefCOCOg).

| Group | Method | RefCOCO | | | RefCOCO+ | | | RefCOCOg | |
|---|---|---|---|---|---|---|---|---|---|
| | | val | testA | testB | val | testA | testB | val | test |
| Single-task | GLIP-T | 50.42 | 54.30 | 43.83 | 49.50 | 52.78 | 44.59 | 66.09 | 66.89 |
| | Grounding-DINO-T | 50.41 | 57.24 | 43.21 | 51.40 | 57.59 | 45.81 | 67.46 | 67.13 |
| Decoder-only | Kosmos-2 | 52.32 | 57.42 | 47.26 | 45.48 | 50.73 | 42.24 | 60.57 | 61.65 |
| | (Ours) | 54.50 | 58.60 | 50.91 | 49.75 | 54.94 | 44.63 | 56.50 | 56.21 |

*Table 12.* **Preservation of image–text capabilities.** "+ LLaVA-OV" and "+ SPECTRUM-25M only" are ablations that add the named data to BAGEL individually; MODUS combines BAGEL initialization with LLaVA-OneVision mixing and SPECTRUM-25M training. Training on SPECTRUM-25M alone causes large drops, while the full MODUS setup recovers most of BAGEL's original VQA and captioning capabilities. ↑ Higher is better.

| Benchmark | BAGEL | + LLaVA-OV | + SPECTRUM-25M only | MODUS |
|---|---|---|---|---|
| MMMU ↑ | 53.20 | 52.30 | 47.90 | 51.10 |
| MME-P ↑ | 1687.00 | 1663.56 | 1157.99 | 1637.69 |
| MME-S ↑ | 2377.17 | 2316.77 | 1560.85 | 2313.40 |
| POPE ↑ | 87.23 | 87.77 | 73.67 | 87.27 |
| VizWiz ↑ | 59.41 | 57.77 | 8.34 | 58.21 |
| DocVQA ↑ | 94.05 | 93.37 | 90.02 | 90.11 |
| ChartQA ↑ | 86.72 | 85.47 | 81.09 | 82.45 |

(Edge→Depth 0.302 vs RGB→Depth 0.285; Depth→Normal 29.51 vs RGB→Normal 20.02), but achievable within the same pretrained model and without task-specific pathways, which the underlying teacher models cannot do.

## C.5. Contamination Check

To ensure that SPECTRUM-25M's pseudo-label supervision does not produce evaluation contamination, we checked the overlap between teacher training data and our evaluation benchmarks. **Depth:** DepthAnything V2 (Yang et al., 2024) and Marigold (Ke et al., 2025) are not trained on DIODE or NYUv2. **Grounding:** our GLaMM-based pipeline (Rasheed et al., 2024) uses a subset of SA-1B and remains zero-shot on COCO, which is not used during SPECTRUM-25M construction. **Features:** DINOv2 (Oquab et al., 2023) is not trained on the ImageNet splits used for our retrieval evaluation. **Source images:** SPECTRUM-25M is built on BLIP-3o sources (SA-1B, JourneyDB, CC12M), none of which directly overlap with our evaluation benchmarks. Together, these checks indicate that MODUS's reported numbers reflect cross-modal modeling rather than memorization of teacher behavior.

## C.6. GenEval Per-Category Breakdown and Self-Verification

To analyze where MODUS gains and loses relative to BAGEL on text-to-image generation, we report a per-category GenEval breakdown in Table 14. MODUS matches or slightly improves on simpler categories (single-object, colors) but loses on harder compositional ones (counting, position, color+position). A simple Best-of-4 test-time search using MODUS's own grounding and VQA capabilities to score candidate generations recovers most of this gap, lifting overall GenEval from 0.81 to 0.84. This is a direct consequence of the unified design: MODUS can act as its own verifier within the same model, without external scoring components.

**Connection to test-time scaling for image generation.** Best-of-$N$ verification is part of a broader emerging direction of test-time scaling for image generation, where additional inference compute substitutes for additional training. Prior work explores this through repeated sampling with verifiers (Brown et al., 2024; Snell et al., 2024), inference-time search tailored to diffusion models (Ma et al., 2025a; Singhal et al., 2025; Zhang et al., 2025), and tokenizer-aware ordered-token search for autoregressive image generators (Gao et al., 2026; Chen et al., 2025c). Our self-verification result shows that a unified any-to-any model is naturally suited to this paradigm: the same decoder that generates an image also produces the auxiliary modalities (grounding, VQA) used to score it, removing the need for any external verifier.

*Table 13.* **Multi-condition and any-to-any results beyond RGB/text-centric settings.** Standard, multi-condition, and direct any-to-any mappings on DIODE depth and NYUv2 normal estimation. Multi-conditioning improves over standard RGB→Normal; Edge+Depth→Normal is nearly as strong as the RGB-based mapping despite not using RGB as input. Direct any-to-any transfers between sparser modality pairs are naturally weaker but achievable within the same pretrained model. ↓ Lower is better.

| Mapping | Source → Target | Dataset | MODUS ↓ |
|---|---|---|---|
| Standard | RGB → Depth | DIODE | 0.285 |
|  | RGB → Normal | NYUv2 | 20.02 |
| Multi-cond. | RGB + Depth → Normal | NYUv2 | 19.58 |
|  | Edge + Depth → Normal | NYUv2 | 19.72 |
| Any-to-any | Edge → Depth | DIODE | 0.302 |
|  | Depth → Normal | NYUv2 | 29.51 |

*Table 14.* **GenEval breakdown by category and effect of self-verification.** Compared with BAGEL, MODUS matches or slightly improves on simple categories (single-object, colors) but loses on harder compositional ones (counting, position, color+position). A simple Best-of-4 self-verification (re-ranking with MODUS's own grounding and VQA) recovers most of the gap. ↑ Higher is better.

| Model | Overall | Single Obj. | Two Obj. | Count | Colors | Position | Color+Pos. |
|---|---|---|---|---|---|---|---|
| BAGEL | 0.86 | 0.98 | 0.94 | 0.74 | 0.94 | 0.84 | 0.74 |
| MODUS | 0.81 | 0.99 | 0.90 | 0.68 | 0.92 | 0.77 | 0.62 |
| MODUS (Best-of-4, self-verify) | 0.84 | 1.00 | 0.90 | 0.75 | 0.94 | 0.80 | 0.65 |

## C.7. Inference Efficiency: Direct vs Chained

MODUS's 2D Expert uses parallelized flow matching rather than token-by-token autoregression, so chained generation does not introduce a prohibitive latency cost. Table 15 benchmarks RGB→Normal and RGB→Canny→Normal at $512 \times 512$ resolution on a single GH200 GPU over the 654-image NYUv2 split. Two observations: (i) direct generation is already accurate at 5–10 steps; (ii) chained generation reaches accuracy comparable to long direct chains at substantially lower latency, *e.g.*, 10-step chained (4.28 s/img, 20.08 normal error) matches 50-step direct (9.07 s/img, 20.02 normal error) at less than half the latency. This trade-off makes chained generation practical even when intermediate modalities are added.

*Table 15.* **Inference efficiency: direct vs chained generation.** Latency and NYUv2 normal estimation accuracy at $512 \times 512$ resolution on a single GH200 GPU over 654 NYUv2 images. Chained generation does not introduce a prohibitive autoregressive bottleneck: 10-step chained (4.28 s/img) reaches the accuracy of 50-step direct (9.07 s/img) at less than half the latency. ↓ Lower is better.

| Task | Steps | Sec/img | NYUv2 Normal ↓ |
|---|---|---|---|
| RGB → Normal | 2 | 0.59 | 34.06 |
| | 5 | 1.14 | 21.91 |
| | 10 | 2.05 | 20.30 |
| | 50 | 9.07 | 20.02 |
| RGB → Canny → Normal | 2 | 1.38 | 36.82 |
| | 5 | 2.57 | 22.21 |
| | 10 | 4.28 | 20.08 |
| | 50 | 20.05 | 19.87 |

# D. More Visualizations

## D.1. Chained Generation

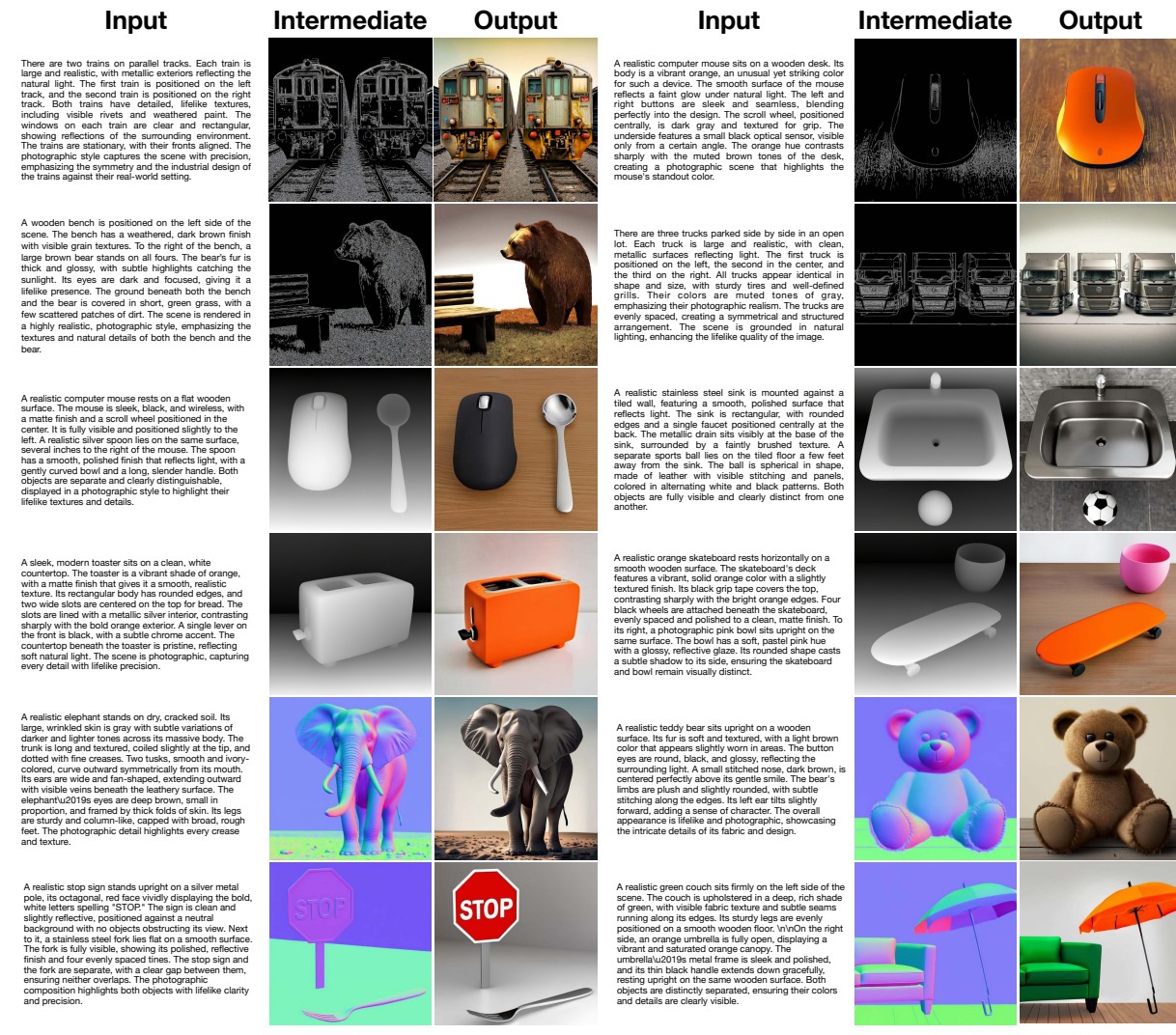

*Figure 12.* **Chained text-to-image generation.** Text prompts are transformed into intermediate 2D modalities, including Canny edges, depth maps, and surface normals, before producing the final RGB image. The examples demonstrate high visual quality and strong cross-modality consistency throughout the chained generation process.

In Section 4.3 of the main paper, we discussed chained generation, where MODUS produces intermediate modalities before generating the final target output. In Figure 12 and Figure 13, we include additional visualizations to further illustrate this capability. Across tasks such as text to image and image to normal, we observe that intermediate predictions, including Canny edges, depth maps, and surface normals, remain coherent and consistent with the source input. The final outputs follow these intermediate representations closely, showing that MODUS can maintain structural and semantic consistency throughout the chained generation process.

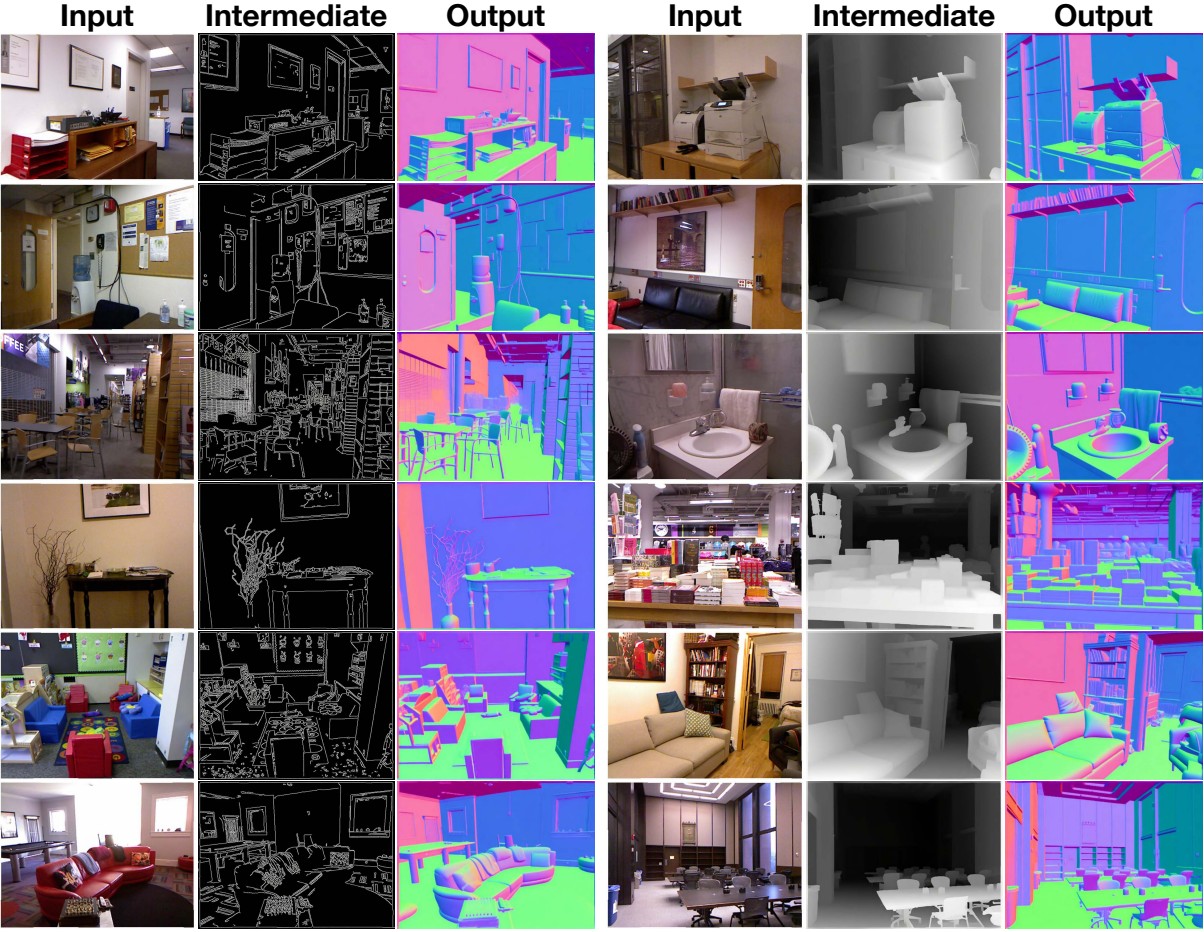

*Figure 13.* **Chained image-to-normal prediction.** The input image is first transformed into intermediate modalities, such as depth or Canny edges, and the resulting representations are then used to produce the final surface normal map. The examples illustrate consistent and coherent predictions across the chained modalities.

## D.2. Self Verification

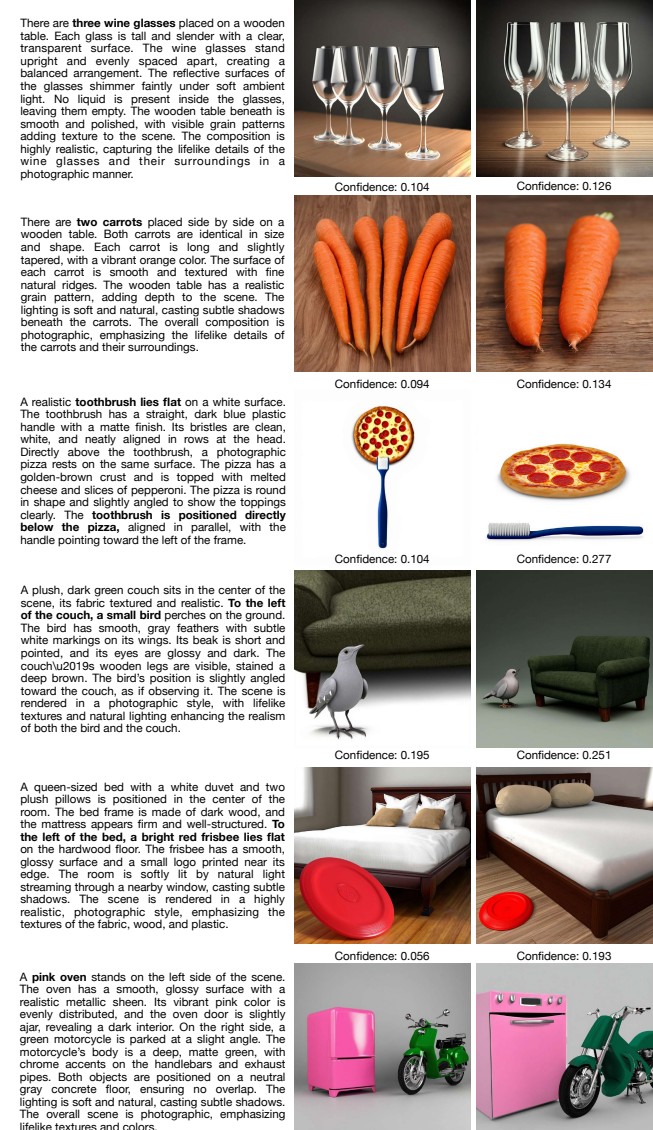

*Figure 14.* **Text-to-image generation with self-verification.** We apply the grounding capability of MODUS to evaluate the quality of its own text-to-image outputs and select the sample with the highest verification score. As shown, this simple test-time search, using a task already supported by MODUS, leads to improved image quality and better alignment with the input prompt.

In Sec. 4.4 of the main paper, we discussed how the referring object grounding modality in MODUS provides verification for text-to-image generation. Here, we present additional visualizations in Figure 14. For each text prompt, we generate several image candidates and simply read the grounding logits produced by MODUS when asked to localize the referred objects. We observe that these confidence values often correlate with whether the generated image includes the requested objects and with cues such as approximate count or location. Based on this observation, we choose the sample with the highest grounding confidence as the representative output. The visualizations illustrate that grounding confidences can provide a simple signal for examining object presence and prompt adherence in text-to-image generations, without introducing additional training or external scoring models.

### D.3. Visual Representation Composition

In Sec. 4.5 of the main paper, we reported an intriguing form of visual hallucination that emerges when MODUS generates 2D modalities using only high-level semantic features. In MODUS, each 2D modality is represented using two complementary

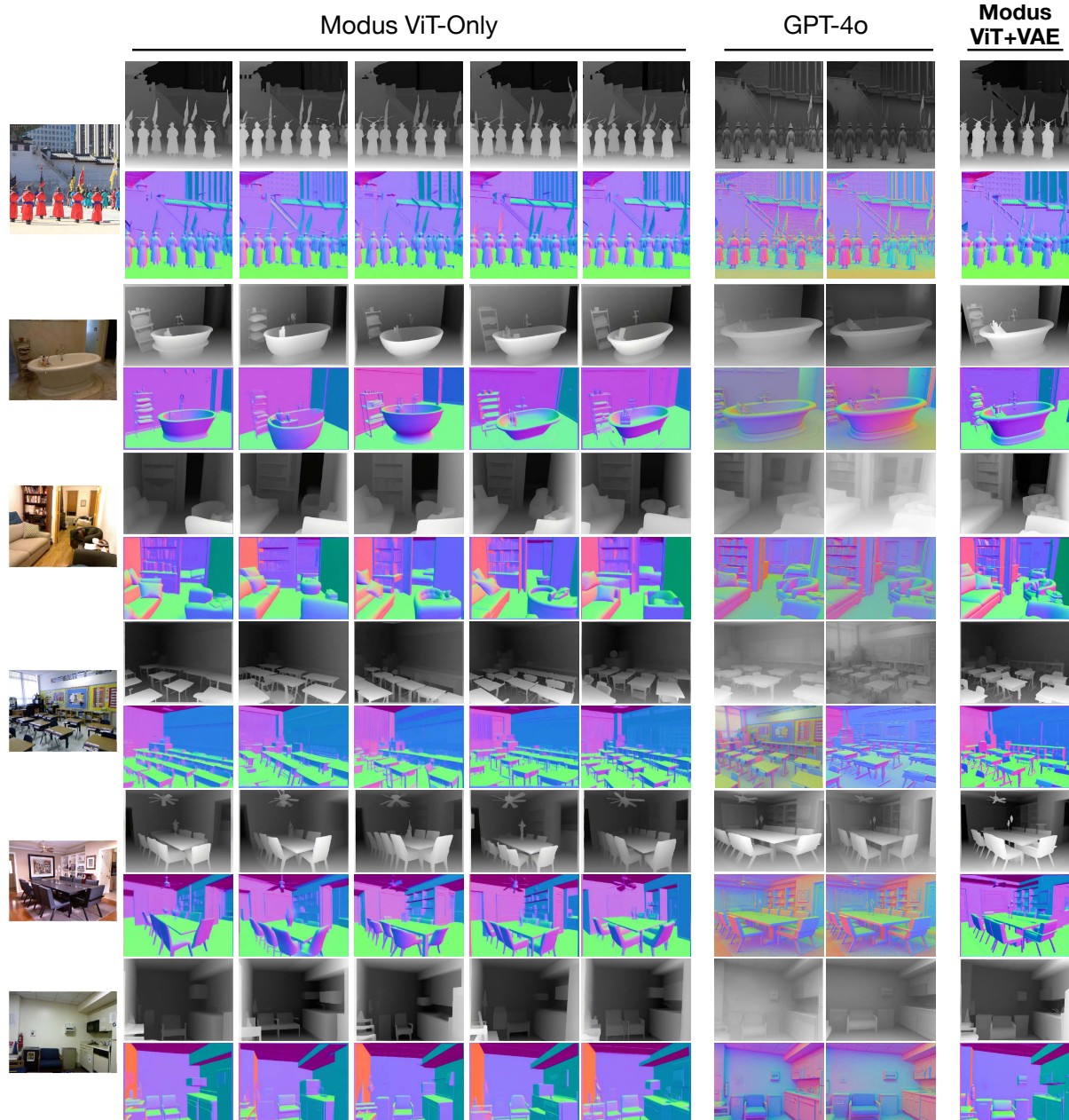

*Figure 15.* **Illustration of hallucination with 2D modality feature representations.** Each 2D modality in MODUS is represented using a ViT-based semantic feature and a VAE-based reconstruction feature. Conditioning the generation on the semantic feature can produce plausible but structurally incorrect depth or normal maps, showing changes in object shape, number, or fine details. GPT-4o exhibits similar behavior when asked to generate depth maps, including mild modality confusion. The final column shows accurate results from MODUS when both semantic and reconstruction features are used as conditions.

encodings: (1) a ViT-derived semantic feature for global understanding, and (2) a VAE-based reconstruction feature that preserves local geometry and fine details. To better illustrate the impact of these two branches, we provide additional qualitative results in Figure 15.

When we condition the generation on ViT features, the model is still able to produce plausible depth or normal maps that are semantically consistent with the input image. However, these outputs often deviate from the exact scene structure. Across multiple samples per image, five for depth and five for normals, we observe systematic distortions: object shapes may change, the number of instances may vary, and fine-grained geometric details may be altered or lost. These variations

indicate that high-level semantic features alone do not sufficiently constrain the spatial layout, allowing the model to generate semantically aligned but structurally inconsistent predictions.

Interestingly, this phenomenon echoes observations reported in recent evaluations of GPT-4o's image generation capabilities (Ramachandran et al., 2025). We collected GPT-4o outputs using prompts such as "Please generate the gray-scale depth map of the input image and keep the original resolution." Similar hallucinations occur: missing object parts, changes in inferred camera viewpoint, and in some cases mild modality confusion that may stem from limited training coverage on dense geometric modalities. These parallels indicate that GPT-4o likely relies on a comparable semantic reconstruction process when asked to generate structure-sensitive outputs. It also suggests that high-level–only conditioning, even in large proprietary systems, may drive a form of semantic understanding rather than precise geometric prediction.

In the final column of Figure 15, we show the corresponding results from MODUS when conditioned on both ViT semantic features and VAE low-level features. This combination reinstates geometric fidelity, producing depth and normal maps that accurately preserve scene structure, object boundaries, and local surface details. Overall, these comparisons highlight the importance of coupling high-level semantics with low-level geometric cues in unified multimodal generation, and they provide insight into why models relying solely on high-level conditioning may exhibit consistent visual hallucinations.

*Table 16.* **Training settings.** Training Configuration for used in three stages.

| Configuration | Stage 1 | Stage 2 | Stage 3 |
|---|---|---|---|
| Training length | 30B | 20B | 15B |
| Warmup length | | 1B | |
| Optimizer | | AdamW | |
| Opt. momentum | | $\beta_1, \beta_2 = 0.9, 0.95$ | |
| Base learning rate | | 2e-5 | |
| Sequence Length | 14336 | 14336 | 16384 |
| Weight decay | | 0 | |
| Gradient clipping | | 1.0 | |
| Learning rate schedule | | Constant | |
| Num. of condition modalities | 1 | 1 | 3 |
| Modalities | RGB, Text, Grounding, DINOv2 | All | All |
| ViT max resolution | | (224, 518) | |
| VAE resolution | | (512, 1024) | |
| Augmentation | | None | |
| Data type | | bfloat16 | |

# E. Implementation Details

## E.1. Dataset

We train MODUS on SPECTRUM-25M, a large-scale multimodal dataset constructed from the BLIP-3o (Chen et al., 2025a) corpus of 25M image-text pairs and extended to diverse modalities. To obtain supervision at this scale, especially for modalities where human annotations are scarce, we apply high-quality pseudo-labelers with an emphasis on both accuracy and efficiency. Specifically, we use Grounded-SAM (Ren et al., 2024) for automatic open vocabulary instance segmentation, DepthAnything V2 (Yang et al., 2024) for depth estimation, and Marigold (Ke et al., 2025) for surface normal prediction.

We also generate canny edge maps using the classical Canny operator and extract global visual feature maps using DINOv2 (Oquab et al., 2023). For grounding supervision, we use ground truth bounding boxes from the GLaMM (Rasheed et al., 2024) dataset, which overlaps with the BLIP-3o data sources and provides reliable annotations. In addition, we incorporate the LLaVA-OneVision (Li et al., 2024) dataset to improve VQA training for the text modality.

## E.2. Training

During training, we sample a sequence of modalities and feed them to the model. Each sequence contains several conditioning modalities and one target modality. The attention pattern between different modalities is kept causal. Within each modality, 1D modalities use causal attention, while 2D modalities use bidirectional attention for both ViT tokens and VAE tokens. We also apply sequence packing to maintain efficient and stable sequence lengths during training. Depending on the type of the target modality, we compute its loss using cross entropy for 1D tokens and mean squared error for 2D features. Detailed training configuration for different stages is shown in Table 16.

# F. Limitation Discussion

While MODUS is designed as a unified decoder-only model capable of any-to-any multimodal generation, several aspects remain open for future improvement.

First, MODUS focuses on transformations across different modalities, but it does not yet support *iterative editing within the same modality* (*e.g.*, depth → refined depth, segmentation → edited segmentation). This capability requires curated datasets

that contain sequential edits or multi-turn modality refinements, which are scarce in existing multimodal corpora. We believe that a unified any-to-any model is a useful step toward enabling the creation of such synthetic multimodal editing datasets, which in turn can feed back into future training to improve the model's ability to handle iterative within-modality edits.

Second, MODUS is trained purely in a pre-training setting and therefore does not explicitly target reasoning tasks. While the model already demonstrates generalization across modalities, supporting tasks that require complex reasoning would benefit from incorporating dedicated post-training objectives and datasets, such as multimodal chain-of-thought (*e.g.*, DeepEyes (Zheng et al., 2025)). Extending MODUS with a lightweight post-training stage is a natural next step and would broaden its utility without altering the unified architecture.

Overall, these limitations highlight opportunities for improvement in dataset coverage and task formulation. Addressing them will require further investigation, but the results suggest that there remains strong potential to support an even broader range of tasks within a single unified decoder framework.

# G. SPECTRUM-25M Examples

We present example visualizations of pseudo labels from SPECTRUM-25M in Figure 16. We will release the full dataset.

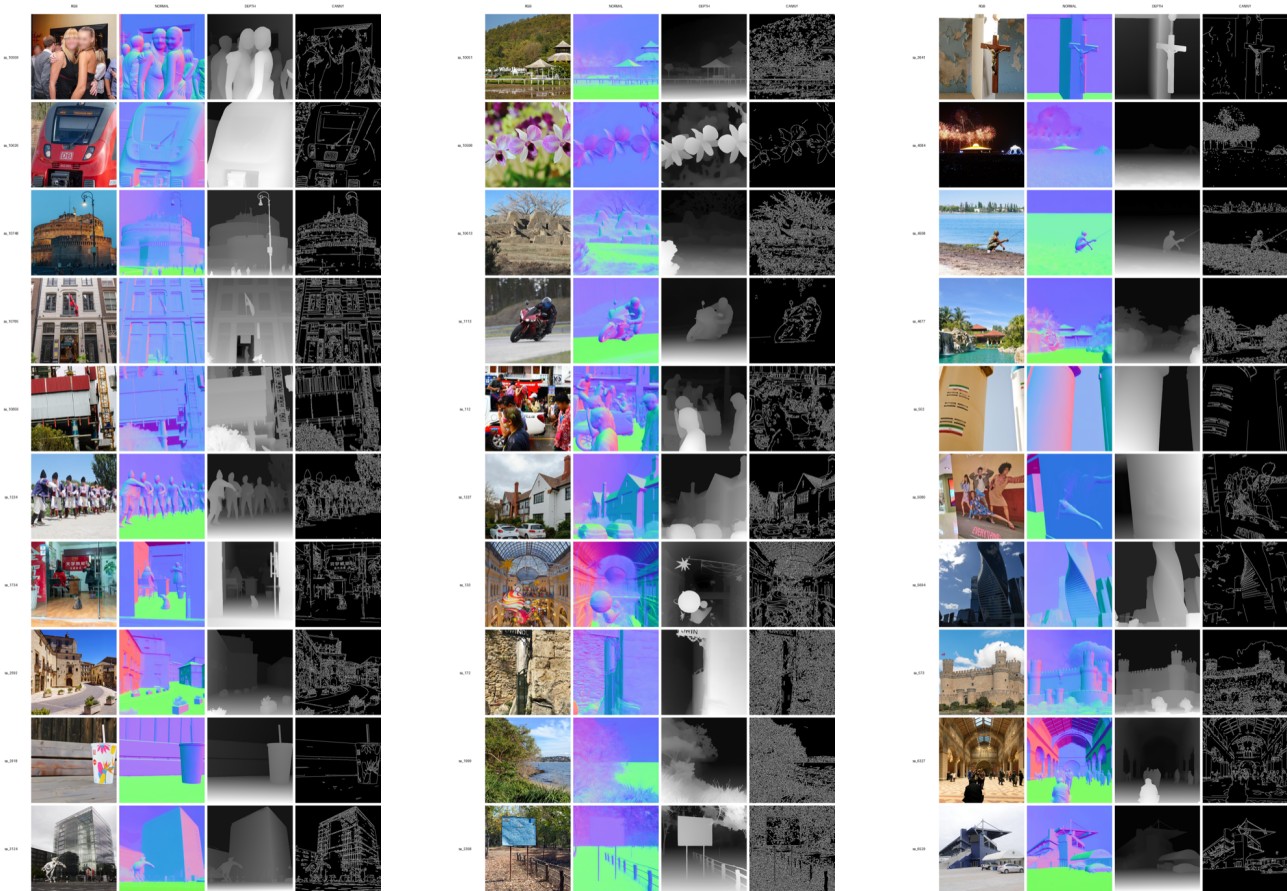

*Figure 16.* **Pseudo label visualization for surface normals, depth, and Canny edges.**

