# Modus: A Decoder-only Any-to-Any Model of Diverse Modalities

# Multimodal Pre-training

## Decoder-only Transformer

- Simplicity in training

- Clear scaling laws

- Strong zero-shot performance

- Efficient inference

- **Mainly on Image-Text modalities**

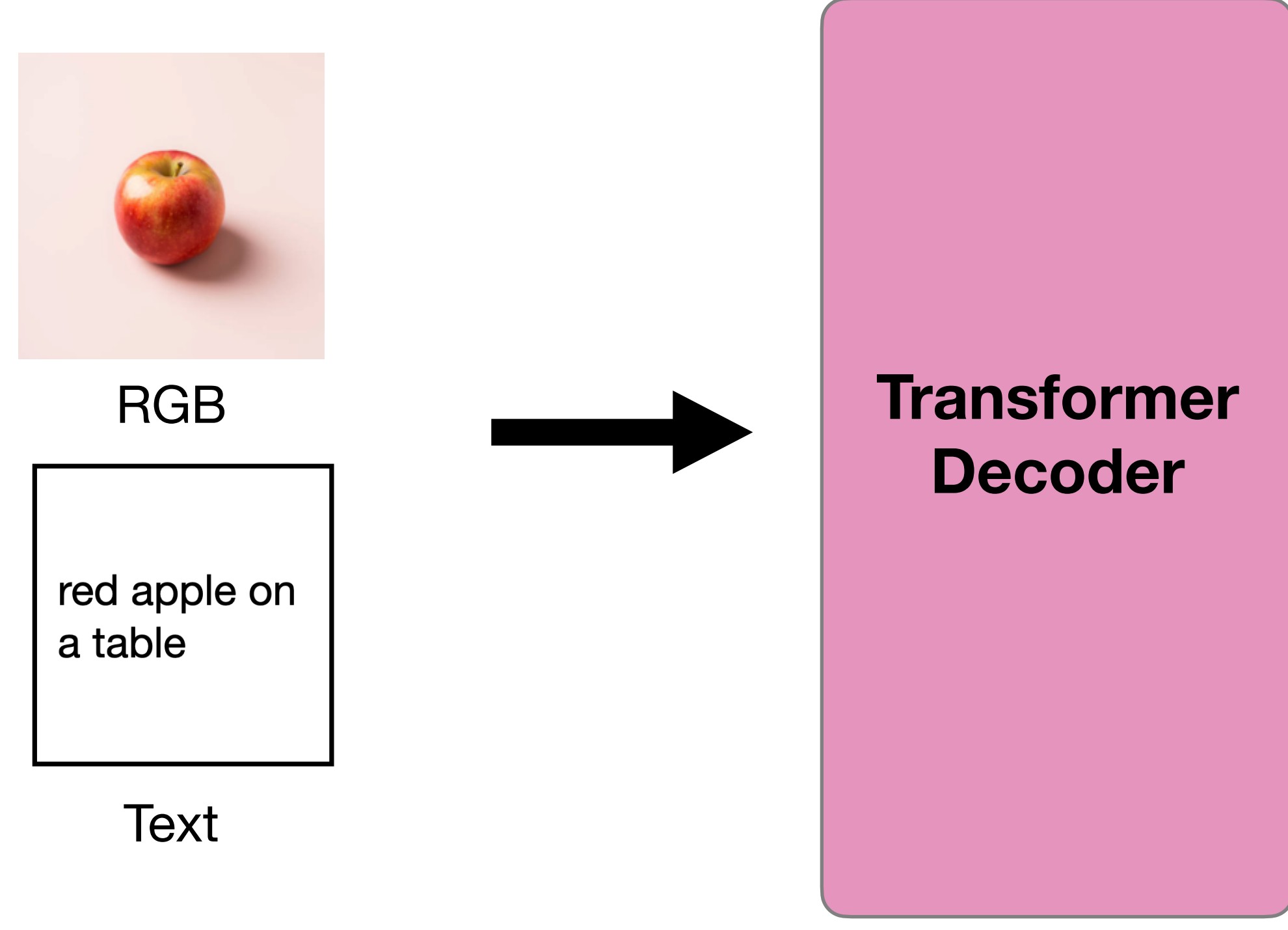

RGB

red apple on a table

Text

**Transformer Decoder**

# Multimodal Pre-training

**Beyond Image-Text
Our world is multimodal**

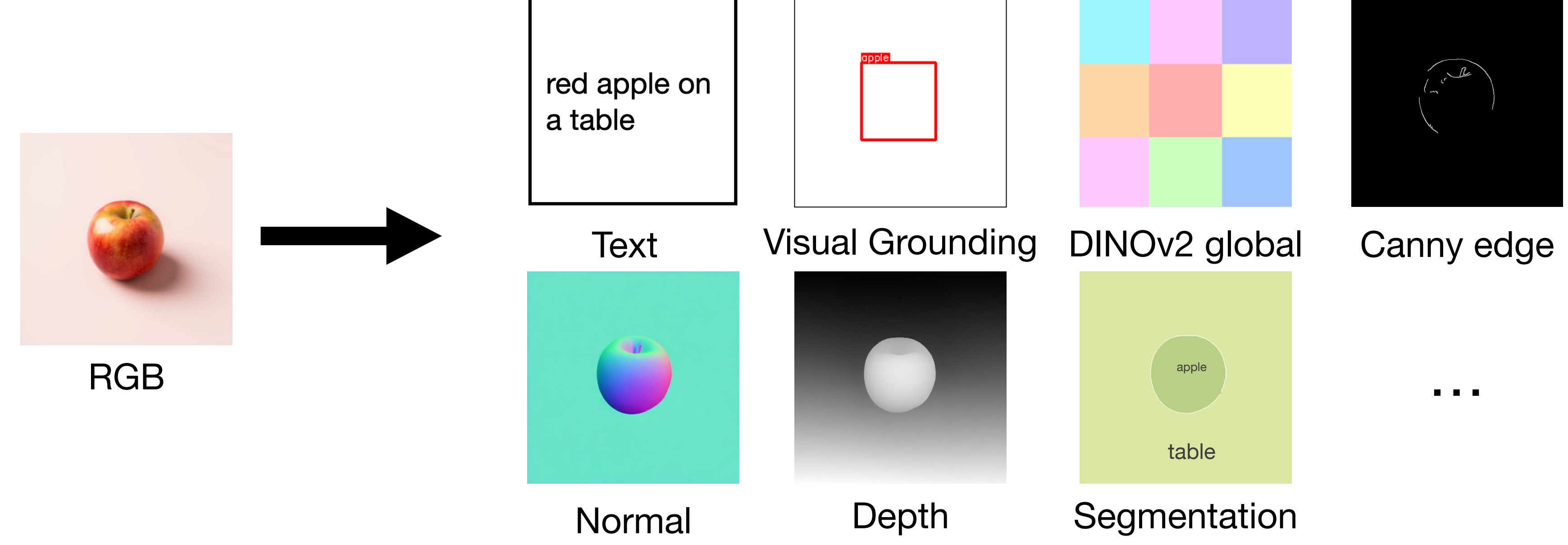

# Goal

## A Decoder-only Transformer for any-to-any modeling

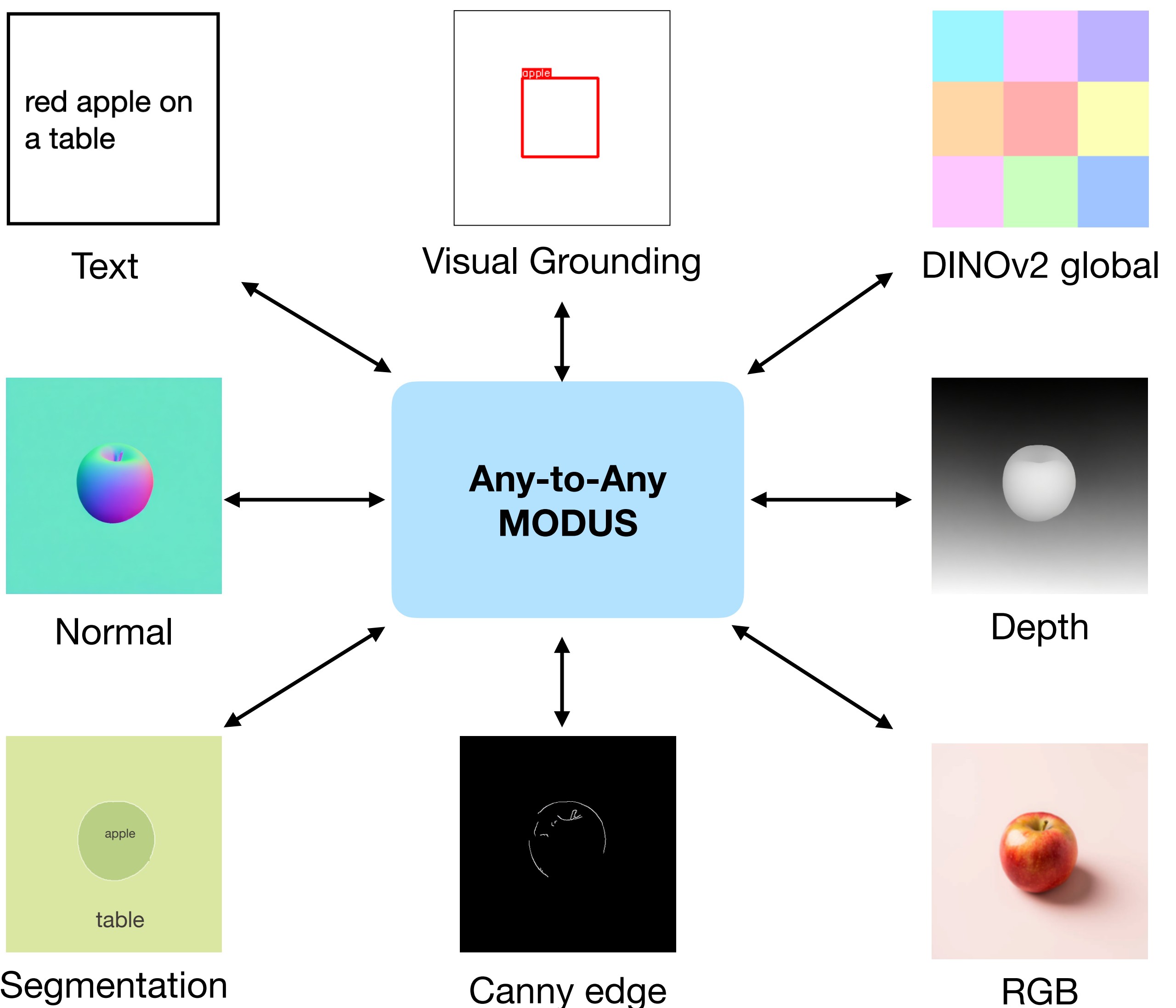

Text

Visual Grounding

DINOv2 global

Normal

Any-to-Any
MODUS

Depth

Segmentation

Canny edge

RGB

# Model

## Modus: Training with condition and target modalities

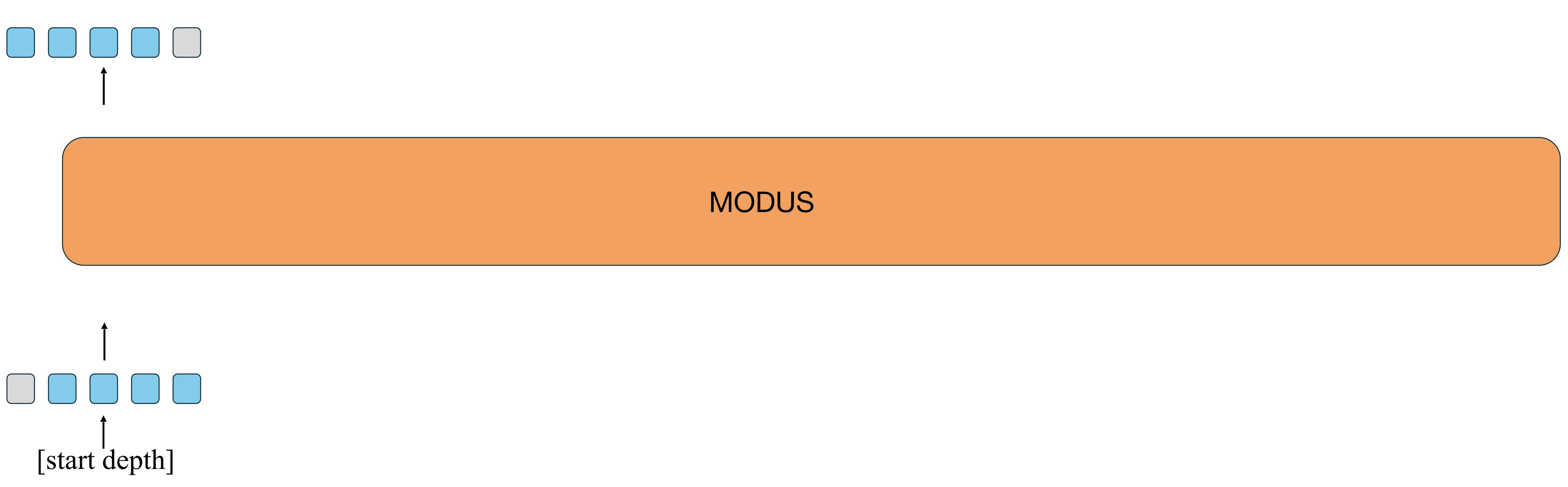

[start depth]

# Model

## Modus: Training with condition and target modalities

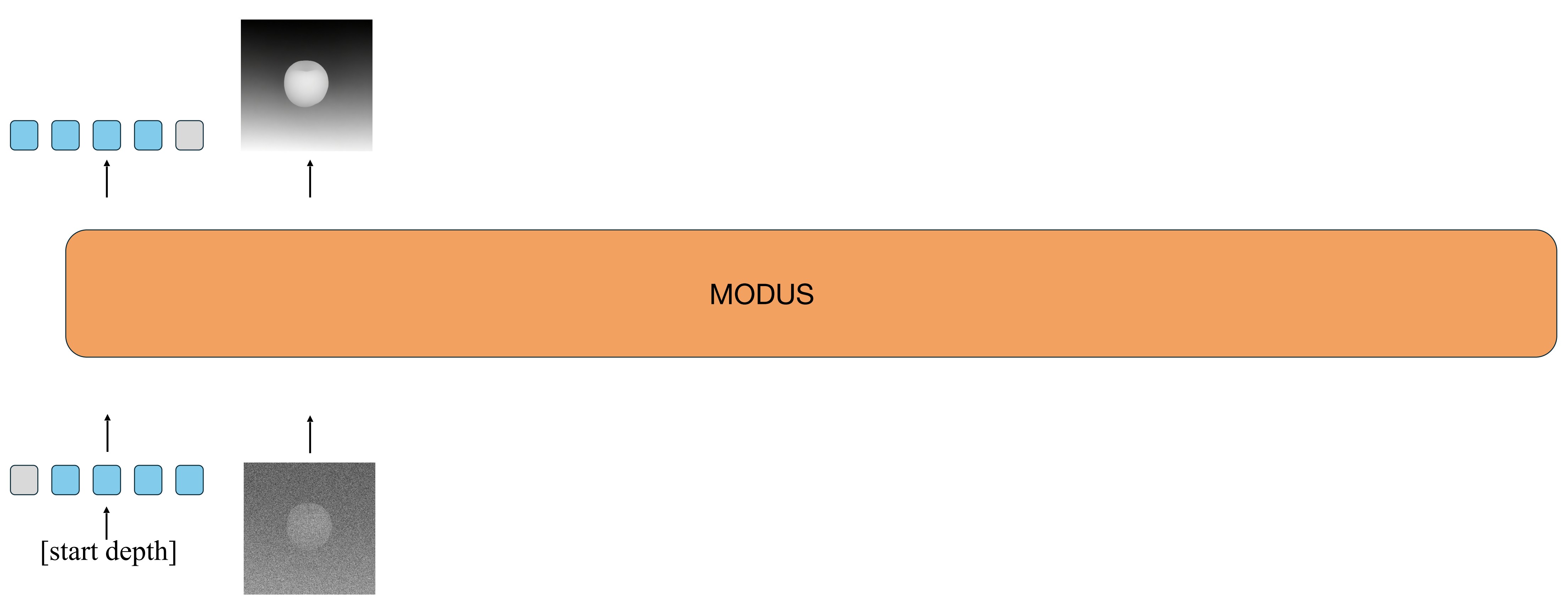

[start depth]

MODUS

# Model

## Modus: Training with condition and target modalities

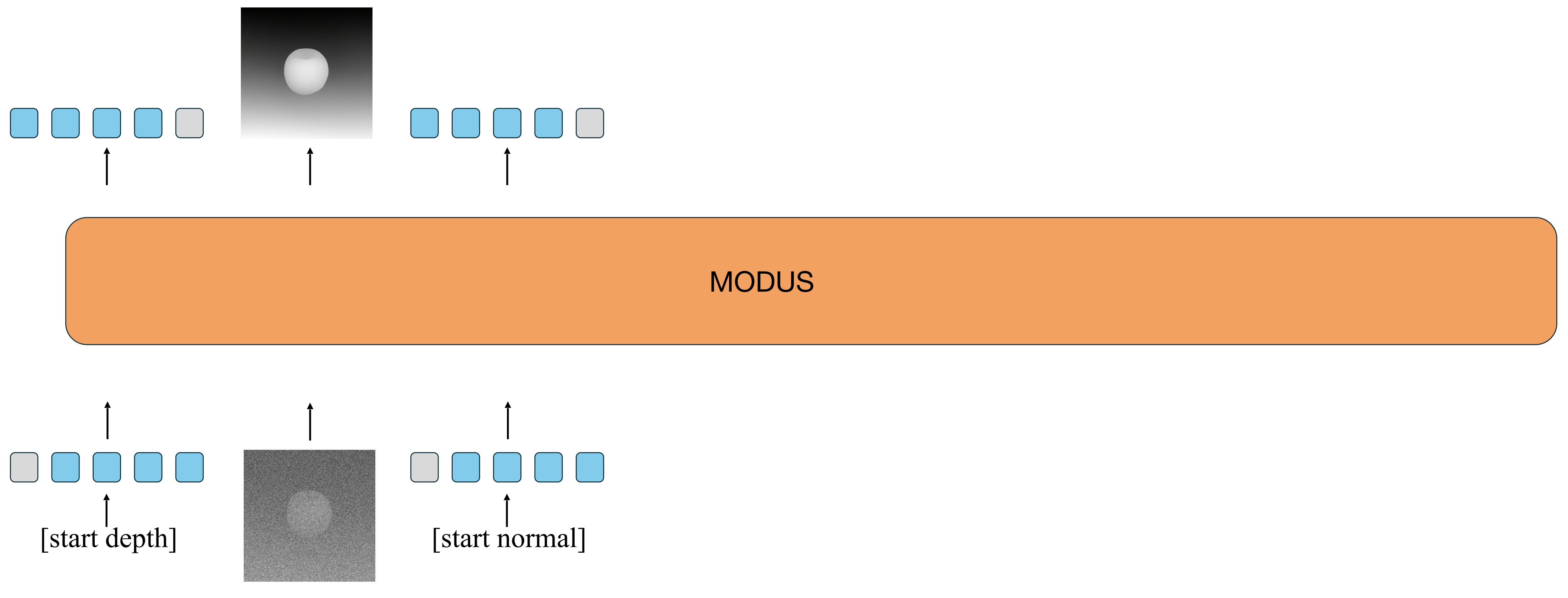

[start depth]          [start normal]

# Model

## Modus: Training with condition and target modalities

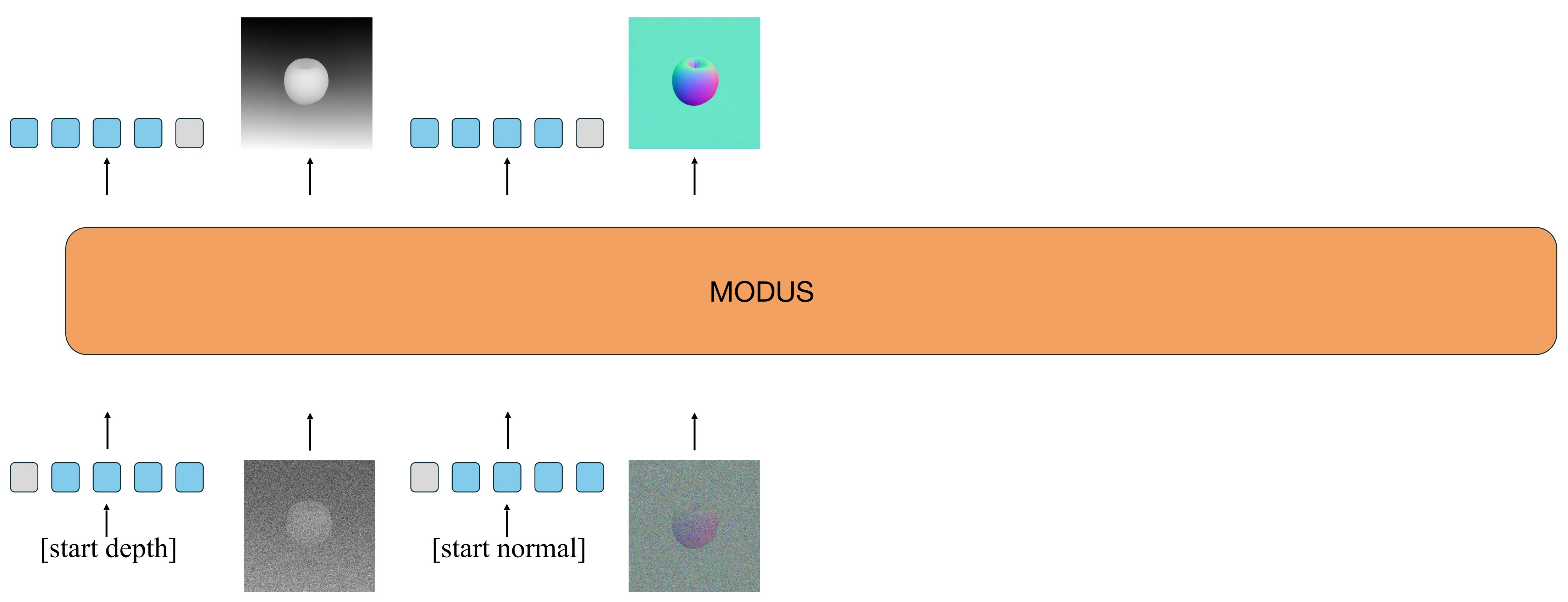

[start depth]     [start normal]

# Model

## Modus: Training with condition and target modalities

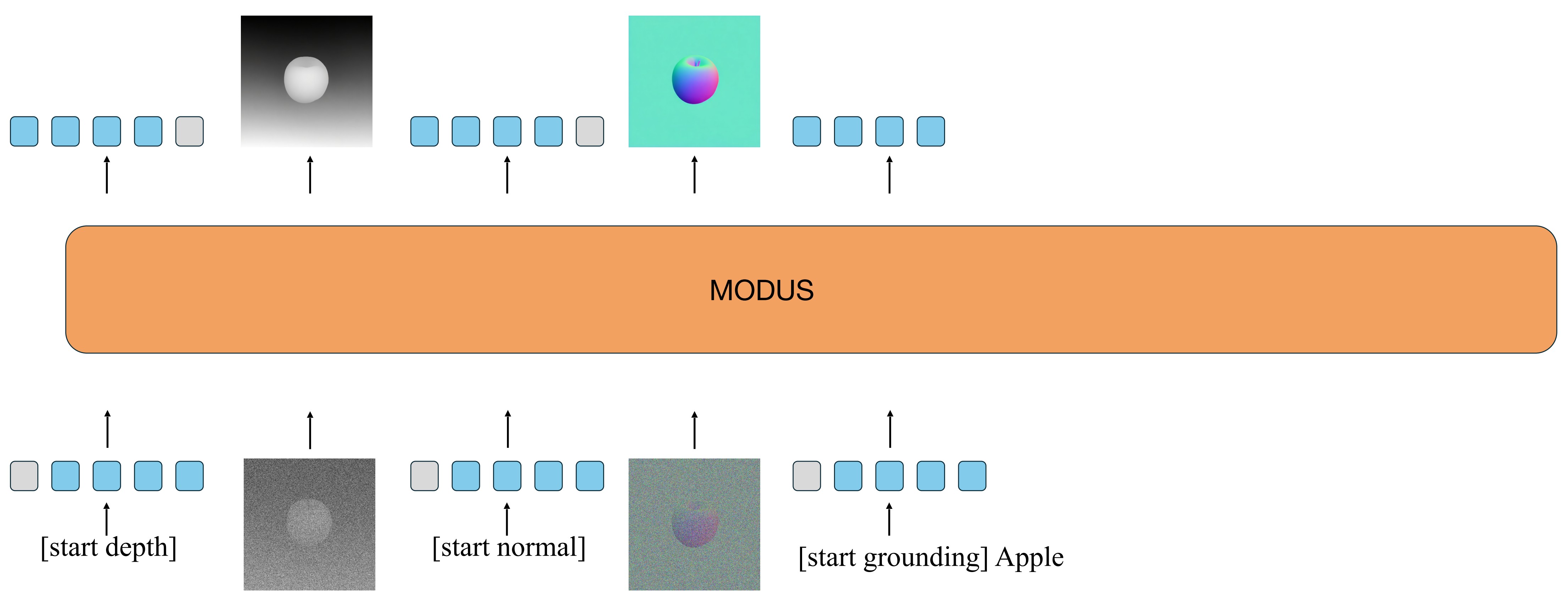

[start depth]        [start normal]        [start grounding] Apple

# Model

## Modus: Training with condition and target modalities

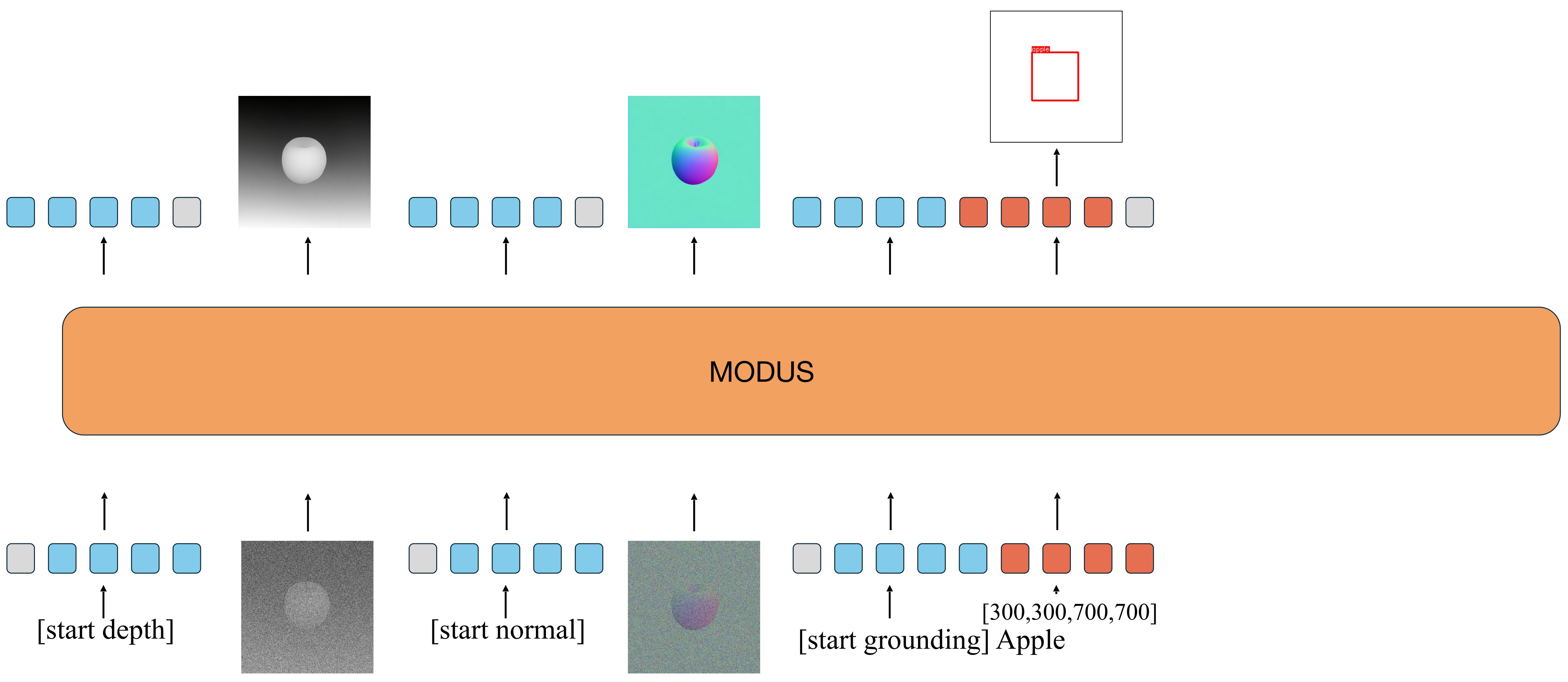

# Model

## Modus: Training with condition and target modalities

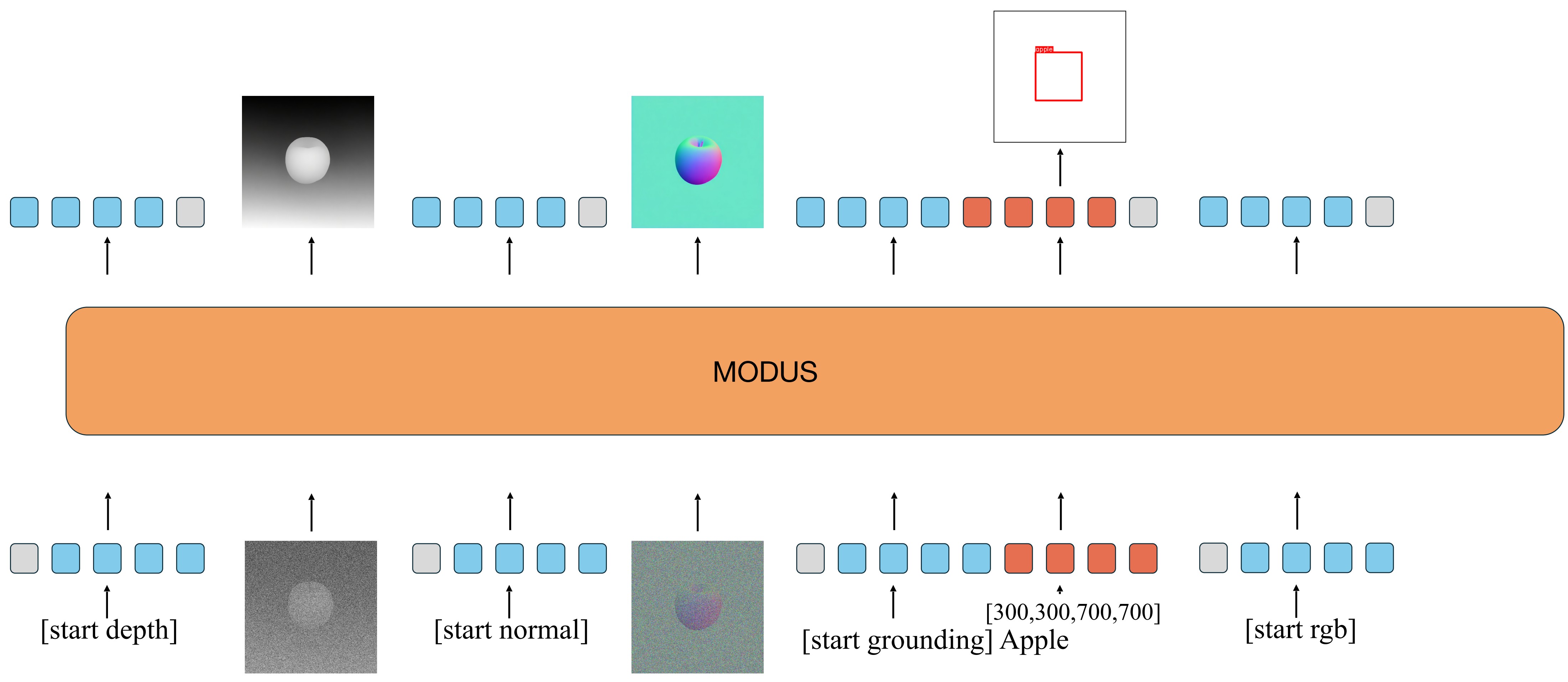

# Model

## Modus: Training with condition and target modalities

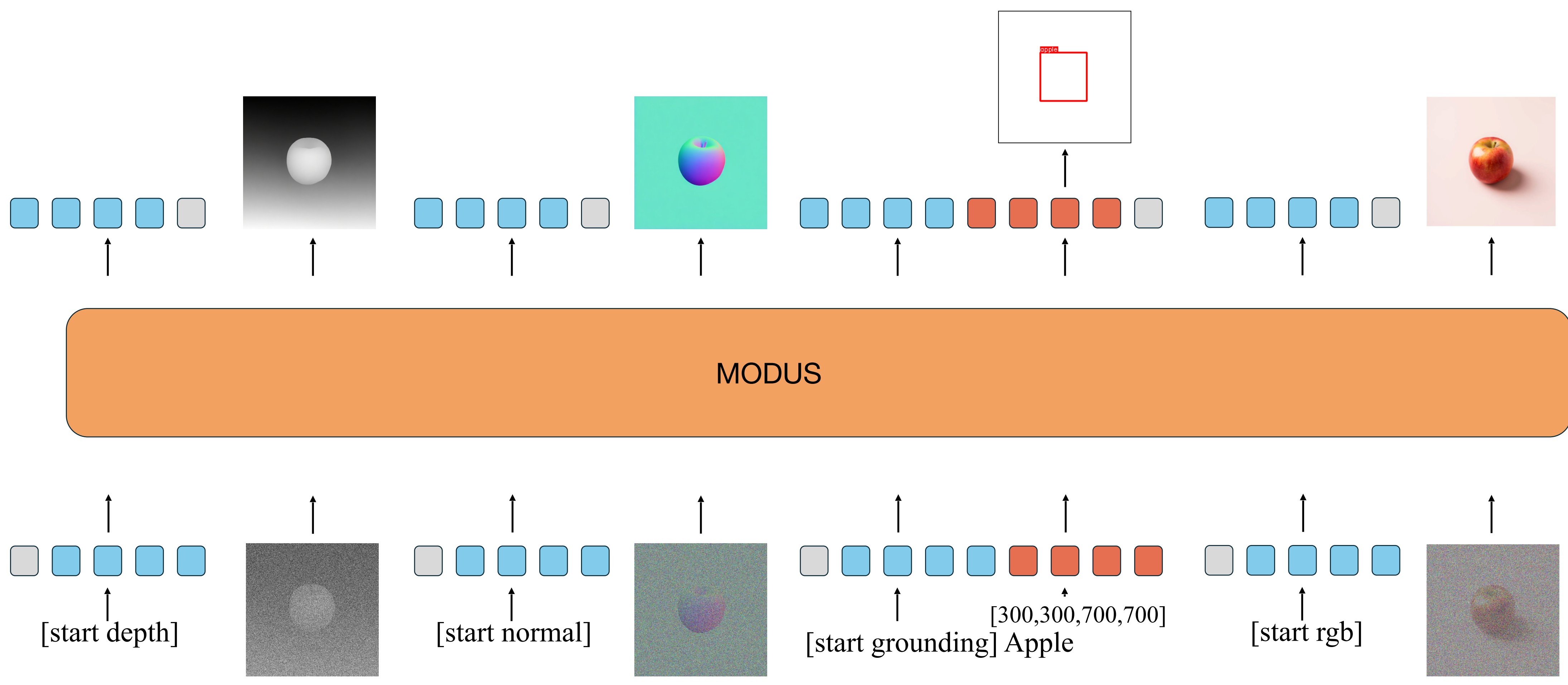

# Model

## Modus Architecture

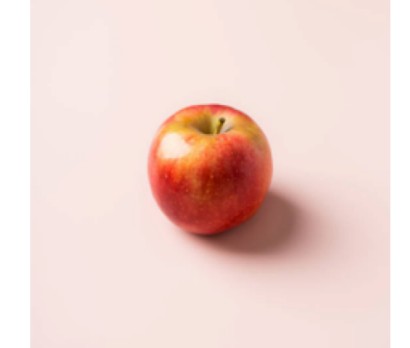

Next Token Prediction

Flow Prediction

**1D Expert**

**2D Expert**

Tokenizer

Und. Encoder

Gen. Encoder

[start rgb]

# Model

**Modus Architecture**
- **1D Expert for sequential modalities with discrete token**
- **2D Expert for spatial modalities with continuous latent**

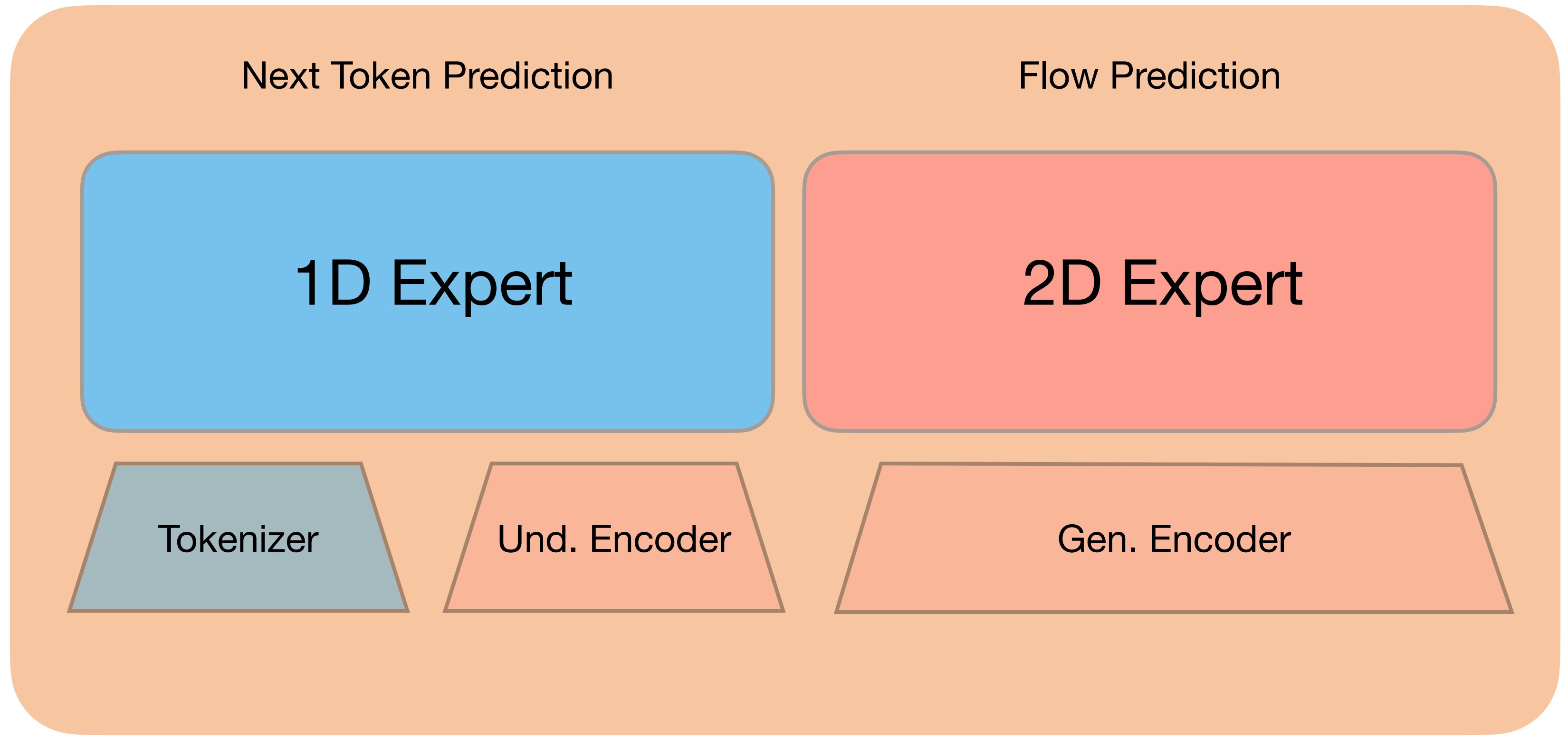

# Model

## Modus Architecture

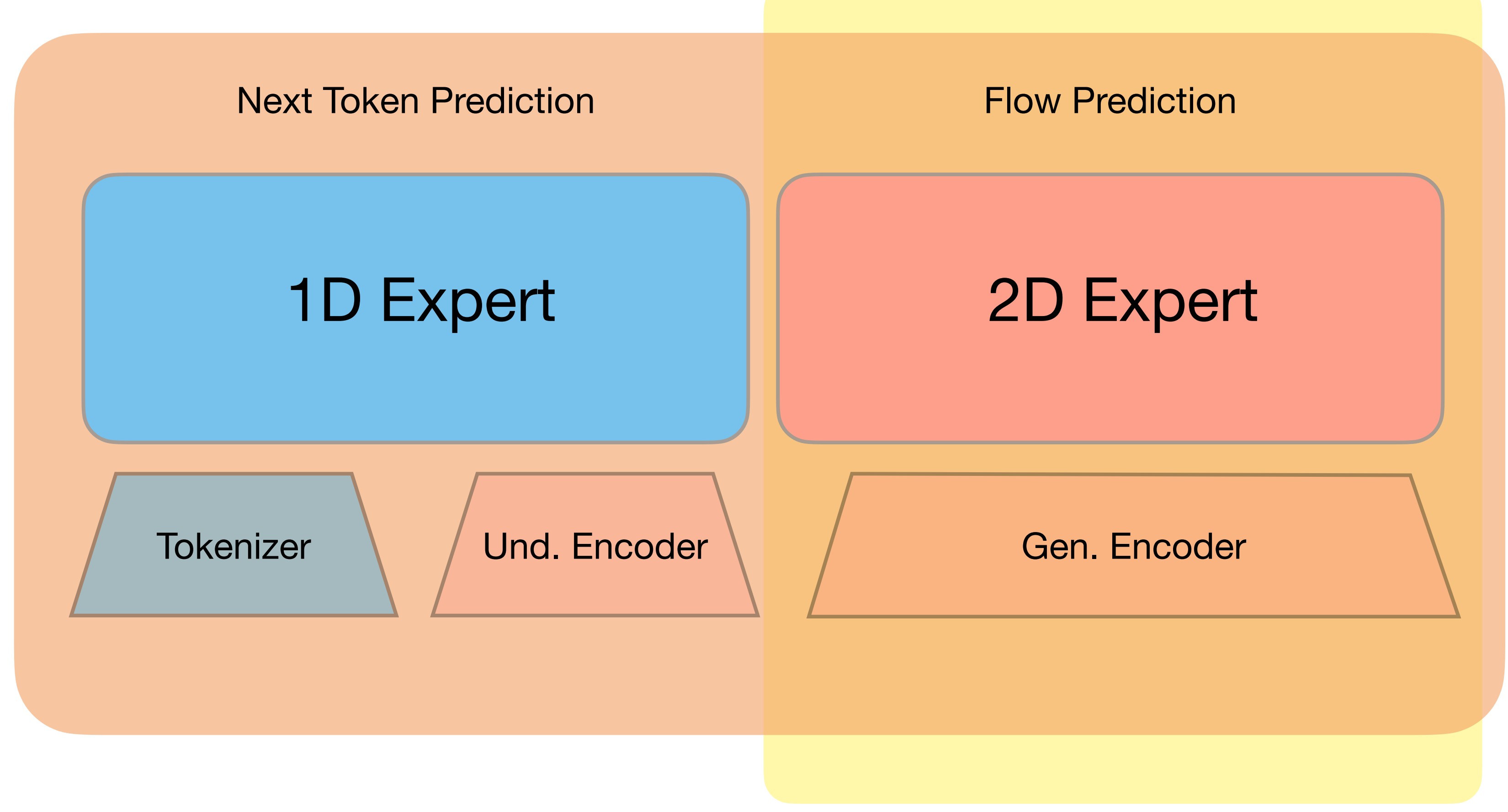

# Model

**2D Expert Training**
- **Modality confusion issue**
- **Timestep sampling matters !**

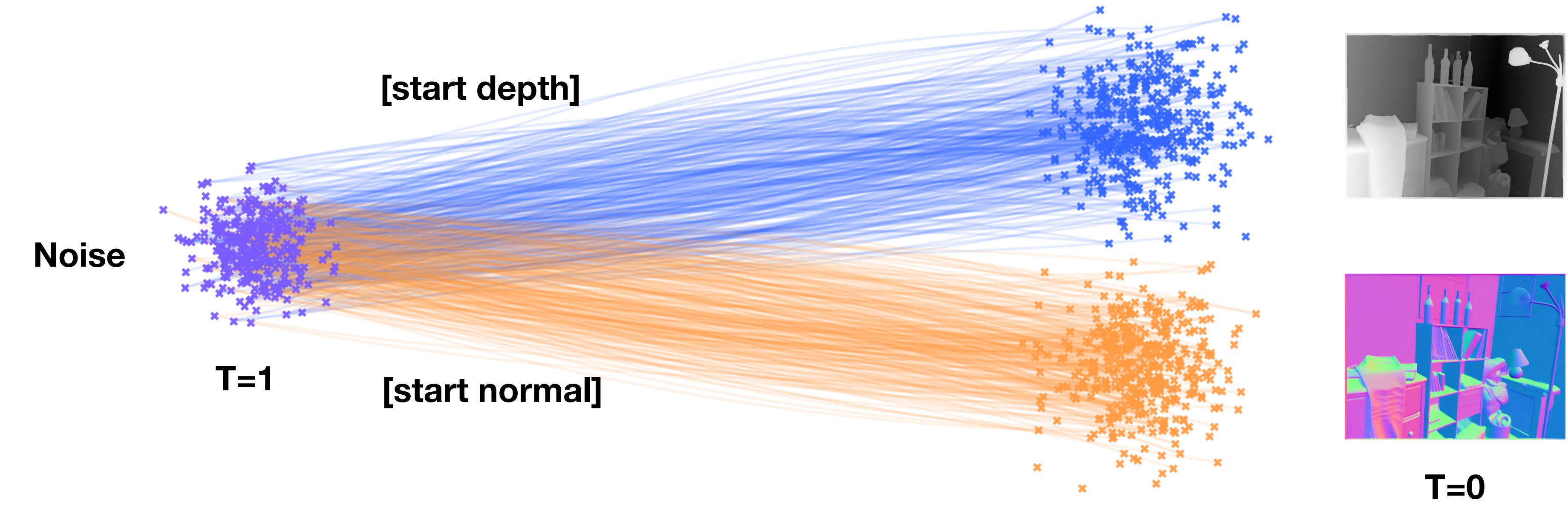

# Model

## 2D Expert Training

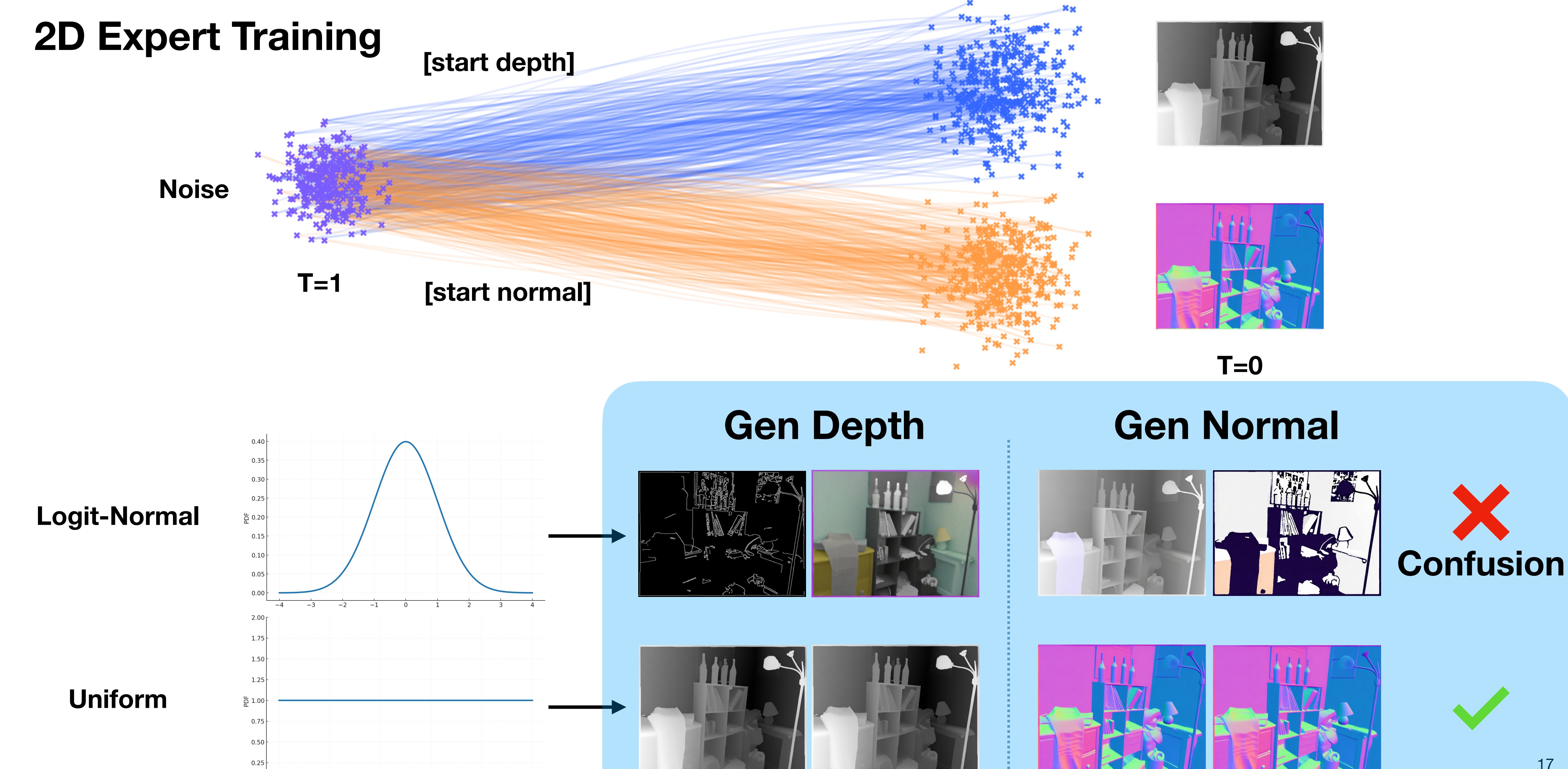

**[start depth]**

**Noise**

**T=1**

**[start normal]**

**T=0**

**Logit-Normal**

**Uniform**

**Gen Depth**

**Gen Normal**

❌ **Confusion**

✅

# Out-of-the-box

- **Unified model for various out-of-the-box tasks**

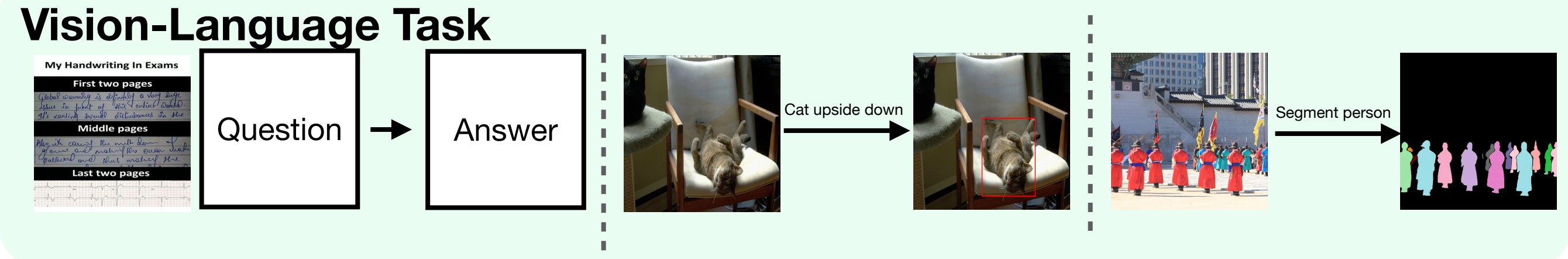

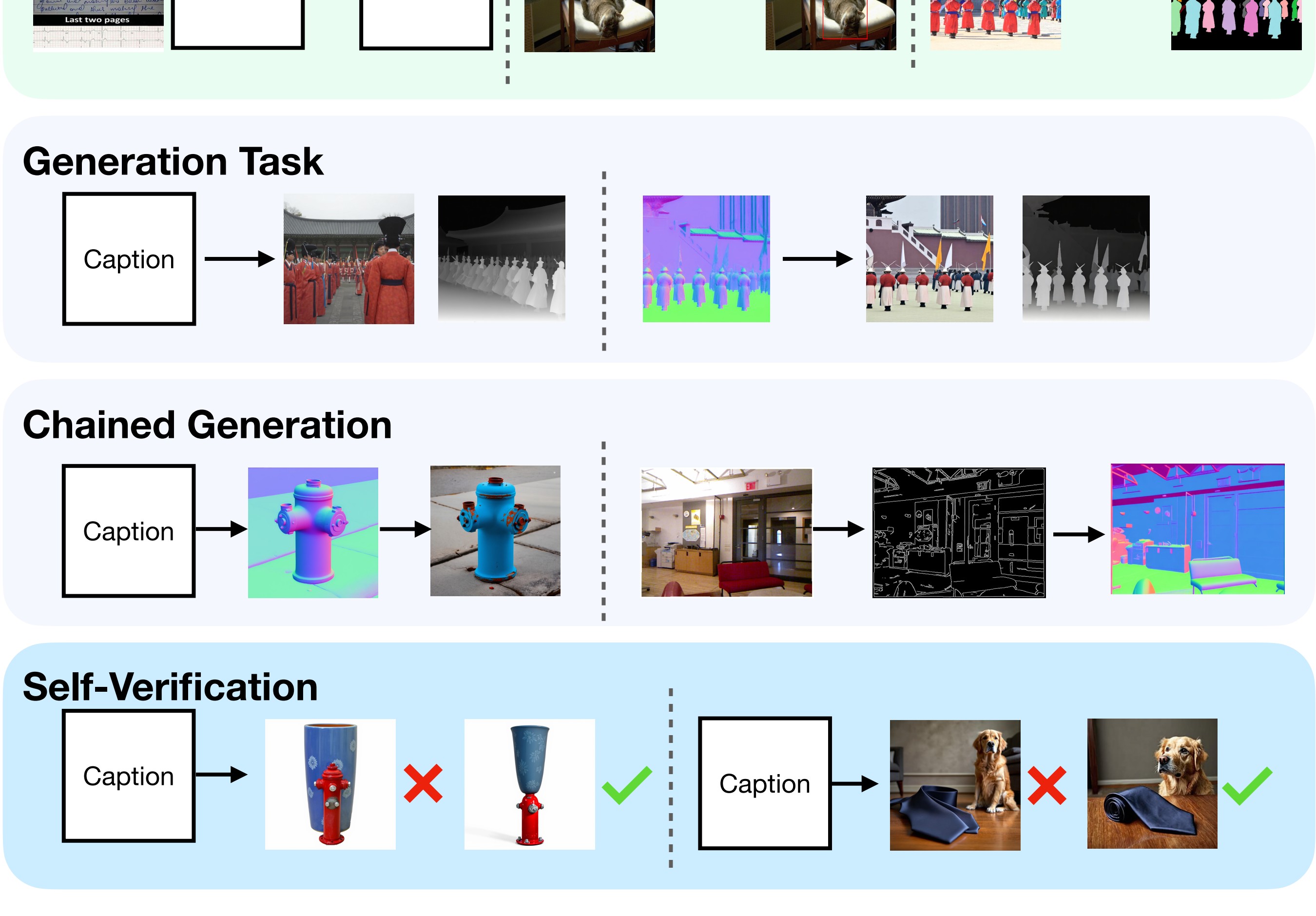

# Any-to-any

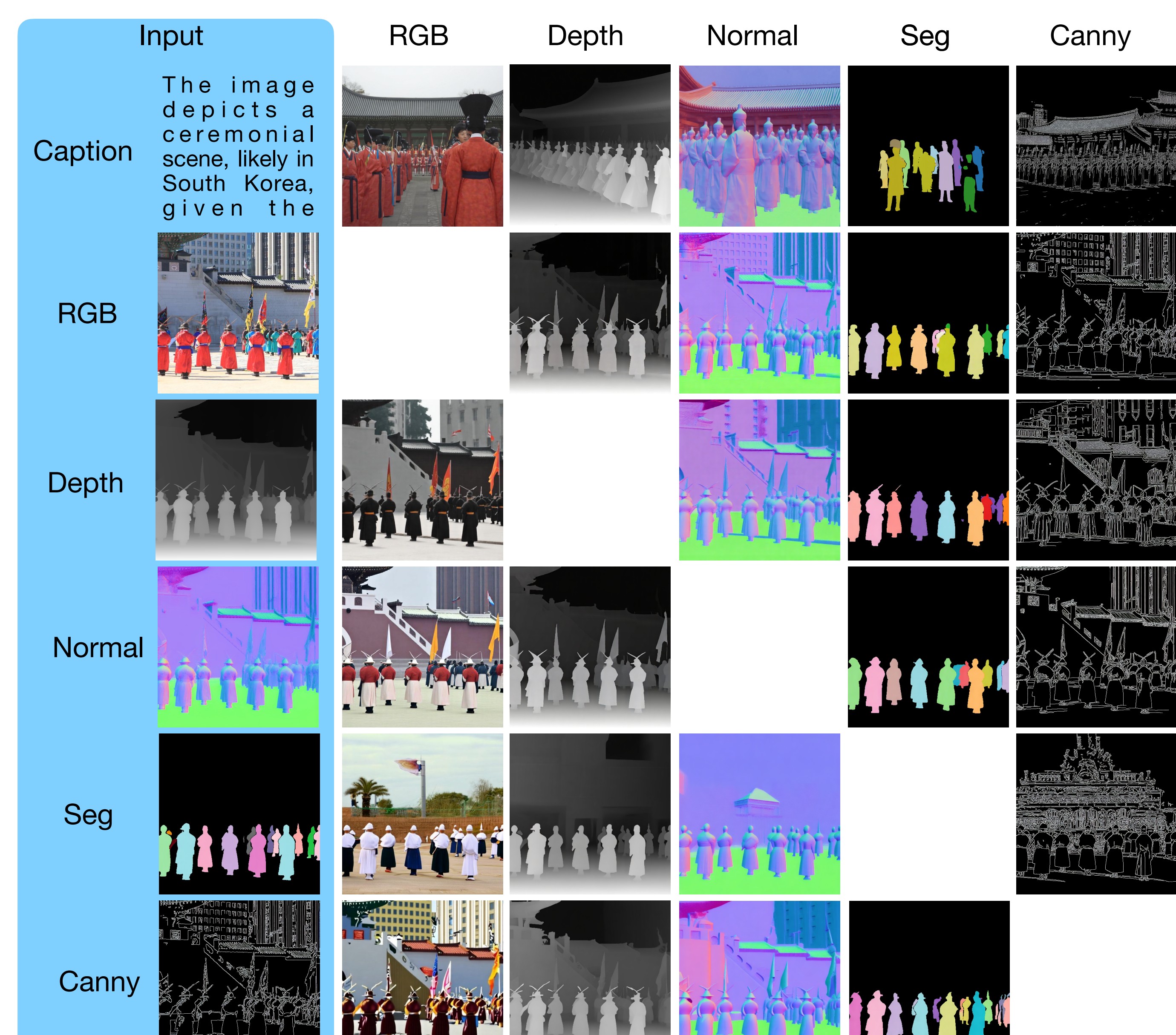

# Any-to-any

| Input | RGB | Depth | Normal | Seg | Canny |
|---|---|---|---|---|---|
| **Caption** The image depicts a ceremonial scene, likely in South Korea, given the | | | | | |
| **RGB** | | | | | |
| **Depth** | | | | | |
| **Normal** | | | | | |
| **Seg** | | | | | |
| **Canny** | | | | | |

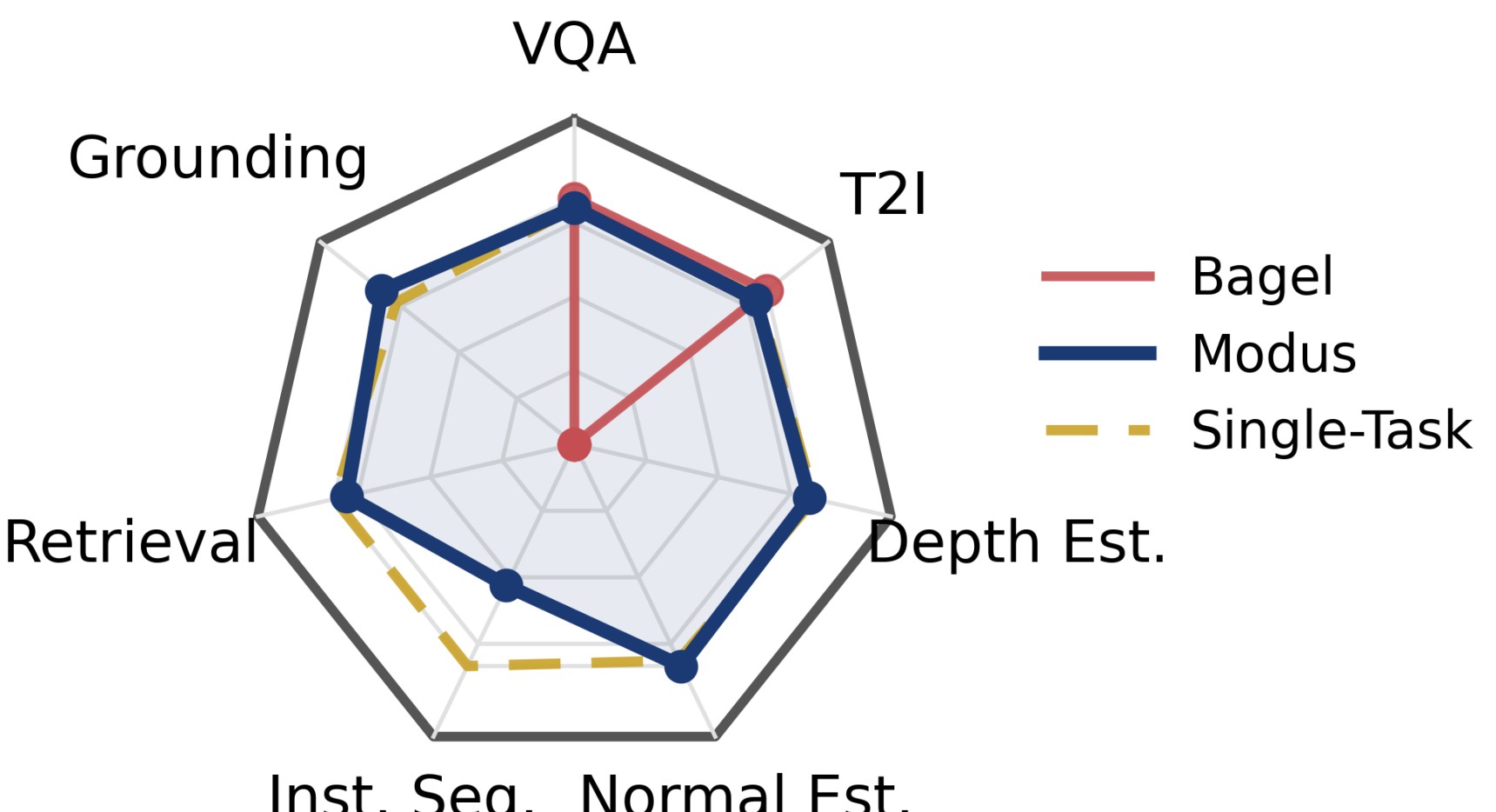

**Extend decoder-only model to diverse modalities**

**Comparable with single-task specialist model**

# Chained Generation

| Input | Intermediate | Target |
|:---:|:---:|:---:|

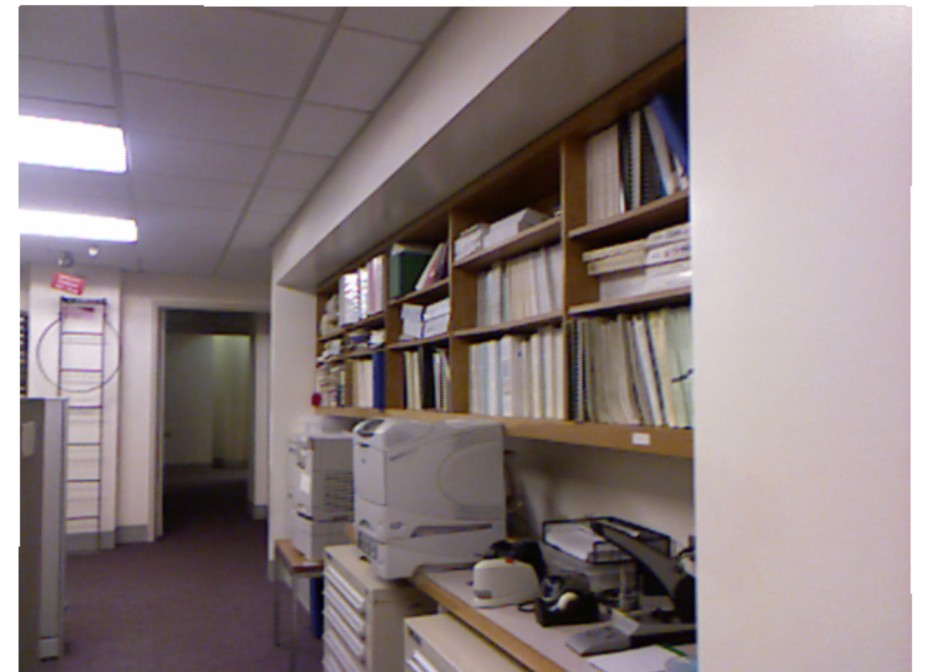

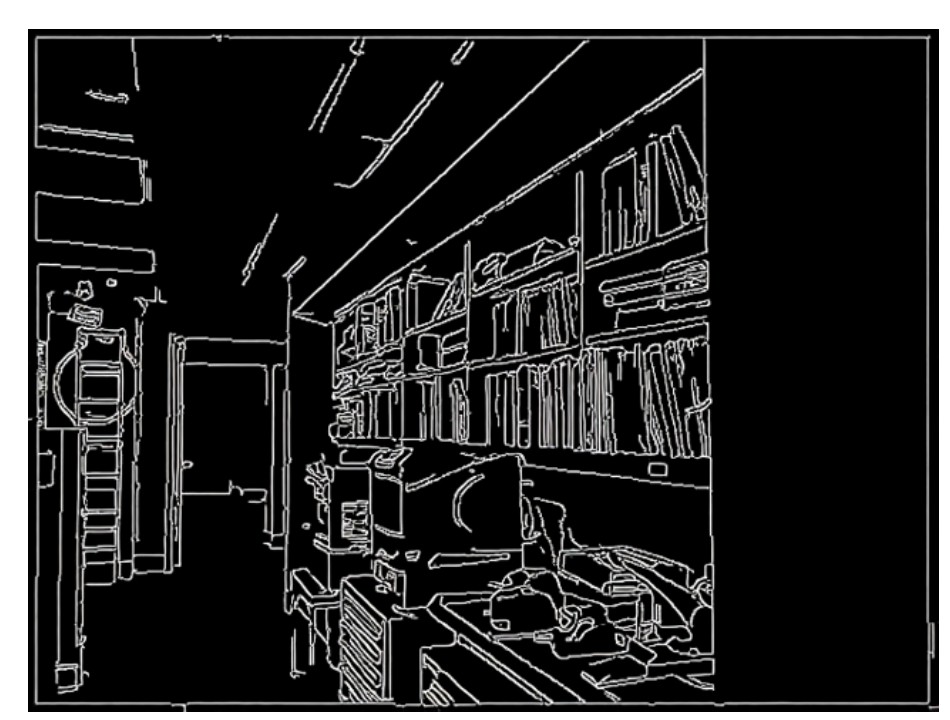

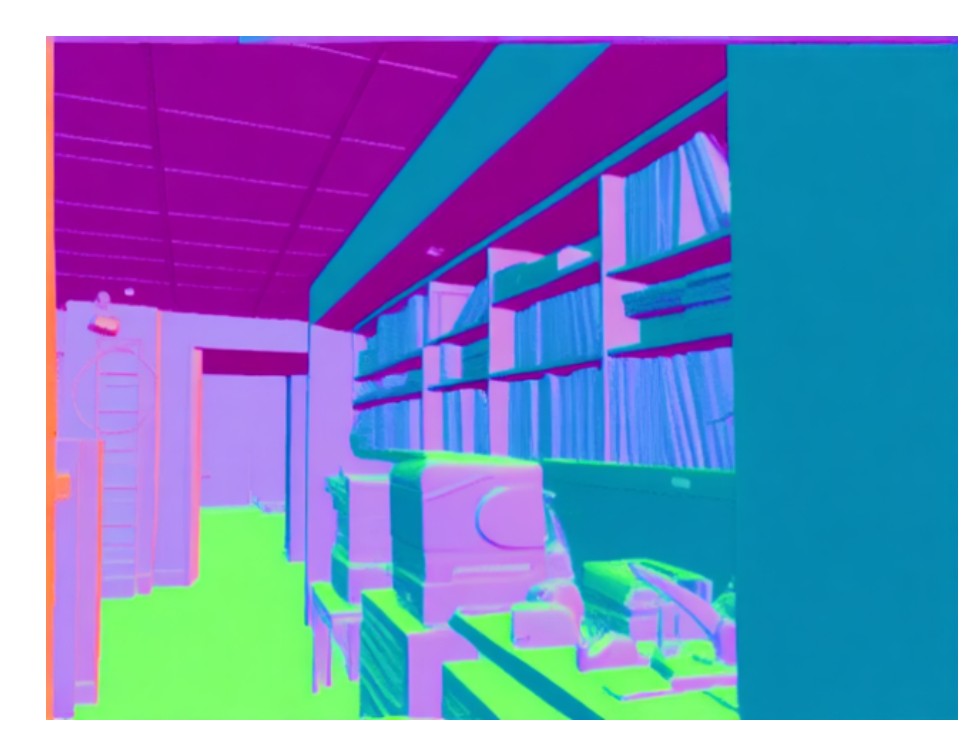

A realistic ceramic pot sits on a flat surface.

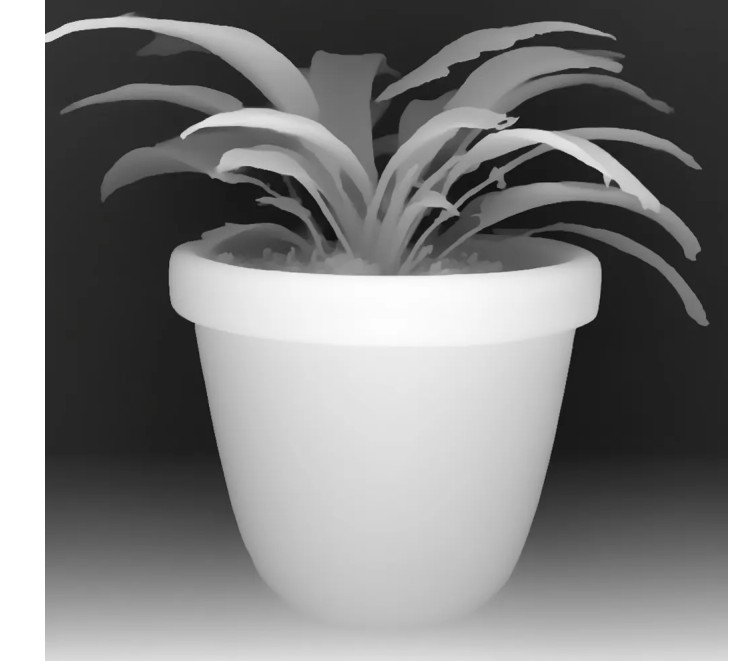

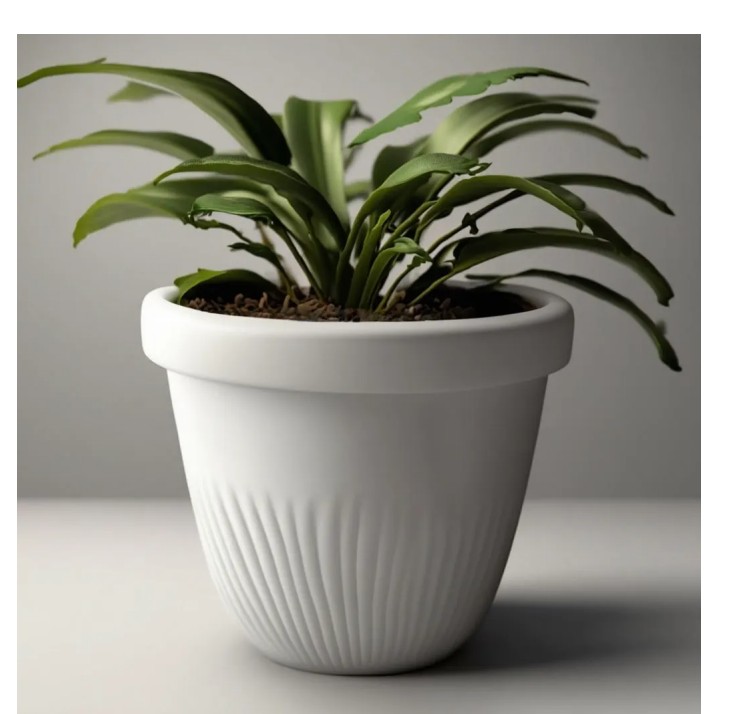

**Consistency across the intermediate and target modalities**

# Self-Verification

A queen-sized bed with a white duvet and two plush pillows is positioned in the center of the room. The bed frame is made of dark wood, and the mattress appears firm and well-structured. **To the left of the bed, a bright red frisbee lies flat** on the hardwood floor. The frisbee has a smooth, glossy surface and a small logo printed near its edge. The room is softly lit by natural light streaming through a nearby window, casting subtle shadows. The scene is rendered in a highly realistic, photographic style, emphasizing the textures of the fabric, wood, and plastic.

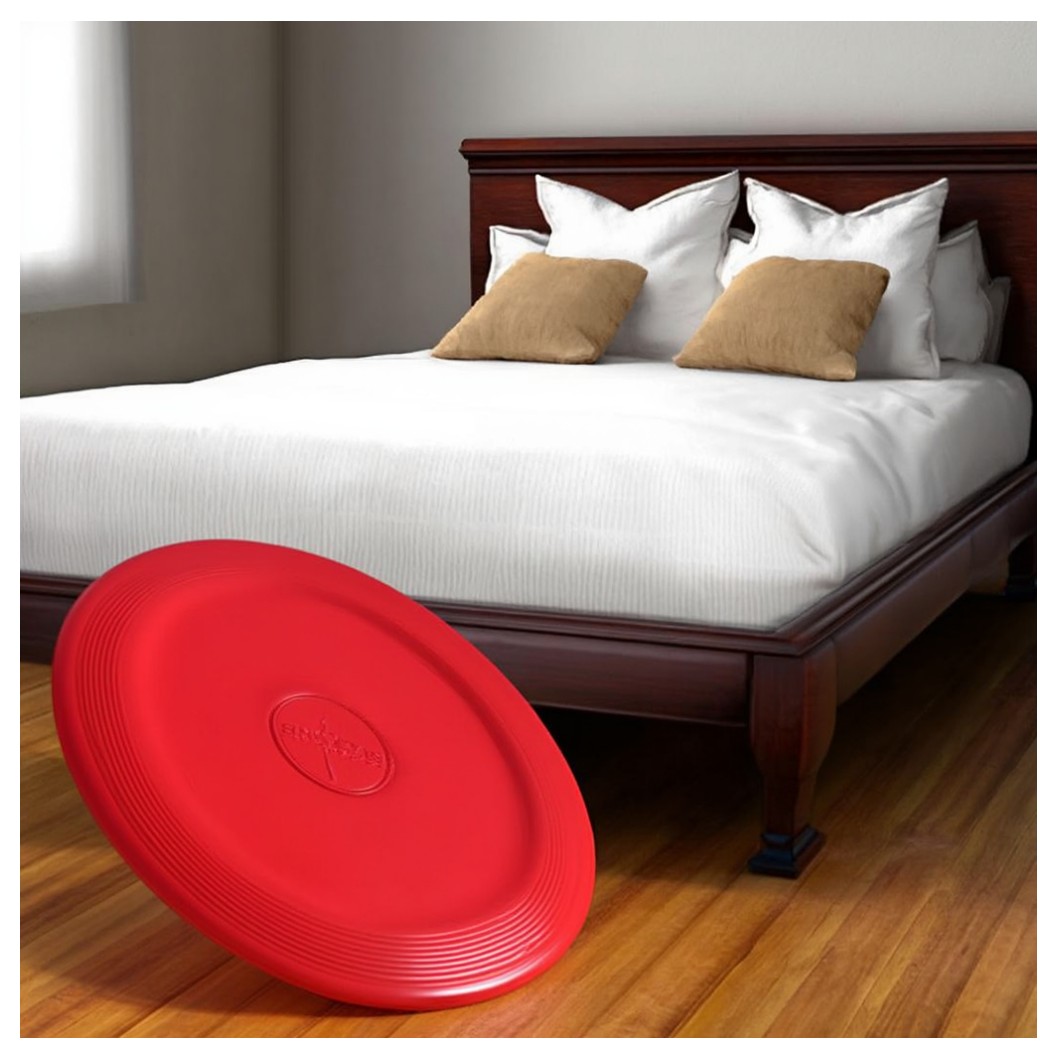 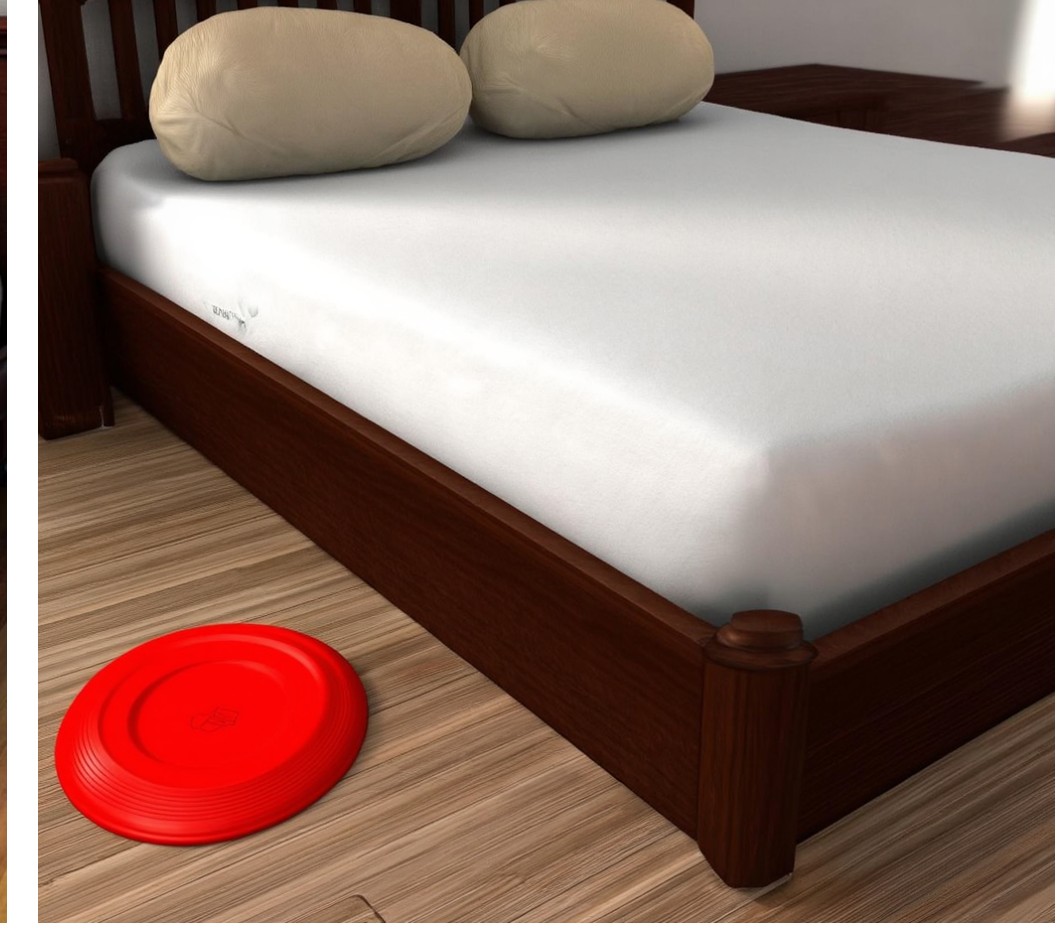

Confidence: **0.056**          Confidence: **0.193**

## Modus Grounding score as a verifier for Modus Generation

# 2D Feature Representation

Modus ViT-Only GPT-4o **Modus ViT+VAE**

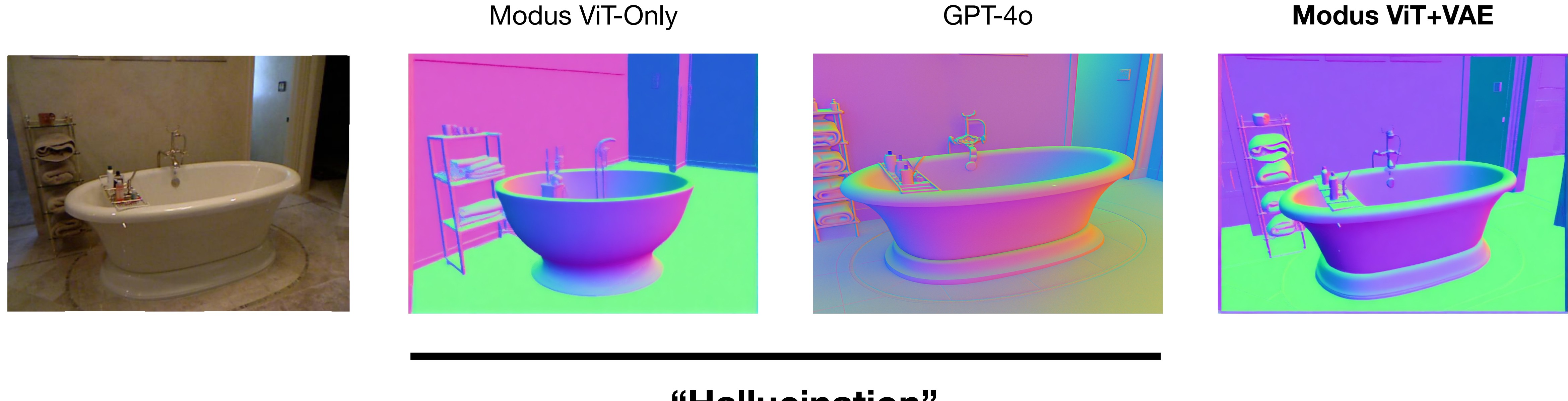

**"Hallucination"**

**Image-to-Normal: Interesting "hallucination" when only conditioning on 2D image high-level semantic features, and similar observation when evaluating GPT-4o**

# Thanks