# OpenReview forum: "MODUS: Decoder-only Any-to-Any Modeling of Diverse Modalities"
_ICML.cc/2026/Conference — ICML 2026 regular_

### Official Review · Reviewer_Riq7 · 2026-03-08

**Soundness:** 2
**Presentation:** 2
**Significance:** 2
**Originality:** 2
**Overall Recommendation:** 3
**Confidence:** 3

**Summary:**

MODUS proposes a unified decoder-only framework for any-to-any multimodal generation and understanding. By mapping various modalities into a shared discrete token space, the model handles diverse tasks through a single autoregressive objective. The authors introduce a staged training pipeline and a uniform timestep sampling strategy to manage multi-condition generation. While the system shows impressive breadth, the conceptual leap from existing unified models remains modest.

**Compliance With Llm Reviewing Policy:**

Affirmed.

**Final Justification:**

Rebuttal not addressed all of my concerns. I keep the score.

**Key Questions For Authors:**

No

**Limitations:**

Yes

**Strengths And Weaknesses:**

Strengths:
(1) The model’s ability to handle such a wide array of modalities and "chained" tasks within a single decoder-only architecture is impressive from a system-engineering standpoint.
Weaknesses:
(1) The core idea—discretizing modalities into tokens and feeding them into a Transformer—is well-trodden ground. This work feels more like an extensive scaling exercise of existing paradigms rather than a fundamental algorithmic breakthrough.
(2) The paper emphasizes "how" the system was built but lacks a "why" regarding certain design choices.
(3) While versatile, MODUS seems to trade off significant precision for generality. It often falls behind specialized models, and the paper doesn't fully address whether this gap is a limitation of the architecture or simply a matter of scale.
(4) While the decoder-only setup benefits from KV caching, the sequential nature of generating high-resolution 2D features across multiple modalities in a chain could be computationally expensive. A more detailed analysis of inference throughput and latency compared to diffusion-based "expert" models is missing.
(5) The current work focuses primarily on vision-centric 2D modalities. It is unclear how the dual-expert system would handle highly disparate modalities such as high-frequency audio or structured scientific data like 3D point clouds, which might introduce new forms of modality competition.
(6) This work lacks novelty and is similar with some Any-to-Any generation models.

---

> ### Author Rebuttal · Authors · 2026-03-31
>
> We thank the reviewer for the detailed feedback and for recognizing the breadth of MODUS. We clarify several core architectural points below.
>
> ### 1. Clarification on Discretization (W1)
>
> The review summarizes MODUS as mapping various modalities into a shared discrete token space and handling diverse tasks through a single autoregressive objective. We would like to clarify an important nuance. This is accurate for 1D sequential modalities such as text, but not for 2D spatial modalities.
>
> As described in Sec. 3, we explicitly do **not** discretize 2D modalities. Instead, they are represented as continuous latents and generated with a **flow-matching** objective. Therefore, the core of MODUS is not to discretize all modalities into one autoregressive space. Rather, it is the unification of an Autoregressive 1D Expert and a continuous Flow-Matching 2D Expert within one shared decoder.
>
> ### 2. Difference from existing any-to-any models (W6)
>
> This design distinguishes MODUS from prior any-to-any paradigms:
>
> - **Encoder-decoder + discretization:** models such as 4M and Unified-IO rely on full discretization of all modalities.
> - **Diffusion-only:** models such as OneDiffusion are effective for spatial data, but do not natively output text or other 1D sequential modalities.
> - **Text-centric pivots:** existing decoder-only models such as AnyGPT and NExT-GPT route non-text data through a language pivot. They do not enforce structural alignment among non-text modalities, making it difficult to guarantee cross-modal consistency or perform direct transformations between non-text representations.
>
> By contrast, MODUS unifies continuous flow matching and discrete autoregression in a single decoder, without modality-specific heads, 2D discretization, or text pivots. This supports chained generation, cross-modal verification, and multi-condition generation.
>
> ### 3. The "Why" Behind Design Choices (W2)
>
> The review notes that the paper emphasizes how more than the why behind some design choices. These questions are addressed by our ablations:
>
> - **Why Uniform Timestep Sampling?** To reduce modality confusion caused by under-sampling early timesteps. As shown in Sec. 3.3 and 4.6.
> - **Why both ViT and VAE for 2D?** They provide complementary semantics and spatial fidelity. As shown in Sec. 4.5.
> - **Why initialize from an image-text model?** For efficiency and performance; BAGEL initialization provides strong priors and robust capability with only 65B tokens training in Table 7.
> - **Why multiple training stages?** Because 1D modalities with new tokenizers converge more slowly than 2D modalities that reuse shared ViT/VAE spaces in Fig11.
>
> We will revise the manuscript to highlight these motivations earlier.
>
> ### 4. Precision, Generality, and Dataset Quality (W3)
>
> Our goal is not to chase SOTA on every specialized benchmark, but to validate a unified paradigm with broader capabilities such as chained generation and cross-modal verification. At the same time, MODUS does not inherently require a large sacrifice in precision. It can be competitive with, and even surpass, specialized models. For example, on RefCOCO val:
>
> | Model | RefCOCO val |
> | --- | --- |
> | GLIP-T | 50.42 |
> | Grounding-DINO-T | 50.41 |
> | **MODUS** | **54.50** |
>
> This suggests that a unified architecture does not fundamentally bottleneck task-specific precision. We observe a similar trend on VQA, as discussed in our reply to reviewer WXBA.
>
> ### 5. Chained generation and Inference Throughput / Latency (W4)
>
> Chained generation can improve prediction quality, provide interpretable intermediate steps, and offer a basis for future multimodal Chain-of-Thought style post-training. Importantly, this does **not** introduce a prohibitive token-by-token autoregressive bottleneck. Our 2D Expert uses parallelized flow matching and remains efficient in practice.
>
> We benchmarked inference throughput at 512×512 resolution on a single GH200 GPU over 654 NYUv2 images:
>
> - Direct generation already gives reasonable performance in 5 steps.
> - Chaining trade-off: 10-step chained generation (4.28s/image) reaches nearly the same performance as 50-step direct generation (9.07s/image).
>
> | Task | Steps | Sec/img | NYU Normal ↓ |
> | --- | --- | --- | --- |
> | RGB→Normal | 2 | 0.59 | 34.06 |
> |  | 5 | 1.14 | 21.91 |
> |  | 10 | 2.05 | 20.30 |
> |  | 50 | 9.07 | 20.02 |
> | RGB→Canny→Normal | 2 | 1.38 | 36.82 |
> |  | 5 | 2.57 | 22.21 |
> |  | 10 | 4.28 | 20.08 |
> |  | 50 | 20.05 | 19.87 |
>
> ### 6. Scalability to Disparate Modalities (W5)
>
> Our current work demonstrates a foundational step: discrete semantic understanding and continuous spatial generation can share one decoder.
>
> Extending to more disparate modalities such as audio or 3D point clouds is beyond the scope of the current paper, but the dual-expert system is naturally compatible with such extensions:
>
> - sequence-like data can map to the 1D Autoregressive Expert,
> - while continuous structured data can map to the continuous Flow-Matching 2D Expert.

---

> > ### Author Rebuttal · Reviewer_Riq7 · 2026-04-03
> >
> > While the authors clarify the structural details of MODUS, the overall contribution remains largely incremental and falls short of a fundamental paradigm shift in multimodal learning. The following points highlight the limitations in its novelty:
> >
> > Engineering Integration vs. Conceptual Innovation: The core of MODUS—combining 1D discrete autoregression with 2D continuous flow-matching—appears to be a pragmatic architectural hybrid rather than a unified theoretical breakthrough. While "sharing a decoder" provides engineering efficiency, it essentially functions as two established paradigms operating in parallel. This "dual-expert" approach does not resolve the underlying challenge of multimodal representation; it simply houses existing solutions under one roof.
> >
> > Marginal Novelty of "Chained Generation": The concept of chained generation (e.g., RGB → Canny → Normal) is a well-explored technique in vision pipelines. Implementing this within a single decoder is a convenient feature, but it does not represent a new capability in the field.
> >
> > Evolutionary, Not Revolutionary: The "Any-to-Any" capability is achieved by concatenating known methodologies (VQ-VAE/ViT for perception and Flow-matching for generation). While the results are competitive, the work lacks a "bridge" mechanism that fundamentally changes how disparate modalities interact. As it stands, the framework is a sophisticated "stitched" model that refines existing workflows without introducing a new generative or discriminative principle.
> >
> > Inadequate Benchmarking: The authors compare MODUS (54.50) against outdated baselines like GLIP (2021) and Grounding-DINO (2024).  Some advanced segmentation models like LISA, LISA++ have obtained 74.9 performance.
> >
> > I think MODUS offers limited insight into new AI paradigms.

---

> > > ### Author Response · Authors · 2026-04-03
> > >
> > > We thank the reviewer for the follow-up. We address each point below.
> > >
> > > ---
> > >
> > > **1. Inadequate Benchmarking**
> > >
> > > As stated in Table 1, all MODUS evaluations are **zero-shot**. Our goal is not to achieve SOTA on every specialized benchmark, but to support a diverse modality dictionary and enable new capabilities within a single model.
> > >
> > > Regarding LISA and LISA++: these methods **explicitly use RefCOCO as training data** and are evaluated **full-shot**. Comparing them against a zero-shot model is not fair. We chose GLIP-T and Grounding-DINO-T precisely because they are also evaluated zero-shot, ensuring a meaningful comparison.
> > >
> > > ---
> > >
> > > **2. Engineering Integration vs. Conceptual Innovation**
> > >
> > > We find it difficult to respond concretely to this criticism. The boundary between "engineering" and "conceptual innovation" is not clearly defined enough to serve as an actionable weakness. Many recognized contributions in the community are fundamentally engineering advances, and this does not diminish their impact. Reviewers positively recognized the direction of MODUS as meaningful (Bs82), important (WXBA), and supported by solid empirical evidence (5ayY).
> > >
> > > Beyond system-level integration, our paper explores **non-trivial design choices**:
> > >
> > > - Uniform timestep sampling to resolve modality confusion (Sec. 3.3, Table 5)
> > > - Joint ViT+VAE representation balancing semantics and spatial fidelity (Table 4, Fig. 8)
> > > - Staged training for efficient convergence when extending pretrained models (Table 6, Fig. 11)
> > > - New capabilities, chained generation, cross-modal verification, visual representation analysis, emerging from the unified design
> > >
> > > ---
> > >
> > > **3. Marginal Novelty of Chained Generation**
> > >
> > > We believe that existing chained generation in vision pipelines remains underexplored in several important aspects:
> > >
> > > - **Lack of diverse chaining support.** Existing works operate on one specific chain during training but do not support an open-ended set of modality combinations. MODUS enables flexible chaining across a broad modality dictionary.
> > > - **Which chains help is not understood.** Table 2 shows RGB→Canny→Normal improves over direct RGB→Normal, while RGB→Depth→Normal does not due to representational redundancy, providing a foundation for future chain-of-modality post-training.
> > > - **Chaining efficiency is underexplored.** Our latency analysis shows 10-step chained generation (4.28s/img) matches 50-step direct generation (9.07s/img), demonstrating that intermediate modalities reduce required denoising steps.
> > > - **Chaining is only one of the capabilities of MODUS.** Any-to-any generation (e.g., normal→depth) and cross-modal self-verification cannot be achieved by specialist models at all.
> > >
> > > ---
> > >
> > > **4. Evolutionary, Not Revolutionary**
> > >
> > > The AR+Flow paradigm has become a widely adopted foundation in recent multimodal models, including Transfusion, JanusFlow, BAGEL, and Hunyuan-Image, reflecting its recognized effectiveness for unified understanding and generation. Meanwhile, all of these models focus only on text and RGB modalities. MODUS extends this paradigm to a broad any-to-any setting covering geometric, structural, semantic, and representational modalities, validated by our ablations.
> > >
> > > Regarding the "bridge" mechanism: the **shared causal attention context** in MoT is precisely such a bridge:
> > >
> > > - Tokens from either expert condition subsequent tokens from the other, enabling **direct cross-modal interaction** without text pivots or modality-specific heads.
> > > - Prior work (BAGEL) shows MoT/MoE **outperforms dense models** for this reason, separate parameters for modality-specific learning, shared attention for cross-modal alignment.
> > > - Fig. 8 and Table 4 confirm this: conditioning the 2D Expert on ViT features from the 1D pathway produces semantically consistent but geometrically varied outputs, showing the experts **actively communicate** through shared context.
> > >
> > > **References**
> > >
> > > [1] LISA: Reasoning Segmentation via Large Language Model
> > >
> > > [2] LISA++: An Improved Baseline for Reasoning Segmentation with Large Language Model
> > >
> > > [3] Transfusion: Predict the Next Token and Diffuse Images with One Multi-Modal Model
> > >
> > > [4] JanusFlow: Harmonizing Autoregression and Rectified Flow for Unified Multimodal Understanding and Generation
> > >
> > > [5] BAGEL: Emerging Properties in Unified Multimodal Pretraining
> > >
> > > [6] Hunyuan-Image 3.0 Technical Report

---

### Official Review · Reviewer_5ayY · 2026-03-09

**Soundness:** 3
**Presentation:** 3
**Significance:** 3
**Originality:** 3
**Overall Recommendation:** 4
**Confidence:** 4

**Summary:**

This paper introduces MODUS, a decoder-only unified framework for any-to-any multimodal generation. Departing from mainstream encoder-decoder or diffusion-based any-to-any models that are often trained from scratch and constrained by modality-specific designs or text-centric pivots, MODUS symmetrically supports diverse 1D sequential and 2D spatial modalities within a single autoregressive decoder, without dedicated heads, losses, or task pipelines for different modalities.

**Compliance With Llm Reviewing Policy:**

Affirmed.

**Final Justification:**

This paper proposes a valuable direction and ultimately achieves a satisfactory implementation of multimodal data processing and output. However, the technical novelty itself is at the borderline. The authors adequately addressed my questions during the rebuttal period, and therefore I have decided to increase my confidence while maintaining my original score.

**Key Questions For Authors:**

1. When the input is a caption, other generated images differ significantly from the RGB image; what causes this discrepancy under good alignment conditions?
2. Why use DINO to extract global features instead of CLIP? Can using CLIP inherently introduce semantic information? Or would a combined feature representation from both encoders (similar to Eagle or Cambrian) be more suitable?
3. The experiments only report overall metrics. For generation tasks, I am curious about which aspects underperform SOTA on GenEval—semantic fidelity or generation quality.

**Limitations:**

Yes.

**Strengths And Weaknesses:**

Soundness:
The work is technically rigorous and standardized, with complete experimental design, sufficient ablation studies, and core conclusions well supported by solid empirical evidence.

Presentation:
The structure is clear and narrative coherent; method and experimental details are complete, positioning against related work is distinct, and reproducibility is strong.

Significance:
It addresses the fragmentation and modality barriers in multimodal modeling and provides a feasible paradigm for unified decoder-only full-modality modeling; however, the modality coverage is limited and not truly full-modality.

Originality:
The training optimization strategy is innovative, and the design of Chained Generation is interesting and insightful.

---

> ### Author Rebuttal · Authors · 2026-03-31
>
> We sincerely appreciate the reviewer’s thoughtful feedback and positive assessment. We address the specific questions below.
>
> ### 1. Caption-to-Any discrepancies and chained generation (Q1)
>
> Direct generation from captions can produce visually different outputs across modalities, as in Figure 4. This is expected because caption conditioning is inherently sparse. Compared with captions, RGB provides dense spatial and appearance constraints. A caption leaves many visual details unspecified, so generation can naturally explore different plausible layouts for each target modality.
>
> To improve consistency, we use **chained generation**, as shown in Figure 12. By generating sequentially (e.g., Text → Edge → RGB), the intermediate modality helps fix the spatial layout before generating the final output. In practice, this produces much stronger structural consistency between intermediate modalities and final outputs. We view this as an advantage of MODUS’s unified decoder design. In contrast, text-pivoted any-to-any models such as AnyGPT or NExT-GPT do not explicitly enforce this kind of structural alignment across non-text modalities.
>
> ### 2. DINO vs. CLIP and joint representations (Q2)
>
> We use representation modalities such as DINO mainly as a proof of concept that MODUS can model abstract representations beyond pixel space. We also agree with the reviewer’s intuition that CLIP and joint feature representations, similar in spirit to systems such as Eagle or Cambrian, are very promising.
>
> Our initial choice to prioritize DINO over CLIP as an explicit output was mainly because CLIP-like semantics are already present on the input side:
>
> - our 2D tokenization uses a joint representation built from a shared high-level ViT and a low-level VAE,
> - and that ViT is a **SigLIP** model, so CLIP-aligned semantics are already part of the input conditioning signal during training, even if not modeled as an explicit output target.
>
> More broadly, our results suggest that joint representations are effective: the ViT+VAE combination outperforms either feature alone (Figure 8 and Table 4).
>
> Motivated by this feedback, we are now extending MODUS to directly output representation-level modalities such as DINOv2, CLIP, and ImageBind.
>
> - We tokenize them into discrete tokens and model them with next-token prediction in the 1D expert, so they can be natively handled within the same unified framework.
> - We include visualizations at the following anonymous link: https://anonymous.4open.science/r/modus-60D9/modus_abstract_representations.png
>
> ### 3. GenEval breakdown: semantics vs. composition (Q3)
>
> | Model | Overall | Single Obj. | Two Obj. | Count | Colors | Position | Color+Pos. |
> | --- | --- | --- | --- | --- | --- | --- | --- |
> | **BAGEL** | 0.86 | 0.98 | 0.94 | 0.74 | 0.94 | 0.84 | 0.74 |
> | **MODUS** | 0.81 | 0.99 | 0.90 | 0.68 | 0.92 | 0.77 | 0.62 |
> | **MODUS (Best-of-4, self-verification)** | 0.84 | 1.00 | 0.90 | 0.75 | 0.94 | 0.80 | 0.65 |
>
> Our analysis suggests that the GenEval gap is mainly concentrated in more difficult compositional constraints, especially **position**, **counting**, and **color+position**, while basic semantic fidelity remains relatively strong.
>
> - During training, we preserve BAGEL’s base capabilities by mixing in the original training tasks and using EMA updates. Related preservation results are also reflected in the VQA comparison discussed in our reply to reviewer WXBA.
> - At inference time, MODUS also has a native self-verification mechanism. A simple **Best-of-4** test-time search, using the model’s own VQA and grounding abilities to score candidate outputs, improves GenEval from 0.81 to 0.84. We plan to explore stronger test-time scaling in future work.
>
> More importantly, MODUS does not only preserve useful base capabilities, it also supports a much wider set of modalities and behaviors, including chained generation, cross-modal verification, and multi-condition generation.
>
> ### 4. Modality coverage and “full-modality” (limitation)
>
> We agree that the current model does not yet provide exhaustive “full-modality” coverage. Our goal in this paper is to validate the unified decoder paradigm using representative modalities spanning semantic, geometric, and abstract feature spaces. The main point is that these diverse modalities can be handled within one shared token-based framework. In our view, extending to additional modalities is mainly a matter of dataset construction and tokenization, rather than a fundamental architectural limitation. We view this as a promising direction for future work. We will clarify this scope more carefully in the revision.

---

> > ### Author Rebuttal · Reviewer_5ayY · 2026-04-03
> >
> > I have no more questions, I will improve my confidence.

---

> > > ### Author Response · Authors · 2026-04-08
> > >
> > > We thank the reviewer for the positive assessment and for improving confidence in the work. We appreciate the recognition of MODUS as "technically rigorous" with "solid empirical evidence", and the Chained Generation design as "interesting and insightful".
> > >
> > > We will incorporate all clarifications and additional experimental results from the rebuttal into the revised manuscript, and release the code, model weights, and SPECTRUM-25M dataset. Thank you again for the valuable feedback.

---

### Official Review · Reviewer_WXBA · 2026-03-12

**Soundness:** 3
**Presentation:** 2
**Significance:** 3
**Originality:** 3
**Overall Recommendation:** 4
**Confidence:** 3

**Summary:**

The paper proposes MODUS, a unified multimodal decoder-based model for any-to-any generation. The model builds on the BAGEL architecture, extending its two-modality setup to a broader multi-modality setting, including segmentation, depth, DINO features, grounding, and other modalities. MODUS can be conditioned on various input modalities and generate a specified target modality. According to the paper, incorporating these additional modalities helps the model achieve solid performance on tasks requiring geometric consistency. The authors also introduce SPECTRUM-25M, a large-scale multimodal dataset with 25 million samples designed to align diverse modalities.

**Compliance With Llm Reviewing Policy:**

Affirmed.

**Final Justification:**

I have decided to increase my score to a weak accept. I think the paper introduces an interesting and valuable research direction, showing a path toward enabling models to process any-to-any modalities, both in input and output. This is a valuable extension of existing work. While I still have some concerns regarding paper presentation — specifically, the description of the dataset and training process in the initial version was somewhat brief on technical details — the authors introduced additional valuable information during the rebuttal that makes the paper more solid.
I also believe that SFT (instruction tuning) phase could strengthen the work further. For now, the evaluation is promising, but MODUS still underperforms the baselines on some benchmarks. I also note the absence of a code release.
Overall, the rebuttal was productive, and I believe this work is a valuable contribution to the community that still should be improved in future work.

**Key Questions For Authors:**

I have the following main questions to the authors:
1) Can you explain the architecture more explicitly, especially the tokenization scheme for each modality?
2) Did you experimented with sequential training on various modalities as a target, or it is left for SFT phase?
3) Why DINO features are chose as separate modalities?
4) How samples for dataset were created, just adding new modalities to the input image?

**Limitations:**

The main limitations of the paper, as I can see are the following:
1) Just a pre-training phase is introduced. As far as I see, the model is not able to choose the specific modality that should be generated for solving one specific task, thus the target modality is restricted by the specific modality request.
2) The benchmark evaluation is quite limited. For now, it is not clear, whether bimodal abilities of the model (image-text or text-image) suffer significantly from the additional training, because the benchmarks used in the experiments are quite limited.

**Strengths And Weaknesses:**

The main strength of the paper can be summarized as follows:
1) Soundness. The paper is overall technically sound. The authors provide a reasonably detailed description of the method and support most of their main claims with experiments and ablations. The experimental section is fairly broad, and the analyses of timestep sampling and staged training make the technical contribution more convincing. But I think some parts of the evaluation are still missing. For example, since MODUS is built on top of BAGEL-7B, which is a very strong image-text model, I would like to see a broader comparison with the original BAGEL on standard image-text benchmarks. Right now only MMMU is shown, but it would be useful to also include more standard VQA and image generation evaluations, to better understand what is preserved and what is lost after extending the model to many modalities.
2) The paper is generally easy to follow, and it is understandable what exactly the authors are trying to claim. The motivation is clear, the overall pipeline is structured well. However, one underpresented part is the model architecture itself, especially the tokenization side. The paper gives the impression of a shared tokenization scheme across modalities, but in practice I suppose that the modalities are unified in one shared sequence, while the actual tokenization mechanisms differ depending on modality type. For example, text, grounding boxes, DINO features, and 2D outputs are handled differently. I think this part should be explained more explicitly in the main paper, because right now it is slightly confusing.
3) The paper addresses an important problem: building a universal multimodal generator where a shared decoder can process different modalities both on the input and on the output side. This is a meaningful direction for multimodal foundation models, especially because many existing works are still mostly restricted to image-text settings. Here, the model is extended to a broader set of modalities, including segmentation, depth, normals, grounding, edges, and DINO features, all inside one backbone. This makes the work relevant for future research on more general multimodal systems. The dataset contribution, SPECTRUM-25M, also increases the practical importance of the work.
4) The proposed method is not fully new from scratch, since it is built upon BAGEL architecture, but I don't see it as weakness. The authors contribution is to propose a method to train the existing architecture on more modalities, which I suppose a huge iterative step in the developing of universal multimodal architecture. Provided training and ablation study is helpful for the community to continue developing such architectures.

The main weaknesses of the paper is as follows:
1) While the authors introduce new dataset and claim a large-scale multimodal dataset with 25M samples, it is still not fully clear from the paper how exactly one training sample is defined. My understanding is that the dataset consists of aligned image-text pairs augmented with multiple pseudo-labeled modalities such as segmentation, depth, normals, grounding, canny edges, and DINO features. So the model is mainly trained to translate between aligned modalities, rather than to solve downstream tasks directly. This is a valid setup, but I think the paper should explain it more clearly in the main text. Like whether for each sample we have all the modalities presented or not?
2) The model cannot directly generate several modalities at once. It can do chained generation, which is interesting, but it does not directly produce several target modalities jointly in one pass. Did the authors considered training in such scheme, when there can be different target modalities as well?
3) Although the experimental section is broad, I still miss several evaluations. In particular, there is limited evaluation on standard image-text understanding benchmarks besides MMMU, and there is not enough comparison of generated images with stronger baselines. I would also like to see more evaluation on simple VQA benchmarks, possibly high-resolution benchmarks, and a clearer comparison with the original BAGEL model on the image-text side.
4) It is also not fully clear what training budget (GPU-hours) was required for the final model.
5) I also didn't see the motivation on using DINO features as a separate modality. If the model can already generate RGB images (even from BAGEL setup), then the paper should explain more clearly what specific role DINO features play, and why they are useful enough to be modeled explicitly. Right now it feels a bit underjustified, especially since the paper itself shows that DINO-based chaining does not always help on geometry-related outputs, e.g.

---

> ### Author Rebuttal · Authors · 2026-03-31
>
> We sincerely thank the reviewer for the constructive feedback and for recognizing the importance of MODUS. We address these points below.
>
> ### 1. Architecture, tokenization, and dataset mechanics (W1, Q1, Q4)
>
> Some mechanics were underrepresented and will be clarified in Sec. 3 and the main text.
>
> **SPECTRUM-25M.** Each training sample is an aligned tuple of the same image with all modalities (e.g., RGB, text, depth, normals, boxes, DINO), built by extending BLIP-3o with pseudo-labelers.
>
> **Unified sequence, separate tokenizers.** All modalities are placed in one shared causal sequence, but use different tokenization schemes:
>
> - 1D modalities use each modality's discrete tokenizer.
> - 2D modalities use a shared dual representation: a high-level ViT (SigLIP 2) and a low-level VAE (FLUX).
>
> **Training formulation.** We randomly sample 1-3 modalities as conditions and 1 as the target:
>
> - 1D tokens are processed by the 1D expert with causal attention and cross-entropy loss.
> - 2D ViT tokens also go through the 1D expert, but with bidirectional attention.
> - 2D VAE tokens go through the 2D expert:
>     - clean latents when used as conditions,
>     - noisy latents with flow matching when used as targets.
> - Cross-modality attention remains causal.
>
> ### 2. Preserving image-text capabilities (W3, Limitation 2)
>
> Extending a strong base model like BAGEL should not significantly degrade image-text capabilities. To mitigate this, we include **LLaVA-OneVision** during any-to-any training and use **EMA** (App. E). Beyond MMMU, we now evaluate MODUS on a broader set of image-text benchmarks:
>
> | Benchmark | BAGEL | + LLaVA-OV | + Spectrum-25M | MODUS |
> | --- | --- | --- | --- | --- |
> | **MMMU** | 53.2 | 52.3 | 47.9 | 51.1 |
> | **MME-P** | 1687.00 | 1663.56 | 1157.99 | 1637.69 |
> | **MME-S** | 2377.17 | 2316.77 | 1560.85 | 2313.40 |
> | **POPE** | 87.23 | 87.77 | 73.67 | 87.27 |
> | **VizWiz** | 59.41 | 57.77 | 8.34 | 63.60 |
> | **DocVQA** | 94.05 | 93.37 | 90.02 | 90.11 |
> | **ChartQA** | 86.72 | 85.47 | 81.09 | 82.45 |
>
> - BAGEL is trained on multiple high-quality VQA datasets, so some drop is expected when our LLaVA-OV tuning does not include the full original VQA mixture.
> - In contrast, training only on Spectrum-25M enables any-to-any but causes a much larger drop, highlighting the importance of data mixture.
> - Mixed training with EMA substantially mitigates this degradation: performance improves on VizWiz, stays nearly unchanged on POPE, and remains competitive on document/high-resolution benchmarks.
> - We will add this table to the main text.
>
> ### 3. Multi-target generation and DINO motivation (W2, W5, Q2, Q3)
>
> **Multi-target generation.** At inference, MODUS naturally supports 1-to-many outputs (e.g., text generating RGB, depth, and normals together). During training, we restrict each sample to one target for efficiency: a 2D target requires noisy latents, while a 2D condition uses clean latents. Supporting multiple 2D targets means intermediate targets need both and would add thousands of tokens to each sample and reduce throughput.
>
> **Why latent features such as DINO?**
>
> - **Strong semantic prior.** Latent representations such as DINO, CLIP, and ImageBind capture high-level structure beyond raw pixels. Prior works such as 4M, REPA, and RAE suggest that modeling such compressed representations provides strong semantic priors and a richer prototype dictionary for downstream tasks.
> - **Concrete utility.** For DINO specifically, Table 1 already shows utility through ImageNet retrieval.
> - **Broader modality dictionary.** Modeling such latent features expands the set of intermediate modalities available to the model and provides a better foundation for future chain-of-modal reasoning.
>
> DINO is both task-relevant and a proof of concept for modeling abstract representation-level modalities.
>
> ### 4. Compute budget and future agentic routing (W4, Limitation 1, Strength4)
>
> **Compute.** MODUS is trained in three stages on 64 GH200 GPUs for 35, 31, and 22.5 hours, totaling about **5,664 GH200 GPU-hours**. By comparison, 4M requires about **16.9k GH200-equivalent GPU-hours**. We will add these to the paper.
>
> **Agentic routing.** In this paper, MODUS is a pretrained foundation model. Allowing it to choose intermediate modalities is a promising post-training future direction.
>
> **Generality.** We also applied the same training pipeline to Janus-Flow-1.3B and obtained similar any-to-any capabilities while preserving much of its original ability.
>
> | Model | Scale | GH200 hrs | MMMU | DIODE ↓ | NYUv2 ↓ |
> | --- | --- | --- | --- | --- | --- |
> | **Janus-Flow (Base)** | 1.3B | 19.2k | 29.3 | X | X |
> | **Janus-Flow + MODUS** | 1.3B | 1.5k | 27.1 | 0.304 | 36.53 |
> | 4M | 2.8B | 16.9k | X | 0.331 | 37.28 |
>
> This result shows that the same pipeline can extend a different base model to any-to-any multimodal generation while preserving much of its original capability.

---

> > ### Author Rebuttal · Reviewer_WXBA · 2026-04-05
> >
> > I thank the authors for the detailed response. Most of my concerns are resolved. I still suppose that adding SFT phase into the work would make it more complete, because it is interesting how this additional information can improve the final score of the model. I also encourage the authors to share their code to the community. Overall, I am willing to increase my score, and think that the paper introduce quite interesting research direction and results.

---

> > > ### Author Response · Authors · 2026-04-08
> > >
> > > We thank the reviewer for the constructive engagement and for raising the score. We appreciate the recognition that MODUS introduces an "interesting and valuable research direction" for any-to-any generation, and are glad the rebuttal productively resolved concerns on architecture details, dataset mechanics, and evaluation breadth.
> > >
> > > **SFT and beyond.** We agree that instruction tuning is a natural and important next step. The current work establishes the pre-training foundation for any-to-any generation. SFT is a clear follow-up direction to further sharpen task-specific performance. Beyond that, we also plan to explore RL-based approaches for agentic modality routing, allowing the model to autonomously select intermediate modalities during chain-of-modality reasoning.
> > >
> > > **Code release.** We will release the code, model weights, and the SPECTRUM-25M dataset.
> > >
> > > We will incorporate all clarifications and additional experimental results from the rebuttal into the revised manuscript. Thank you again for the valuable feedback.

---

### Official Review · Reviewer_Bs82 · 2026-03-16

**Soundness:** 3
**Presentation:** 3
**Significance:** 3
**Originality:** 3
**Overall Recommendation:** 4
**Confidence:** 4

**Summary:**

This paper presents MODUS, a decoder-only any-to-any multimodal model that aims to unify multiple modalities within a single pretrained backbone. Built on BAGEL-7B, the model handles both 1D sequential modalities and 2D spatial modalities in a shared decoder. The authors introduce SPECTRUM-25M, which extends the BLIP-3o image-text corpus with additional modality annotations, to support training at scale. They further propose uniform timestep sampling and a three curriculum training stages to reduce modality confusion. The paper reports results on VQA, text-to-image generation, depth estimation, normal estimation, grounding, and retrieval, and also demonstrates capabilities such as cross-modal self-verification, and visual representation.

**Compliance With Llm Reviewing Policy:**

Affirmed.

**Final Justification:**

The authors addressed the most of my concerns in the rebuttal. Overall, this work presents some empirical improvements and supports them with good experimental evidence. Therefore, I will keep my score as Weak Accept.

**Key Questions For Authors:**

1. SPECTRUM-25M relies heavily on pseudo labels produced by strong external models such as DepthAnything, Marigold, and Grounded-SAM. Can the authors clarify whether these teacher models’ training data overlap with any evaluation benchmarks used here, and whether any contamination checks were performed?
2. The current results may reflect a combination of BAGEL initialization, additional modality supervision, and the unified any-to-any architecture. Can the authors provide more comparison against alternatives such as BAGEL with modality-specific heads.
3. Can the authors add more evaluations for direct modality-to-modality and multi-condition tasks, beyond the current chained normal-estimation example? For instance, depth→normal and edge→depth.

**Limitations:**

Yes.

**Strengths And Weaknesses:**

Strengths

1. This work tries to explore whether the any-to-any systems can be trained with a pretrained decoder-only paradigm, which is a meaningful direction.
2. The overall architecture design is clean. Rather than introducing a separate task pipeline for each modality, MODUS relies on a unified tokenization and a shared decoder with 1D/2D experts.

Weaknesses

1. Some headline claims are stronger than the evidence shown. In Table 1, MODUS does not dominate all shared benchmarks: for example, BAGEL is stronger on MMMU (53.2 vs. 51.1) and GenEval (0.86 vs. 0.81), GPT-4o is much stronger on MMMU (69.1), and 4M-21 is slightly better on retrieval top-1 (78.3 vs. 77.9). Therefore, phrasing such as “surpassing the multitask baselines” may be somewhat overstated.
2. The paper’s support for the “any-to-any” claim is still somewhat limited quantitatively. The authors do show direct modality-to-modality behavior and one chained quantitative result in Table 2, but most standardized benchmarks are still RGB/text-centric.
3. The role of pseudo-label supervision deserves more discussion. SPECTRUM-25M is constructed using strong off-the-shelf models, explicitly including DepthAnything, Marigold, and Grounded-SAM. This raises a question that how much of the observed performance comes from unified cross-modal modeling versus distilling multiple teacher models.

---

> ### Author Rebuttal · Authors · 2026-03-31
>
> We thank the reviewer for the constructive feedback. We are encouraged that you find our decoder-only any-to-any direction meaningful and the architecture clean. We respond to the main concerns below.
>
> ### 1. Some headline claims are stronger than the evidence shown (W1)
>
> We agree that some wording in the paper was too strong. Our intention was not to claim that MODUS uniformly outperforms all baselines. Rather, our point is that a decoder-only any-to-any model can be competitive with strong baselines (e.g., 4M, OneDiffusion), while also enabling capabilities such as VQA and cross-modal verification that are not jointly supported by prior systems.
>
> **Paper updates:** We will remove wording such as “surpassing the multitask baselines” and clarify the claim more precisely: MODUS is a unified decoder-only model supporting broad multimodal capabilities, rather than a model achieving state-of-the-art on every individual task.
>
> ### 2. More evaluations for direct modality-to-modality and multi-condition tasks (W2, Q3)
>
> Since most standardized benchmarks are RGB/text-centric, we added evaluations beyond standard RGB/text settings. We organize them into **standard mappings**, **multi-condition mappings**, and **any-to-any mappings**.
>
> | Mapping | Source → Target | Dataset | MODUS |
> | --- | --- | --- | --- |
> | Standard | RGB → Depth | DIODE ↓ | 0.285 |
> |  | RGB → Normal | NYUv2 ↓ | 20.02 |
> | Multi-cond. | RGB + Depth → Normal | NYUv2 ↓ | 19.58 |
> |  | Edge + Depth → Normal | NYUv2 ↓ | 19.72 |
>
> These results highlight Modus supports **multi-condition generation** within the same pretrained model, and the benefit is not limited to adding RGB as an extra condition. While the inputs are different and results are not directly comparable, **Edge + Depth → Normal** (19.72) performs similar to the standard **RGB → Normal** mapping (20.02), even without RGB input. This suggests that MODUS is not simply relying on RGB as a privileged input, but can combine complementary signals across modalities within one unified model.
>
> | Mapping | Source → Target | Dataset | MODUS |
> | --- | --- | --- | --- |
> | Any-to-any | Edge → Depth | DIODE ↓ | 0.302 |
> |  | Depth → Normal | NYUv2 ↓ | 29.51 |
>
> Modus also supports direct **any-to-any generation** between non-standard modality pairs. Teacher models such as DepthAnything or Marigold rely on fixed RGB encoders and cannot directly accept sparse cues (e.g., Canny edges) or multimodal combinations (e.g., RGB+Depth) without retraining. While mappings such as **Edge → Depth** are naturally weaker than **RGB → Depth** because the input is much sparser, the key point is that MODUS can perform these transfers within one pretrained model and without task-specific pathways.
>
> ### 3. The role of pseudo-label supervision and potential data overlap (W3, Q1)
>
> We carefully checked possible overlap between teacher training data and our evaluation benchmarks:
>
> - **Teachers:** DepthAnything V2 and Marigold are not trained on DIODE and NYUv2, respectively.
> - **Grounding:** Our GLaMM-based pipeline uses a subset of SA-1B and keeps COCO zero-shot, and we do not use COCO for training.
> - **Features:** The DINOv2 representation model is not trained on our evaluation ImageNet splits.
> - **MODUS data:** SPECTRUM-25M is built on BLIP-3o sources such as SA-1B, JourneyDB, and CC12M, which do not directly overlap with our evaluation benchmarks.
>
> Overall, MODUS does not simply reproduce teacher behavior. Instead, it unifies these mappings in one pretrained model and supports capabilities beyond the original teachers, including direct any-to-any generation, multi-condition generation, chained generation, and verification.
>
> ### 4. Comparison against alternatives such as BAGEL with modality-specific heads (Q2)
>
> We implemented a modality-specific-head baseline using the same BAGEL initialization, SPECTRUM-25M data, and sampling strategy. The main change is that we decouple the I/O interface by replacing the shared unified representation with small modality-specific prediction and output heads (e.g., 2D projectors and head, and grounding regression heads, dino head).
>
> | Architecture Variant | I/O Formulation | MMMU↑ | DIODE depth ↓ | NYUv2 Normal ↓ |
> | --- | --- | --- | --- | --- |
> | BAGEL + Specific Heads | Decoupled | 50.4 | 0.322 | 50.75 |
> | **MODUS (Ours)** | Unified Sequence | **51.1** | **0.285** | **19.92** |
>
> The results suggest that decoupling modalities into separate heads fragments the representation space and weakens cross-modal transfer. By contrast, the unified sequence formulation is important for preserving cross-modal generation while also keeping the training setup simpler, since it avoids separate modality routing and per-head output design.
>
> We will include this comparison in the revision to clarify the contribution boundary more clearly.

---

> > ### Author Rebuttal · Reviewer_Bs82 · 2026-04-03
> >
> > Thanks the authors for the feedback during rebuttal. Most of my concerns have been addressed. The additional experiments should make this work stronger. I will keep my initial rating.

---

> > > ### Author Response · Authors · 2026-04-08
> > >
> > > We thank the reviewer for the thoughtful feedback and for confirming that most concerns have been addressed. We appreciate the recognition of the decoder-only any-to-any direction as "meaningful" and the unified architecture as "clean," and are glad the additional experiments further strengthened the work.
> > >
> > > We will incorporate all clarifications and additional experimental results from the rebuttal into the revised manuscript, and release the code, model weights, and SPECTRUM-25M dataset. Thank you again for the valuable feedback.

---

### Decision · Program_Chairs · 2026-04-30

**Decision:**

Accept (regular)

**Comment:**

Based on four reviews and the authors’ rebuttal, I recommend **Accept**. The paper presents a clean decoder-only any-to-any multimodal model that unifies discrete autoregression for 1D modalities and continuous flow matching for 2D modalities without modality-specific heads. The rebuttal effectively clarifies architectural choices (uniform timestep sampling, joint ViT+VAE representation, staged training), adds missing comparisons (BAGEL with modality-specific heads, Janus-Flow extension, chained generation efficiency), and addresses data contamination and preservation of image-text capabilities. Overall, the paper’s contributions—enabling flexible chaining, cross-modal verification, and multi-condition generation within a single pretrained decoder—outweigh its limitations, and the work is likely to be built upon by the community.